# Score-Repellent Monte Carlo: Toward Efficient Non-Markovian Sampler with Constant Memory in General State Spaces

Jie Hu [1]   Lingyun Chen [2]   Geeho Kim [3]   Jinyoung Choi [5]   Bohyung Han [3 4]   Do Young Eun [2]

## Abstract

History-dependent sampling can reduce long-run Monte Carlo variance by discouraging redundant revisits, but existing schemes typically encode history through empirical measure on finite state spaces, which is infeasible in high-dimensional discrete configuration spaces or ill-posed in continuous domains. We propose *Score-Repellent Monte Carlo* (SRMC) framework that summarizes trajectory history by a running average of score evaluations in $\mathbb{R}^d$, where $d$ is the dimension of the score and state representation. This history is converted into a surrogate target through an exponential *score tilt*, indexed with $\alpha$ that represents the *strength of repellence* in controlling the magnitude of the history-based repulsion. The surrogate family is normalization-free in the standard MCMC sense, yielding a generic wrapper: at each iteration, any base kernel targeting $\pi$ can instead be run on the current surrogate $\pi_{\theta_n}$ while the history is updated online. We analyze the coupled evolution of the history recursion and Monte Carlo estimators using stochastic approximation with controlled Markovian noise, establishing almost sure convergence and a joint central limit theorem. We further identify regimes in which the asymptotic covariance decreases as $\alpha$ increases, with scaling $O(1/\alpha)$, extending the near-zero-variance effect of finite-state history-dependent samplers to general state spaces with constant memory. Experiments on continuous targets and discrete energy-based models demonstrate improved estimator variance and mode coverage, while retaining $O(d)$ memory usage and modest per-iteration overhead.

[1]CSE, Oakland University, Rochester, USA [2]ECE, North Carolina State University, Raleigh, USA [3]ECE & [4]IPAI, Seoul National University, Seoul, Korea [5]AIGS, Ulsan National Institute of Science and Technology, Ulsan, Korea. Correspondence to: Do Young Eun <dyeun@ncsu.edu>.

*Proceedings of the 43rd International Conference on Machine Learning*, Seoul, South Korea. PMLR 306, 2026. Copyright 2026 by the author(s).

## 1. Introduction

Markov chain Monte Carlo (MCMC) remains the workhorse for sampling from complex probability models, enabling Bayesian inference, probabilistic generative modeling, and uncertainty quantification across machine learning, physics, chemistry, and biology (Andrieu et al., 2003). Yet, as targets grow more complicated, e.g., multi-modal posteriors, rugged energy landscapes, and large discrete configuration spaces, the focus of MCMC algorithms is increasingly on sampling efficiency. A sampler can be theoretically ergodic but still be practically trapped; it may spend most of its time revisiting the same subset of states, failing to exploit the target's geometry and, thus, yielding strongly correlated samples and unreliable Monte Carlo estimates.

A large body of work addresses this problem by refining the *Markovian* transition rule. In discrete state spaces, locally informed proposals and balancing strategies improve acceptance rates and reduce backtracking (Zanella, 2020; Sun et al., 2022; Xiang et al., 2023). In continuous domains, gradient-informed dynamics (e.g., Langevin-type methods) and nonreversible samplers reduce random-walk behavior and improve exploration by breaking detailed balance (Neal, 2011; Bierkens, 2016; Hu et al., 2021). However, these methods remain memoryless: when the chain repeatedly returns to the same region, the kernel has no long-range mechanism to 'remember' that it has already been explored many times. This motivates an emerging idea: *use history itself as a control signal* to discourage redundant revisits.

Recent progress in *non-Markovian* sampling provides a striking demonstration of this principle in finite-state settings. Self-Repellent Random Walks (SRRW) (Doshi et al., 2023; Hu et al., 2024) and the History-Driven Target (HDT) framework (Hu et al., 2025) establish a clean theory in which feeding back visit history yields near-zero variance as the repellence strength increases, while retaining the same scale-invariant properties that made classical MCMC methods practical (i.e., operating without normalizing constants). These works are, however, inherently designed for *finite* state spaces (e.g., graph sampling), where history is encoded into the empirical measure $\widehat{\delta}_n \triangleq \frac{1}{n+1} \sum_{i=0}^{n} \delta_{X_i}$. On a finite space $\mathcal{X}$ with $|\mathcal{X}| = N$, $\widehat{\delta}_n$ lies on the probability simplex

$\Delta^N \subset \mathbb{R}^N$, one coordinate per state, so storing $\widehat{\delta}_n$ requires $\Omega(N)$ memory cost.[1] This is benign for moderate-scale graphs, but it becomes prohibitive for the majority of modern MCMC applications: in discrete *configuration* spaces, e.g., $\{0,1\}^d$ or $\{1,\ldots,K\}^d$, $N$ grows exponentially in dimension $d$; and in continuous domains, the empirical measure is a full probability measure (an infinite-dimensional object), so counting the number of visits to each and every state over the past history becomes clearly infeasible.

History dependence has also appeared in sampling over continuous domains, but existing methods typically trade off increased memory requirements or complicated bias-correction schemes for improved repulsion effects. Stein self-repulsive dynamics (Ye et al., 2020) enforce repulsion from a history buffer of samples via pairwise distance evaluations, but both memory and computation scale with the buffer size, and asymptotic unbiasedness is recovered only in the infinite-buffer limit. Adaptive biasing potential/force methods (and metadynamics variants) (Darve & Pohorille, 2001; Valsson et al., 2016; Benaïm & Bréhier, 2016; 2019; Benaim et al., 2020; Hénin et al., 2022) build a history-dependent bias to flatten free energy along a reaction coordinate; while effective in molecular simulations, they usually require discretization, density accumulation and importance reweighting to recover unbiased estimates for the original target, which incurs substantial implementation and tuning overhead. A complementary line of work aims to adaptively *flatten* energy landscapes across strata based on the sampler's own trajectory, e.g., Wang-Landau and stochastic-approximation variants (Wang & Landau, 2001; Liang et al., 2007), with Langevin-type extensions such as contour stochastic gradient Langevin dynamics (SGLD) (Deng et al., 2020).

Another direction modifies Langevin with additional drift terms built from running averages of gradient, e.g., momentum/Adam-style SGLD with adaptive drift (Kim et al., 2022). While such schemes stabilize gradients and establish asymptotic correctness, their formulation is specific to Langevin dynamics; extensions to general MCMC kernels, as well as characterizations of long-run performance (e.g., central limit theorem and asymptotic variance), are not addressed. More broadly, non-Markovian stochastic dynamics such as generalized Langevin dynamics (Jakšić & Pillet, 1997; Jung et al., 2018; Vroylandt & Monmarché, 2022; Vroylandt, 2022) are often motivated by physical modeling rather than general MCMC algorithms with asymptotic convergence to an arbitrarily prescribed target $\pi$.

--------

[1]Hu et al. (2025) mitigates memory cost of $\widehat{\delta}_n$ by keeping an LRU cache of recently visited states on a finite graph, but this only improves constant factors while preserving dependence on the (effective) number of states; its impact on unbiasedness and sampling efficiency remains unknown.

These considerations lead to a concrete design question:

*Can we design long-term, history-dependent sampling methods, akin to SRRW and HDT, that preserve exploration benefits while remaining applicable to general MCMC algorithms in both continuous and discrete configuration spaces, requiring only constant memory, minimal per-iteration cost, and maintaining the normalization-free implementation?*

Our approach is to record history not through state counts (an $|\mathcal{X}|$-dimensional object) but as a running average of *scores* (a $d$-dimensional object). Concretely, for a differentiable target density $\pi$ on $\mathcal{X} \subseteq \mathbb{R}^d$ with score $s(x) \triangleq \nabla_x \log \pi(x)$, we maintain $\theta \in \mathbb{R}^d$ as the average score along the trajectory and use it to define a history-dependent *score-tilted* surrogate family $\{\pi_\theta\}$ via an exponential tilt of $\pi$ (formalized in Eq. (2)). This construction is normalization-free in the usual MCMC sense, enabling seamless integration with a broad class of base samplers.

In this paper, we make the following contributions:

- We develop *Score-Repellent Monte Carlo* (SRMC), a general framework that embeds long-term history through constant-memory score averaging and an exponential-tilt surrogate family that is applicable to a broad class of base MCMC kernels in both continuous targets and discrete configuration models;
- We analyze the coupled evolution of the history variable and Monte Carlo estimator through stochastic approximation (SA) with controlled Markovian noise, establishing almost sure convergence and a joint central limit theorem. We further show that, in certain special cases, the limiting covariance scales as $O(1/\alpha)$, where $\alpha \geq 0$ represents the repellence strength (see Eq. (2)). This behavior is conceptually similar to SRRW/HDT (Doshi et al., 2023; Hu et al., 2025), but implemented with score-driven history requiring only $\Omega(d)$ memory, instead of $\Omega(|\mathcal{X}|)$ that may grow as large as $2^d$ or even become infinite;
- We empirically validate SRMC on continuous targets and discrete energy-based models using multiple base samplers. Experiments demonstrate consistent gains in variance reduction and mode exploration, i.e., up to $5\times$ lower MSE for mean estimation in continuous settings and 84% reduction in KL divergence for discrete mode-mixing, while preserving plug-and-play compatibility with standard samplers.

## 2. Score-Repellent Monte Carlo

We first explain the core mechanism of our SRMC framework: (i) a constant-memory history built from score evaluations, and (ii) an exponential tilt that converts this history into a repulsive modification of the target injected into a base sampler.

Let $\pi(x) \propto \exp\{-U(x)\}$ be a twice-differentiable target

distribution on $\mathcal{X} \subseteq \mathbb{R}^d$ with score $s(x) = -\nabla_x U(x)$. A basic identity (a special case of Stein's identity) is that, under standard regularity conditions, $\mathbb{E}_{X \sim \pi}[s(X)] = 0 \in \mathbb{R}^d$, following from integration by parts with vanishing boundary terms (Stein, 1972). Consequently, *a sampler that explores target $\pi$ well should, over time, traverse score values whose empirical mean is close to* 0. If the chain has spent a disproportionate amount of time in some region, it may induce a persistent imbalance in the average of the score field observed along the trajectory so far.

Concretely, we maintain a history variable $\theta_n \in \mathbb{R}^d$ as a running average of the past score evaluations:

$$\theta_{n+1} = \theta_n + \gamma_{n+1}\left(s(X_{n+1}) - \theta_n\right), \gamma_n = (n+1)^{-\rho}, \quad (1)$$

where $\rho \in (\frac{1}{2}, 1]$. For $\rho = 1$, $\theta_n = \frac{1}{n+1}\sum_{i=0}^{n} s(X_i)$ is simply the time average of past score samples. For $\rho < 1$, the update places more weight on recent samples, which can improve responsiveness when the chain is transiently trapped in some subregion. Importantly, the history $\theta_n$ is $d$-dimensional even when $|\mathcal{X}|$ is exponentially large or infinite.

Since $\mathbb{E}_\pi[s(X)] = 0$, the history $\theta_n$ can be viewed as an online estimate of the discrepancy between the empirical distribution of visited states and the target, measured through the score function, i.e.,

$$\theta_n - \mathbb{E}_\pi[s(X)] = \int_{\mathcal{X}} \left[\frac{1}{n+1}\sum_{i=0}^{n} \delta_{X_i}(x) - \pi(x)\right] s(x)dx.$$

Indeed, $\theta_n^\top s(x) = (\theta_n - \mathbb{E}_\pi[s(X)])^\top s(x)$, so the alignment $\theta_n^\top s(x)$ quantifies how strongly the local score at $x$ matches the directions that have been over-represented in the chain's past score observations. Intuitively, if the chain has spent substantial time in a region whose score vectors are concentrated within a particular cone (e.g., repeatedly following similar inward drifts in a metastable basin), then $\theta_n$ points toward that cone, and $\theta_n^\top s(x)$ tends to be positive for points $x$ in that region. Penalizing large positive alignment thus discourages moves that would further amplify the already-dominant score directions, while relatively favoring states whose score directions have been under-explored.

Motivated by this intuition, we adopt the HDT-MCMC principle of modifying the *target* (Hu et al., 2025), but we do so via the $d$-dimensional constant-memory average score $\theta_n$ instead of the empirical measure $\widehat{\delta}_n$. This contrasts with methods that hard-code a history-dependent transition rule, such as SRRW (Doshi et al., 2023) and adaptive drift (Kim et al., 2022). Formally, we define the *score-tilted* surrogate family $\{\pi_\theta\}_{\theta \in \mathbb{R}^d}$ by

$$\pi_\theta(x) \propto \pi(x)\exp\{-\alpha\,\theta^\top s(x)\}, \quad \forall x \in \mathcal{X}, \quad (2)$$

where $\alpha \geq 0$ controls the strength of repellence, and $\pi_\theta(x)$ reduces to the ground-truth $\pi$ at either $\theta = 0$ or $\alpha = 0$. When $\theta^\top s(x) > 0$, the factor $\exp\{-\alpha\theta^\top s(x)\} < 1$ down-weights $x$ under the surrogate, discouraging revisits to the

region around $x$ whose score directions have been repeatedly reinforced by the realized trajectory. The exponential tilt in (2) is chosen for two reasons: (i) it yields a nonnegative target modification that is linear in the score function (analogous to an exponential-family perturbation), and (ii) it preserves the standard *normalization-free* implementation: the normalizing constant

$$Z_\theta \triangleq \int_{\mathcal{X}} \pi(x)\exp\{-\alpha\theta^\top s(x)\}dx \quad (3)$$

cancels out in the Metropolis step and is not required for gradient-based samplers like Langevin dynamics.

Algorithm 1 summarizes SRMC as a generic wrapper: at iteration $n$, we run a base kernel targeting the current surrogate $q = \pi_{\theta_n}$ and then update $\theta_{n+1}$ via (1). Next, we demonstrate the application of SRMC to Metropolis–Hastings (Metropolis et al., 1953; Hastings, 1970) and Langevin dynamics (Roberts & Rosenthal, 1998; Durmus & Moulines, 2017); further examples such as Hamiltonian Monte Carlo and underdamped Langevin, e.g., Neal (2011); Cheng et al. (2018), are provided in Appendix B.

---

**Algorithm 1** Score-Repellent Monte Carlo (SRMC)

---

**Require:** Target $\pi(x) \propto e^{-U(x)}$; score $s(x) = -\nabla U(x)$; strength of repellence $\alpha \geq 0$; step size $\gamma_n$; base MCMC kernel $P_q$.
 1: Initialize $X_0 \in \mathcal{X}$, $\theta_0 \in \mathbb{R}^d$.
 2: **for** $n = 0, 1, 2, \ldots, N-1$ **do**
 3:    Set surrogate target $q = \pi_{\theta_n}$ defined in (2)
 4:    Sample $X_{n+1} \sim P_q(X_n, \cdot)$
 5:    Update $\theta_{n+1} \leftarrow \theta_n + \gamma_{n+1}\left(s(X_{n+1}) - \theta_n\right)$
 6: **end for**
 7: **Output:** Trajectory $\{X_n\}_{n=0}^N$.

---

**Metropolis-Hastings (MH).** Let $q(x, y)$ be a proposal distribution. Replacing $\pi$ with $\pi_\theta$ yields the acceptance probability

$$a_\theta(x, y) = \min\left\{1, \frac{\pi_\theta(y)\,q(y,x)}{\pi_\theta(x)\,q(x,y)}\right\}$$
$$= \min\left\{1, \frac{\pi(y)\,q(y,x)}{\pi(x)\,q(x,y)}e^{-\alpha\theta^\top[s(y)-s(x)]}\right\}, \quad (4)$$

where the surrogate normalizing constant $Z_\theta$ cancels out. Compared to the base MH, the score-repellent MH (SR-MH) algorithm adds the multiplicative factor $e^{-\alpha\theta^\top[s(y)-s(x)]}$ to (4). This term encourages moves that relatively reduce alignment with the score average $\theta$. For instance, the acceptance probability of the proposed $y$ increases when $\theta^\top[s(y)-s(x)] < 0$, and vice versa.

**Langevin dynamics.** Replacing the target $\pi$ with $\pi_\theta$ in Langevin dynamics yields the time-inhomogeneous stochas-

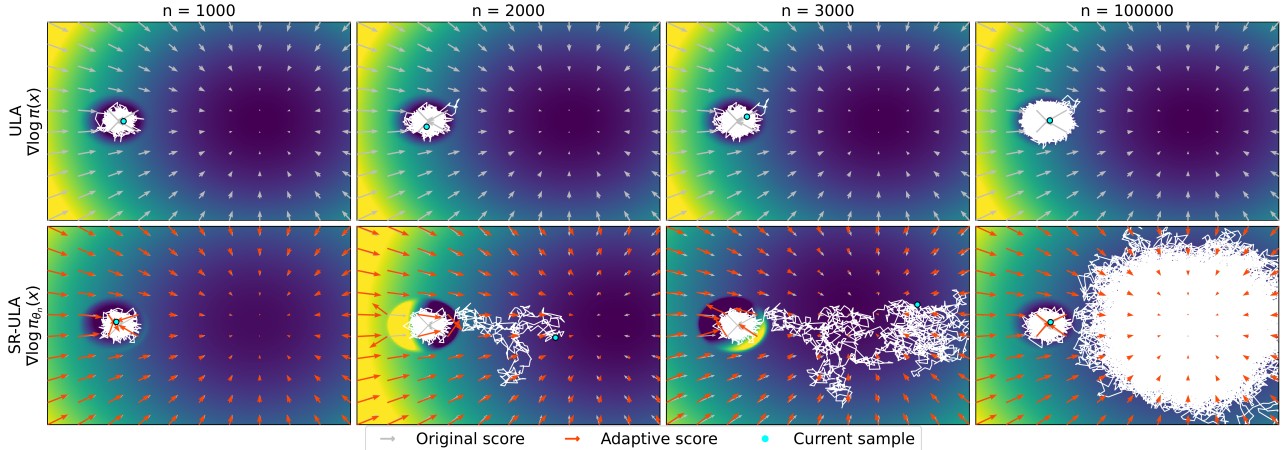

*Figure 1.* **Score-repellent adaptation reshapes the score field to escape a metastable trap.** We consider a two-dimensional, two-mode target distribution $\pi$ (a Gaussian mixture with an imbalanced structure: a dominant narrow mode forming a deep energy well on the left, and a broader mode on the right). Background color shows the (unnormalized) log-density landscape (darker indicates higher density / lower energy). Each column visualizes an increasing iteration index $n \in \{1k, 2k, 3k, 100k\}$. *Top row:* An ULA sampler driven by the original score field $\nabla_x \log \pi(x)$ (gray arrows) remains trapped near the narrow high-density mode; its trajectory (white curve) stays localized even after $100k$ steps. *Bottom row:* SR-ULA uses the base ULA sampler but targets the history-adapted surrogate in Eq. (2). Gray arrows depict the original score field, while red arrows depict the adaptive score field $\nabla_x \log \pi_{\theta_n}(x)$; the cyan dot indicates the current position. As the chain lingers in the left well, the running score average $\theta_n$ accumulates the frequently reinforced directions, and the resulting exponential tilt reduces the effective attraction of that region, visibly deforming the local vector field and pushing the trajectory outward. By $n \approx 2k$, SR-ULA escapes the metastable basin and begins exploring the broader region, illustrating how constant-memory score history can dynamically lower practical 'energy barriers' without explicit state counting. At long times $n = 100k$, SR-ULA has already thoroughly explored the right mode and returned to the left mode, leading to a balanced average score with $\theta_n \approx 0$, while the adapted field drifts back toward the score field $s(x)$ (the red and gray arrows almost overlap).

tic differential equation (SDE)[2]

$$dX_t = s_{\theta_t}(X_t)\, dt + \sqrt{2}\, dB_t, \quad d\theta_t = (s(X_t) - \theta_t) dt$$

where $(B_t)_{t>0}$ denotes standard Brownian motion, $\theta_t$ denotes the continuously evolving analogue of (1), and $s_\theta(x)$ represents the surrogate score given by

$$s_\theta(x) \triangleq \nabla_x \log \pi_\theta(x) = s(x) - \alpha \nabla_x s(x)\theta$$
$$= -\nabla_x U(x) + \alpha \nabla_x^2 U(x)\, \theta, \qquad (5)$$

where $\nabla_x^2 U(x)$ is the Hessian of the potential $U(x)$. Thus, for gradient-driven base samplers, score-repellence is implemented by replacing the original score $s$ with the history-dependent score $s_\theta$. Applying Euler-Maruyama to the above SDE yields the score-repellent unadjusted Langevin algorithm (SR-ULA), and adding a Metropolis correction gives the corresponding SR-MALA algorithm targeting $\pi_{\theta_n}$.

Computationally, evaluating $s_{\theta_n}(x)$ requires the Hessian-vector product $\nabla_x^2 U(x)\theta_n$, which can be computed efficiently by automatic differentiation (Pearlmutter, 1994; Griewank & Walther, 2008) or approximated by finite differences of gradients:

$$\nabla_x^2 U(x)\,\theta \approx (\nabla_x U(x + \epsilon\theta) - \nabla_x U(x))/\epsilon \qquad (6)$$

for small $\varepsilon > 0$. A central difference scheme that uses $\nabla_x U(x + \epsilon\theta/2)$ and $\nabla_x U(x - \epsilon\theta/2)$ can provide higher

---

[2]SRMC applies equally to any Langevin-type sampler whose invariant law is $\pi$, including the 'complete recipe' framework of Ma et al. (2015). This is done by substituting the true score $s(x)$ with the surrogate score $s_\theta(x)$ and then updating $\theta_t$ online.

accuracy when needed, but it incurs extra cost because the score must also be evaluated at the backward point $x - \epsilon\theta/2$.

Figure 1 provides a toy metastability vignette illustrating the effect of the score-tilt surrogate $\pi_\theta$: the history-dependent tilt transiently deforms the local drift field near an overvisited basin, facilitating barrier crossing. For example, in the bottom plot of Figure 1 at $n = 2000$, the two red arrows on the right-hand side of the left mode flip their directions: instead of pointing inward toward the center of the left mode, they now point outward toward the right mode. The full simulation setup is provided in Appendix A.

## 3. Theoretical Analysis

SRMC in Algorithm 1 can be naturally viewed as a *stochastic approximation (SA) scheme with controlled Markovian noise* (Kushner & Yin, 2003; Benveniste et al., 2012; Borkar, 2009; Borkar et al., 2025). This perspective also appears in recent finite-state, history-dependent samplers such as SRRW and HDT-MCMC (Doshi et al., 2023; Hu et al., 2025), but our framework departs from those in a key way: SRMC is defined on general state spaces $\mathcal{X} \subseteq \mathbb{R}^d$, where the surrogate family $\pi_\theta$ must stay well-defined, and the resulting kernel family $\{P_\theta\}$, which has $\pi_\theta$ as its invariant distribution, must satisfy appropriate control conditions for SA limit theory to apply. Unlike finite-state settings where the drift condition is often automatic, we make explicit

assumptions on the surrogate family and on the kernel regularity with respect to $\theta$ through verifiable conditions for our theoretical analysis, and then leverage modern SA results to obtain both almost-sure convergence and a central limit theorem (CLT).

For any test function $f : \mathcal{X} \to \mathbb{R}^m$ in $L^1(\pi)$, our goal is to estimate $\mu \triangleq \mathbb{E}_\pi[f(X)]$. SRMC generates an inhomogeneous Markov chain $\{X_n\}_{n \geq 0}$ through the recursion $X_{n+1} \sim P_{\theta_n}(X_n, \cdot)$ and updates the score-history variable $\theta_n$ and the running Monte Carlo estimator $\mu_n$ by

$$\begin{bmatrix} \theta_{n+1} \\ \mu_{n+1} \end{bmatrix} = \begin{bmatrix} \theta_n \\ \mu_n \end{bmatrix} + \gamma_{n+1} \begin{bmatrix} s(X_{n+1}) - \theta_n \\ f(X_{n+1}) - \mu_n \end{bmatrix}, \quad (7)$$

The step size $\gamma_n$ is as in Eq. (1) and satisfies the Robbins-Monro conditions $\sum_n \gamma_n = \infty$ and $\sum_n \gamma_n^2 < \infty$.

Define the joint iterate $\vartheta_n \triangleq (\theta_n, \mu_n) \in \mathbb{R}^{d+m}$ and the update function $H(\vartheta_n, x) \triangleq (s(x) - \theta_n, \ f(x) - \mu_n)$ for $x \in \mathcal{X}$, so that (7) can be written in the compact SA form

$$\vartheta_{n+1} = \vartheta_n + \gamma_{n+1} H(\vartheta_n, X_{n+1}). \quad (8)$$

For each fixed $\theta$, write

$$\mathcal{S}(\theta) \triangleq \int_\mathcal{X} \pi_\theta(x) s(x) dx, \quad \mathcal{F}(\theta) \triangleq \int_\mathcal{X} \pi_\theta(x) f(x) dx,$$

and define the associated mean field

$$h(\vartheta) \triangleq \mathbb{E}_{X \sim \pi_\theta}[H(\vartheta, X)] = (\mathcal{S}(\theta) - \theta, \mathcal{F}(\theta) - \mu) \in \mathbb{R}^{d+m}.$$

At $\theta = 0$, we recover $\mathcal{F}(0) = \mathbb{E}_\pi[f(X)] = \mu$, and the score identity $\mathcal{S}(0) = \mathbb{E}_\pi[s(X)] = 0$, so $\vartheta^\star \triangleq (0, \mu)$ is the root of the mean-field equation $h(\vartheta^\star) = 0$. Define the covariance matrix of two vector-valued functions $f$ and $g$ under $\pi$ as $\text{Cov}_\pi(f, g) \triangleq \mathbb{E}_\pi[f(X) g(X)^\top] - \mathbb{E}_\pi[f(X)] \mathbb{E}_\pi[g(X)]^\top$.

We now state the following assumptions.

**Assumption 1** (Properties of target distribution $\pi$). For $\mathcal{X} = \mathbb{R}^d$, the score $s(x) = -\nabla_x U(x)$ is $L$-Lipschitz:

$$\|s(x) - s(x')\| \leq L\|x - x'\|, \quad \forall, x, x' \in \mathbb{R}^d. \quad (9)$$

The potential energy $U(x)$ satisfies:
  (i) *Super-linear tail growth:* $U(x) \geq c\|x\|^p$ for $\|x\| \geq R$, for some $p \in (1, 2]$, $c > 0$, $R > 0$.
  (ii) *Asymptotic regularity:* $r^{-(p-2)} \nabla_x^2 U(r\hat{x}) \to M(\hat{x})$ as $r \to \infty$ for each $\hat{x} \in \mathbb{S}^{d-1} \triangleq \{x \in \mathbb{R}^d : \|x\| = 1\}$, for some measurable positive definite matrix $M : \mathbb{S}^{d-1} \to \mathbb{R}^{d \times d}$.

**Assumption 2** (Uniform drift and kernel Lipschitzness in Borkar et al. (2025)). The kernel $P_\theta$ is geometrically ergodic with invariant distribution $\pi_\theta$, satisfies the uniform drift condition (DV3) in Eq. (32), and is Lipschitz in $\theta$ as stated in Eq. (34); see Appendix C.3 for exact expressions.

The L-Lipschitz score function in Assumption 1 is standard in MCMC theory (Durmus & Moulines, 2017; Cheng et al., 2018; Riabiz et al., 2022; Chak et al., 2023). The regularity on $U(x)$ guarantees both that $\pi_\theta$ is well-defined and that the joint iterates $\vartheta_n$ remain bounded. This assumption is fairly mild and in fact covers many commonly used target

distributions, such as (generalized) Gaussian and Gaussian mixture models (Saumard & Wellner, 2014; Liang et al., 2025), as well as Bayesian logistic regression posteriors with Gaussian priors (Durmus & Moulines, 2019) and other strongly log-concave distributions. A detailed discussion of the tail regularity condition is deferred to Appendix C.1.

Assumption 2 is the main technical point that differentiates the continuous-domain theory from finite-state history-driven samplers (see Remark 1). Drift conditions are the standard route to geometric ergodicity in general state spaces (Meyn & Tweedie, 2012, Chapter 16), and uniform versions (often called *simultaneous drift*) are common in adaptive MCMC and SA with controlled Markovian noise (Andrieu & Moulines, 2006; Roberts & Rosenthal, 2007; Fort et al., 2011). The kernel Lipschitz condition is less standard in MCMC because $\theta$ enters only through the surrogate target $\pi_\theta$. As part of our technical contributions, we establish verifiable sufficient conditions that guarantee this kernel Lipschitz property for two standard sampling algorithms, MH and Metropolis-adjusted Langevin in Appendix C.3.

*Remark* 3.1 (On the role and interpretation of assumptions). Assumptions 1 and 2 summarize the *verifiable* conditions we impose within the SRMC framework. While the SA results of Borkar et al. (2025) are stated in a generic controlled-Markovian noise setting, their hypotheses are not directly formulated in MCMC terms and therefore do not immediately yield checkable criteria for our score-tilted construction. One of technical contributions of this work is to *interpret and specialize* those generic conditions to history-adapted MCMC: we translate the required stability and kernel-regularity requirements into assumptions on the surrogate family $\{\pi_\theta\}$ and the induced kernel family $\{P_\theta\}$ that can be verified via standard drift arguments. In particular, this specialization allows us to *prove* boundedness (stability) of the SRMC iterates rather than assuming it a priori, thereby removing a stability assumption commonly imposed in earlier analyses of non-Markovian samplers (Doshi et al., 2023; Hu et al., 2025) based on classical SA theory (Delyon, 2000; Borkar, 2009; Fort, 2015).

On finite state spaces, the normalizing constant $Z_\theta < \infty$ is automatic (since it is a finite sum), so that $\pi_\theta$ is always well-defined for any $\theta \in \mathbb{R}^d$. On $\mathcal{X} = \mathbb{R}^d$, Assumption 1 is enough to ensure that $\pi_\theta$ is well-defined.

**Lemma 3.2** (Well-posedness of $\pi_\theta$). *Under Assumption 1, for every $\theta \in \mathbb{R}^d$ and $\alpha > 0$, we have $Z_\theta < \infty$.*

Let $\{Z_k : k \in \mathbb{Z}\}$ denote a trajectory generated by the base Markov chain $P = P_0$ with its invariant distribution $\pi = \pi_0$. Define the limiting covariance of the sequence $H(\vartheta^\star, Z_k)$

$$\Sigma_\Delta \triangleq \lim_{T \to \infty} \frac{1}{T} \mathbb{E}\left[ \left( \sum_{k=0}^T \Delta_k \right) \left( \sum_{k=0}^T \Delta_k \right)^\top \right], \quad (10)$$

where $\Delta_k \triangleq H(\vartheta^\star, Z_k) - h(\vartheta^\star) = H(\vartheta^\star, Z_k)$. The Jacobian of the mean field $A^\star \triangleq \nabla_\vartheta h(\vartheta^\star)$ then becomes

$$A^\star = \begin{bmatrix} -I_d - \alpha \operatorname{Cov}_\pi(s,s) & 0 \\ -\alpha \operatorname{Cov}_\pi(f,s) & -I_m \end{bmatrix}. \quad (11)$$

See Appendix C.6 for the derivation. We now state our main theorem for the coupled SA iterate $\vartheta_n = (\theta_n, \mu_n)$, which provides both almost sure convergence and a joint CLT.

**Theorem 3.3** (Almost sure convergence and CLT of $\vartheta_n$). *Under Assumptions 1 and 2, the sequence $\vartheta_n$ from (8) converges almost surely to the unique equilibrium $\vartheta^\star = (0, \mu)$.*

*Moreover, the normalized error admits the CLT*

$$\gamma_n^{-1/2}(\vartheta_n - \vartheta^\star) \xrightarrow[n\to\infty]{dist.} \mathcal{N}(0, \Sigma_\vartheta), \quad (12)$$

*where $\Sigma_\vartheta \in \mathbb{R}^{(d+m)\times(d+m)}$ is the unique positive semidefinite solution to the Lyapunov equation*

$$\left(\tfrac{1_{\rho=1}}{2}I + A^\star\right)\Sigma_\vartheta + \Sigma_\vartheta\left(\tfrac{1_{\rho=1}}{2}I + A^\star\right)^\top + \Sigma_\Delta = 0, \quad (13)$$

*where $\rho$ is specified in (1).*

Theorem 3.3 follows by instantiating the general SA theory of Borkar et al. (2025) to the SRMC recursion (8). The main technical novelty is to verify, on $\mathcal{X} = \mathbb{R}^d$, the Lipschitz property on the family of transition kernels $\{P_\theta\}$ targeting our surrogate $\pi_\theta$, and the stability conditions (via an ODE@$\infty$ technique used in Borkar & Meyn (2000); Borkar et al. (2025)) needed to control parameter-dependent Markovian noise and ensure the boundedness of $\{\vartheta_n\}$. Full details of the proof of Theorem 3.3 are provided in Appendix C.8.

Theorem 3.3 has two complementary implications. First, the almost sure limit $\theta_n \to 0$ implies that the history-dependent surrogate $\pi_{\theta_n}$ converges to the true target $\pi$, and $\mu_n \to \mu$ in turn demonstrates that SRMC yields asymptotically unbiased Monte Carlo estimates for any integrable test function $f$. Second, the CLT in Theorem 3.3 quantifies the *long-run fluctuations* of the coupled iterate $\vartheta_n = (\theta_n, \mu_n)$ through the asymptotic covariance $\Sigma_\vartheta$, thereby determining the asymptotic efficiency of SRMC.

A useful structural feature of the Lyapunov equation (13) is that the noise covariance $\Sigma_\Delta$ in (10) is independent of the strength of repellence $\alpha$. Thus, all $\alpha$-dependence in $\Sigma_\vartheta$ enters through matrix $A^\star$ in (11), which depends on $\alpha$ only via the covariance blocks $\operatorname{Cov}_\pi(s,s)$ and $\operatorname{Cov}_\pi(f,s)$. Intuitively, the $\theta$-marginal covariance is governed by the symmetric drift $-I_d - \alpha \operatorname{Cov}_\pi(s,s)$, implying that increasing $\alpha$ strengthens contraction in directions where $\operatorname{Cov}_\pi(s,s)$ has positive spectrum. The $\mu$-marginal covariance depends on $\alpha$ through both $\operatorname{Cov}_\pi(f,s)$ (coupling between $\theta_n$ and $\mu_n$) and the induced $\theta$ fluctuations. These observations lead us to the fine-grained quantification as follows.

**Proposition 3.4.** *Let $\Sigma_{\theta\theta}(\alpha) \in \mathbb{R}^{d\times d}$ denote the $\theta$-block (top-left block) of $\Sigma_\vartheta$. Then $\Sigma_{\theta\theta}(\alpha) = O(1/\alpha)$ as $\alpha \to \infty$*

*(entrywise). Moreover, for any $\alpha_1 \geq \alpha_2 \geq 0$, we have*

$$\|\Sigma_{\theta\theta}(\alpha_1)\|_F \leq \|\Sigma_{\theta\theta}(\alpha_2)\|_F \leq \|\Sigma_{\theta\theta}(0)\|_F, \quad (14)$$

*where $\|\cdot\|_F$ represents the Frobenius norm.*

See Appendix C.9 for proofs and additional details regarding the scaling of the other subblock matrices. Proposition 3.4 shows that increasing $\alpha$ monotonically reduces the limiting variance of the score-history iterate $\theta_n$ and, in the large-$\alpha$ regime, yields an $O(1/\alpha)$ scaling, i.e., near-zero asymptotic covariance of $\gamma_n^{-1/2}\theta_n$ — the scaled running average of $s(X_n)$. While the base MCMC sampler ($\alpha = 0$) typically produces (positively) correlated samples, our SRMC induces negative correlations in the sequence $\{s(X_n)\}$. This is also in line with the geometric behavior in Figure 1: stronger repellence produces stronger negative feedback against repeatedly reinforced score directions (the 'arrow flipping' effect), thereby reducing long-run variability in the history statistic. Although similar qualitative monotonicity is observed in finite-state history-dependent samplers such as SRRW and HDT-MCMC (Doshi et al., 2023; Hu et al., 2025), SRMC achieves it through the discrepancy of the empirical score average from its target value $\mathbb{E}_\pi[s(X)] = 0$, without storing empirical visit counts for every state in $\mathbb{R}^d$.

We next turn to the Monte Carlo estimator covariance $\Sigma_{\mu\mu}(\alpha) \in \mathbb{R}^{m\times m}$ (the bottom-right block of $\Sigma_\vartheta$). In general, $\Sigma_{\mu\mu}(\alpha)$ is obtained implicitly as part of the Lyapunov solution (13) and does not admit a closed-form dependence on $\alpha$ for arbitrary base kernels and test functions. Nevertheless, in several instructive cases as below, we can make it explicit, with full details deferred to Appendix C.10.

**Independent surrogate sampling.** If the base kernel draws $X_{n+1} \sim \pi_{\theta_n}$ independently at each step, then

$$\Sigma_{\mu\mu}(\alpha) = \operatorname{Cov}_\pi(f,s)\, M(\alpha)^{-1} \operatorname{Cov}_\pi(s,f) + R, \quad (15)$$

where $M(\alpha) \triangleq \operatorname{Cov}_\pi(s,s)\big(2\alpha \operatorname{Cov}_\pi(s,s) + (2 - 1_{\rho=1})I\big)$ and $R \succeq 0$ is a residual term independent of $\alpha$, i.e., the entire $\alpha$-effect resides only in $M(\alpha)^{-1}$. Thus, $\Sigma_{\mu\mu}(\alpha)$ is never larger than that of the base sampler and converges to the matrix $R$ in the Loewner order (and thus also in Frobenius norm) at rate $O(1/\alpha)$ as $\alpha$ increases, where $A \preceq B$ denotes the Loewner order, i.e., $v^\top A v \leq v^\top B v$ for every vector $v$.

**Gaussian target and mean estimation.** If $\pi = \mathcal{N}(\mu, V)$, for the sequence of samples $\{X_i\}$ drawn from SRMC applied to a general base sampler, we have the following CLT:

$$\sqrt{n}\left(\frac{1}{n}\sum_{i=1}^n X_i - \mathbb{E}_\pi[X]\right) \xrightarrow[n\to\infty]{dist.} \mathcal{N}(0, \Sigma_X(\alpha)), \quad (16)$$

where the limiting covariance of the sample mean $\Sigma_X(\alpha) = V\Sigma_{\theta\theta}(\alpha)V^\top$. Thus, Proposition 3.4 directly yields that $\Sigma_X(\alpha) = O(1/\alpha)$ as $\alpha \to \infty$. The key observation is that for Gaussian targets, the score is affine in state $x$, thus the CLT

for the score history $\theta_n$ in Proposition 3.4 transfers directly to the sample mean via linear transformation, rendering near-zero asymptotic covariance in the large-$\alpha$ regime. See Appendix C.10.2 for the detailed derivation.

*Remark* 3.5. For an arbitrary target $\pi$ in general state spaces and general base kernels, such explicit formulas are typically unavailable; we discuss the sources of this difficulty and further intuition in Appendix C.10. More broadly, SRMC trades an infeasible infinite-dimensional history (the full empirical measure) for a constant-memory score average; while this compression precludes closed-form variance characterizations in full generality, our preceding theoretical results and empirical findings in the next section suggest that our SRMC framework can bring reduced asymptotic variance relative to the base sampler for suitable $\alpha$ choices. We also provide in Appendix C.11 a cost-based discussion, using a fixed computational budget perspective, to clarify how the variance reduction of the gradient-based SRMC should be interpreted when each iteration incurs additional overhead from the Hessian–vector product evaluation.

**Score function in discrete configuration spaces.** SRMC maintains an online score average $\theta_n$ and uses it to define an exponential-tilt surrogate family $\{\pi_\theta\}$. A key requirement for our analysis is that the score feature satisfy $E_\pi[s(X)] = 0$, so that $\theta^\star = 0$ is the correct equilibrium and $\pi_{\theta_n}$ converges back to the true target $\pi$. In continuous domains, this follows from Stein's identity. In discrete spaces, however, a relaxed gradient $-\nabla_x U(x)$ need not have zero mean under $\pi$, even though such gradients are often useful for proposal construction in discrete samplers, e.g., Gibbs-with-Gradients (GWG) and other locally-balanced or Langevin-type schemes (Grathwohl et al., 2021; Zanella, 2020; Zhang et al., 2022; Xiang et al., 2023).

A general way to obtain a valid discrete score is through *discrete Stein operators* (Bresler & Nagaraj, 2019; Shi et al., 2022). Consider the discrete configuration space $\mathcal{X} = \{0, 1, \ldots, K\}^d$. For $i \in [d]$ and $k \in \{0, 1, \ldots, K\}$, let $x^{(i,k)}$ denote the state obtained from $x$ by replacing only its $i$-th coordinate by $k$. In line with the construction in Shi et al. (2022, Section 3.1 Eq. (7)), one convenient option among many for the discrete score $s(x) = [s_i(x)]_{i=1}^d$ is

$$s_i(x) \triangleq \left( \frac{\pi(x^{(i,K-x_i)})}{\pi(x)} - 1 \right). \tag{17}$$

This depends only on the ratios of $\pi$ and therefore does not require the normalizing constant. By the reindexing argument over the discrete configuration space, $E_\pi[s_i(X)] = 0$ for every $i \in [d]$. Replacing the score function in Algorithm 1 with the discrete score $s(x)$ yields the following proposition, whose proof is deferred to Appendix C.12.

**Proposition 3.6** (Discrete configuration space extension of Theorem 3.3)**.** *Let $\mathcal{X}$ be a finite discrete state space and $s : \mathcal{X} \to \mathbb{R}^d$ be a discrete score function such that*

$\mathbb{E}_\pi[s(X)] = 0$. *If the kernel family $\{P_\theta\}_{\theta \in \mathbb{R}^d}$ admits $\pi_\theta$ as its invariant distribution and satisfies Assumption 2, then Theorem 3.3 continues to hold true for SRMC on $\mathcal{X}$.*

*Remark* 3.7. In the binary case $\mathcal{X} = \{0, 1\}^d$, the above construction reduces to a single bit-flip per coordinate. When the difference of $\pi(x^{(i,1-x_i)})/\pi(x)$ is small, as similarly done in Grathwohl et al. (2021, Eq. (3)) using the first-order Taylor expansion, $s_i(x)$ behaves like a discrete analog of the $i$-th coordinate derivative of $\log \pi(x)$, i.e., $[\nabla_x \log \pi(x)]_i \cdot (1 - 2x_i)$. This explains the practical approximation used in our Static MNIST experiments in Section 4.3: although the exact theory relies on a discrete score feature satisfying $\mathbb{E}_\pi[s(X)] = 0$, in binary energy-based models the original relaxed score can still be used as an effective proxy for constructing the score-repellent update.

# 4. Experiments

We evaluate SRMC in continuous and discrete settings. Section 4.1 gives a brief tuning guide for continuous samplers, Section 4.2 studies continuous targets under both sampling and computational efficiency, and Section 4.3 studies discrete energy-based models, where the theory uses a zero-mean discrete score but our Static MNIST implementation uses a relaxed-gradient proxy for efficiency.

## 4.1. Hyperparameter Tuning Guide

For gradient-based SRMC, the practical hyperparameters are the stochastic-approximation exponent $\rho$, the finite-difference scale $\epsilon$ in (6), and the repellence strength $\alpha$. Our follow-up sensitivity study suggests the following practical guidance. First, for the history update $\gamma_n = (n + 1)^{-\rho}$, the most reliable transient performance is obtained around $\rho \in \{0.6, 0.8\}$, whereas $\rho = 1$ tends to shrink the step size too quickly and slows the adaptation of $\theta_n$. Second, on nonlinear targets, $\epsilon$ should not be taken overly small: values on the order of $\alpha$ are typically stable, while very small $\epsilon$ may lead to poor finite-difference approximations. Third, $\alpha$ should be chosen so that the exponent scale $\alpha|\theta_n^\top s(X_n)|$ remains moderate along the trajectory, thereby avoiding excessive over-tilting of the surrogate target. The goal of these recommendations is not to identify a universal optimum, but to provide a stable default operating regime. Additional sensitivity results for $\rho$ and $\epsilon$, together with implementation details, are deferred to Appendix D.1.4.

## 4.2. Continuous State Spaces

We evaluate SRMC on two canonical benchmark distributions that test complementary aspects of sampling performance. The first is a correlated Gaussian with ill-conditioned geometry (ellipse) that is difficult for gradient-based samplers. The second is a Bayesian (synthetic) logis-

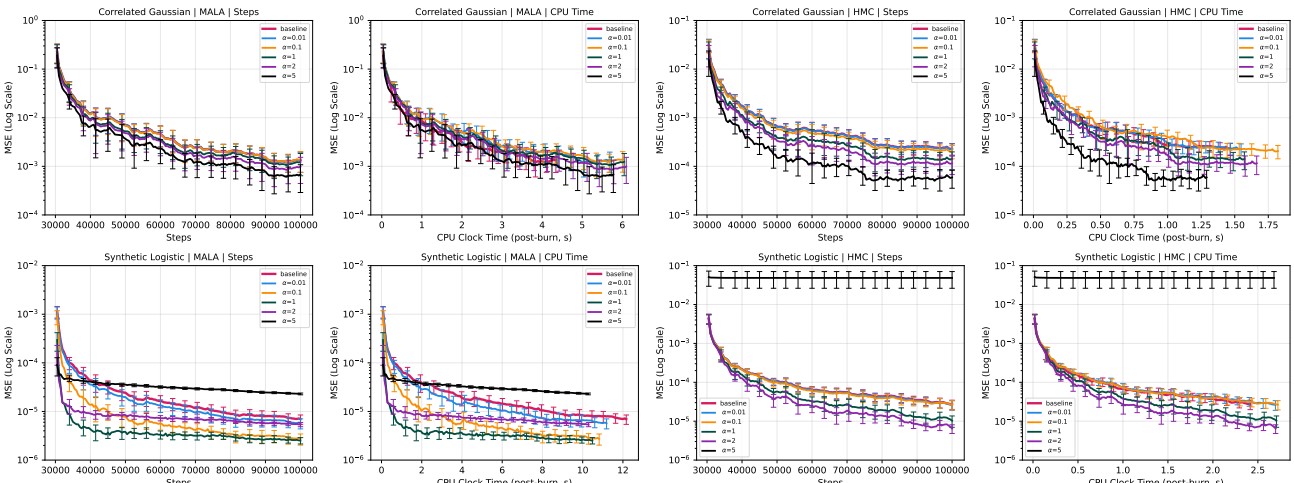

*Figure 2.* MSE for sample-mean estimation on the 10-dimensional correlated Gaussian and synthetic Bayesian logistic regression targets. Top row: correlated Gaussian; bottom row: synthetic logistic regression. Within each row, the first two panels correspond to MALA and SR-MALA, and the last two correspond to HMC and SR-HMC; for each sampler, we report MSE versus post-burn steps and versus post-burn CPU clock time. Curves compare the baseline ($\alpha = 0$) with score-repellent variants using $\alpha \in \{0.01, 0.1, 1, 2, 5\}$. Error bars show 95% confidence intervals over 100 independent runs.

tic regression posterior with 100 observations, representing a realistic inference task with non-trivial uncertainty and complex geometry arising from multicollinearity in the design matrix. Both distributions are in 10 dimensions and detailed simulation setups are provided in Appendix D.1.

We compare baseline MALA and HMC against their score-repellent variants (SR-MALA and SR-HMC) over a wider range of repellence strengths $\alpha \in \{0.01, 0.1, 1.0, 2.0, 5.0\}$. Note that $\alpha = 0$ recovers the baseline algorithm. To separate algorithmic improvement from computational overhead, Figure 2 reports MSE in terms of the number of samples and CPU clock time. Regarding the number of steps, we equalize the total number of gradient evaluations across methods: for MALA, this coincides with the iteration count, while for HMC the horizontal axis corresponds to the total number of leapfrog gradient evaluations (outer iterations times leapfrog steps $L$). The CPU-time results complement this view by reflecting the additional cost of evaluation of surrogate score in SRMC. Reported metrics are averaged over 100 independent runs.

Figure 2 presents both step-based and CPU-time views of MSE convergence. Overall, SRMC improves upon the baseline across most settings, but the practical gain depends on the target, the base sampler, and the repellence strength. On the correlated Gaussian target, the improvement is modest for MALA and more pronounced for HMC: larger repellence ($\alpha = 2$ or $5$) yields the lowest terminal MSE in both the step and CPU-time settings, whereas smaller values ($\alpha = 0.01$ or $0.1$) provide only limited separation from baseline. The CPU-time plots show that these gains remain visible after accounting for the additional surrogate-score cost, although the advantage is narrower for MALA.

For Bayesian logistic regression, the benefit of score repellence is substantially stronger but also more sensitive to tuning. For SR-MALA, intermediate-to-large values ($\alpha = 1$ or $2$) give the best performance in both views, while $\alpha = 5$ behaves too aggressive and is trapped due to the high rejection rate in the MH step. The same pattern is even clearer for SR-HMC: moderate repellence ($\alpha \approx 1$ - $2$) produces the fastest MSE decay and the lowest error, whereas $\alpha = 5$ over-tilts the surrogate and performs worse throughout. This indicates that the variance-reduction effect is still prominent under CPU-time comparison, while the best practical regime is typically moderate rather than maximal, especially on the nonlinear logistic target. Motivated by the transient performance sensitivity, we also examined adaptive-$\alpha$ heuristics that cap early over-tilting while retaining the benefit of stronger repellence later in the run. We defer the discussion about adaptive $\alpha$ as a robust choice when the stable fixed-$\alpha$ regime for a given target distribution is unknown a priori to Appendix D.1.4.

Additional experiments in Appendix D.2 examine mode coverage in CIFAR-10-based continuous energy landscapes. In the synthetic Gaussian-mixture benchmark built from 1,000 CIFAR-10 images, SR-ULA achieves complete discovery of all modes in approximately 1,035 steps, whereas ULA plateaus at only 2.8% coverage. On a pre-trained CIFAR-10 energy-based model, SR-ULA also improves both single-chain trajectory diversity and final-state class coverage. In particular, the parallel-chain study in Appendix D.2.3 shows that the gain becomes more pronounced when the number of available chains is reduced: with only 10 chains, SR-ULA covers 7/10 classes versus 5/10 for ULA, and also yields better KL, total variation, and normalized entropy. This suggests that the score-repellent mechanism can reduce the

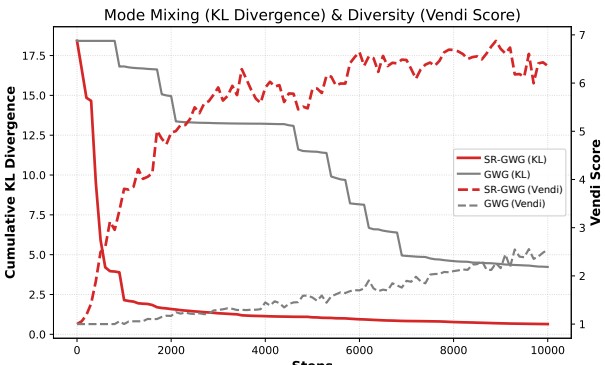

*Figure 3.* Mode mixing (solid) and diversity (dashed) evaluation on Static MNIST.

number of parallel chains needed to obtain useful mode diversity in practice.

### 4.3. Discrete Energy-based Models

We assess SRMC on the Static MNIST dataset using a discrete energy-based model (EBM) on $\{0, 1\}^{784}$. For a fair and standardized comparison, our implementation is built upon the official GWG codebase (Grathwohl et al., 2021), with the sampler modified to include the score-repellent factor targeting the tilted surrogate $\pi_{\theta_n}$; we refer to this implementation as SR-GWG.

To match the theory in Section 3 exactly, the discrete SRMC update should use a discrete score $s(x)$ in (17) satisfying $E_\pi[s(X)] = 0$. However, in our current setup this feature is computationally expensive, since it requires evaluating the target density $\pi(y)$ at every one-bit neighbor $y$ of a state $x$. As the dimension $d$ grows, this becomes prohibitive and leads to issues similar to those in Grathwohl et al. (2021, Section 2.2). Therefore, we use the relaxed gradient as a proxy for the discrete score feature when forming the score-repellent update in our Static MNIST experiments. As a result, the discrete experiments should be interpreted as evaluating an approximate SRMC variant that is computationally practical for high-dimensional EBMs, rather than as a literal implementation of Proposition 3.6. A complete description of the implementation is given in Appendix B.5.3, and the experimental setup details are provided in Appendix D.3.2.

We design a test of mode exploration by initializing all 100 parallel Markov chains from a single real image of the digit '7' and observing their evolution over 10,000 sampling steps. This worst-case initialization forces the sampler to overcome substantial energy barriers to discover other digit classes, providing a rigorous assessment of mode-hopping capability. We quantify performance using two complementary metrics: cumulative KL divergence ($\downarrow$), which measures how effectively accumulated samples approach *uniform* coverage over all ten digit classes, and batch Vendi Score ($\uparrow$) (Friedman & Dieng, 2023), which measures instantaneous diversity

within the parallel chains at each time step.

Figure 3 presents the results. SRMC with $\alpha = 10^{-4}$ exhibits rapid KL decay, with chains escaping the initial mode within 2,500 steps, while baseline GWG maintains persistently high KL divergence throughout, indicating severe mode-trapping. The Vendi Score reinforces this finding: SRMC sustains high diversity (Vendi Score $\approx 6$) across the batch, whereas baseline GWG achieves only $\approx 3$, reflecting strong batch collapse near initialization. At $T = 10,000$, SRMC reduces KL divergence by 84% (0.68 vs. 4.16) and improves Vendi Score from 2.6 to 6.4. Qualitatively, SRMC chains exhibit meaningful digit-to-digit transitions, while baseline remains concentrated around the initial digit.[3]

Qualitative trajectory visualizations displaying sample snapshots at checkpoints $n \in \{0, 2500, 5000, 7500, 10000\}$ are provided in Figure 10 in Appendix D.3.2, illustrating the progressive mode exploration achieved by SRMC. We additionally demonstrate in Appendix D.3.3 that SRMC integrates naturally with Annealed Importance Sampling (AIS) (Neal, 2001) for diverse sample generation, where applying score-repellent dynamics during the annealing process yields improved coverage of the target support.

## 5. Concluding Remarks and Future Works

We introduced Score-Repellent Monte Carlo (SRMC), a history-dependent sampling framework that achieves variance-reduction benefits of non-Markovian methods while requiring only $O(d)$ memory, which is independent of state space cardinality. The key insight is that *score serves as a universal currency* for encoding trajectory history: unlike empirical measures, which scale as $O(|\mathcal{X}|)$ and are ill-posed in continuous domains, the score function is inherently $d$-dimensional and already computed by gradient-based samplers. SRMC maintains a running score average and converts it into repulsive feedback by targeting an exponentially *score-tilted* surrogate target; this construction preserves the normalization-free property and yields a general wrapper around a broad class of base MCMC kernels.

Several directions remain open. Our variance guarantees are asymptotic, so overly large $\alpha$ can hurt finite-horizon performance by inducing a longer calibration phase before $\theta_n$ contracts; developing principled schedules and diagnostic-based tuning rules for $\alpha$ would improve practical robustness. Since SRMC operates by substituting a surrogate score into a base dynamic, it naturally extends to diffusion/score-based generative sampling, where mode exploration and sample diversity remain persistent bottlenecks.

---

[3]Because SR-GWG uses a relaxed-gradient proxy rather than the exact discrete score in (17), these results should be interpreted as empirical evidence that the score-repellent mechanism remains effective even under a practical approximation.

## Impact Statement

This paper presents work whose goal is to advance the field of Machine Learning. There are many potential societal consequences of our work, none of which we feel must be specifically highlighted here.

## Acknowledgment

We thank the anonymous reviewers for feedback that improved the camera-ready version, including clarifying the theory in discrete state spaces, adding tuning guidance and an adaptive-$\alpha$ scheme, and introducing a cost-based CLT with wall-clock comparisons for gradient-based SRMC. Jie Hu was supported in part by Oakland University startup funds and a 2026 Oakland University Research Committee (URC) Faculty Research Fellowship. Lingyun Chen and Do Young Eun were supported in part by the National Science Foundation (NSF) under Grant IIS-2421484. Jinyoung Choi was supported by the Institute of Information & Communications Technology Planning & Evaluation (IITP) grant funded by the Korea government (MSIT) (No. RS-2020-II201336, Artificial Intelligence Graduate School Support (UNIST)). Geeho Kim and Bohyung Han were partly supported by the Brain Pool program funded by the Ministry of Science and ICT through the National Research Foundation of Korea (RS-2024-00408610) and by the Institute of Information & Communications Technology Planning & Evaluation (IITP) grant funded by the Korea government (MSIT) (RS-2022-II220959, No. 2022-0-00959).

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

# A. Simulation Setups for Figure 1

Here we provide detailed simulation settings for the illustrative example in Figure 1.

**Target Distribution.** The target distribution $\pi(x)$ is a two-dimensional Gaussian mixture with two components:

$$\pi(x) = w_1 \mathcal{N}(x; \mu_1, \Sigma_1) + w_2 \mathcal{N}(x; \mu_2, \Sigma_2), \tag{18}$$

where:

- *Component 1 (narrow trap):* $w_1 = 0.8$, $\mu_1 = (-2, 0)$, $\Sigma_1 = 0.0324 \cdot I_2$ (i.e., $\sigma_1 = 0.18$).

- *Component 2 (broad mode):* $w_2 = 0.2$, $\mu_2 = (+2, 0)$, $\Sigma_2 = 1.0 \cdot I_2$ (i.e., $\sigma_2 = 1.0$).

This mixture creates an imbalanced energy landscape: Component 1 forms a deep, narrow potential well that tends to trap standard Langevin samplers, while Component 2 represents a broader, shallower mode. The true mean of the target is

$$\mathbb{E}_\pi[X] = w_1 \mu_1 + w_2 \mu_2 = (-1.2, 0). \tag{19}$$

**Sampler Configurations.** For Figure 1, both ULA and SR-ULA (see Algorithm 3 for the exact pseudocode) are initialized at $X_0 = (-2, 0)$ (inside the narrow trap) with the following hyperparameters:

- Discretization step size: $\eta = 0.01$

- Total iterations: $N = 100{,}000$

- Repellence strength (SR-ULA only): $\alpha = 3.0$

- History step size (SR-ULA only): $\gamma_n = 0.1 \cdot (n+2)^{-0.6}$

The history variable $\theta_n \in \mathbb{R}^2$ is initialized at $\theta_0 = (0, 0)$ and updated according to

$$\theta_{n+1} = \theta_n + \gamma_{n+1}\big(s(X_{n+1}) - \theta_n\big), \tag{20}$$

where $s(x) = \nabla_x \log \pi(x)$ is the score function.

# B. Algorithmic Implementation

In this section, we provide detailed algorithmic descriptions for incorporating the score-tilted surrogate family $\pi_\theta$ (Eq. (2)) into widely used MCMC algorithms. For each algorithm, we explicitly state the modifications required to implement the Score-Repellent Monte Carlo (SRMC) framework.

Recall the core components of SRMC:

- **Target distribution:** $\pi(x) \propto \exp(-U(x))$ with score $s(x) = -\nabla_x U(x)$.

- **Score-tilted surrogate:**

$$\pi_\theta(x) = \pi(x) \exp\bigl(-\alpha\theta^\top s(x)\bigr), \quad \theta \in \mathbb{R}^d.$$

- **Surrogate score:**

$$\tilde{s}_\theta(x) = \nabla_x \log \pi_\theta(x) = -\nabla_x U(x) + \alpha\nabla_x^2 U(x) \cdot \theta.$$

- **History update:**

$$\theta_{n+1} = \theta_n + \gamma_{n+1}\bigl(s(X_{n+1}) - \theta_n\bigr).$$

## B.1. Score-Repellent Metropolis-Hastings (SR-MH)

Given a proposal kernel $q(x, y)$ independent of $\theta$, the standard MH acceptance probability for the surrogate target $\pi_\theta$ becomes

$$a_\theta(x, y) = \min\left\{1, \frac{\pi(y)\, q(y, x)}{\pi(x)\, q(x, y)} \cdot \exp\bigl(-\alpha\theta^\top(s(y) - s(x))\bigr)\right\}.$$

The normalizing constant $Z_\theta$ cancels out in the ratio. The multiplicative score-repellent factor $\exp\bigl(-\alpha\theta^\top(s(y) - s(x))\bigr)$ encourages transitions that reduce alignment with the accumulated score average $\theta$.

---

**Algorithm 2** Score-Repellent Metropolis-Hastings (SR-MH)

---

**Require:** Target $\pi(x) \propto e^{-U(x)}$; score $s(x) = -\nabla_x U(x)$; proposal $q(x, \cdot)$; strength of repellence $\alpha \geq 0$; step size $\gamma_n$.
1: Initialize $X_0 \in \mathcal{X}$, $\theta_0 \in \mathbb{R}^d$.
2: **for** $n = 0, 1, 2, \ldots, N - 1$ **do**
3:  Sample proposal $Y \sim q(X_n, \cdot)$
4:  Compute base ratio $r_0 = \frac{\pi(Y)\, q(Y, X_n)}{\pi(X_n)\, q(X_n, Y)}$
5:  Compute score-repellent factor $\Delta_\theta = \exp\bigl(-\alpha\theta_n^\top(s(Y) - s(X_n))\bigr)$
6:  Set $X_{n+1} \leftarrow Y$ with probability $\min\{1, r_0 \cdot \Delta_\theta\}$; otherwise $X_{n+1} \leftarrow X_n$
7:  Update $\theta_{n+1} \leftarrow \theta_n + \gamma_{n+1}\left(s(X_{n+1}) - \theta_n\right)$
8: **end for**
9: **Output:** Trajectory $\{X_n\}_{n=0}^N$.

---

## B.2. Score-Repellent Unadjusted Langevin Algorithm (SR-ULA)

The unadjusted Langevin algorithm (ULA) discretizes the overdamped Langevin SDE. When targeting $\pi_\theta$, the continuous-time dynamics are

$$dX_t = \tilde{s}_{\theta_t}(X_t)\, dt + \sqrt{2}\, dB_t, \tag{21}$$

where $\tilde{s}_\theta(x) = s(x) + \alpha\nabla_x^2 U(x) \cdot \theta$ is the surrogate score. The Euler–Maruyama discretization yields SR-ULA.

---

**Algorithm 3** Score-Repellent Unadjusted Langevin Algorithm (SR-ULA)

---

**Require:** Target $\pi(x) \propto e^{-U(x)}$; score $s(x) = -\nabla_x U(x)$; discretization step $\eta > 0$; strength of repellence $\alpha \geq 0$; step size $\gamma_n$.
1: Initialize $X_0 \in \mathbb{R}^d$, $\theta_0 \in \mathbb{R}^d$.
2: **for** $n = 0, 1, 2, \ldots, N - 1$ **do**
3:     Compute Hessian-vector product $H_n = \nabla_x^2 U(X_n) \cdot \theta_n$
4:     Compute surrogate score $\tilde{s}_n = s(X_n) + \alpha H_n$
5:     Sample $\xi \sim \mathcal{N}(0, I_d)$
6:     Update $X_{n+1} \leftarrow X_n + \eta \tilde{s}_n + \sqrt{2\eta}\, \xi$
7:     Update $\theta_{n+1} \leftarrow \theta_n + \gamma_{n+1} (s(X_{n+1}) - \theta_n)$
8: **end for**
9: **Output:** Trajectory $\{X_n\}_{n=0}^{N}$.

---

**Efficient Hessian-vector product computation.** The term $\nabla_x^2 U(x) \cdot \theta$ can be computed efficiently via the following two empirical methods:

1. **Automatic differentiation:** Using forward-mode autodiff or the 'Hessian-free' trick (Pearlmutter, 1994):

$$\nabla_x^2 U(x) \cdot \theta = \nabla_x \big[ \langle \nabla_x U(x), \theta \rangle \big].$$

2. **Finite differences:** For small $\epsilon > 0$,

$$\nabla_x^2 U(x) \cdot \theta \approx \frac{\nabla_x U(x + \epsilon\theta) - \nabla_x U(x)}{\epsilon}.$$

## B.3. Score-Repellent MALA (SR-MALA)

The Metropolis-adjusted Langevin algorithm (MALA) adds a Metropolis correction to ULA. For the surrogate $\pi_\theta$, the proposal distribution is

$$q_\theta(x, y) = \mathcal{N}\big(y \mid x + \epsilon \tilde{s}_\theta(x),\ 2\epsilon I_d \big), \tag{22}$$

and the acceptance probability corrects for discretization error.

---

**Algorithm 4** Score-Repellent Metropolis-Adjusted Langevin Algorithm (SR-MALA)

---

**Require:** Target $\pi(x) \propto e^{-U(x)}$; score $s(x) = -\nabla_x U(x)$; discretization step $\eta > 0$; strength of repellence $\alpha \geq 0$; step size $\gamma_n$.
1: Initialize $X_0 \in \mathbb{R}^d$, $\theta_0 \in \mathbb{R}^d$.
2: **for** $n = 0, 1, 2, \ldots, N - 1$ **do**
3:     Compute surrogate score $\tilde{s}_{\theta_n}(X_n) = s(X_n) + \alpha \nabla_x^2 U(X_n) \cdot \theta_n$
4:     Sample $\xi \sim \mathcal{N}(0, I_d)$ and propose $Y = X_n + \eta \tilde{s}_{\theta_n}(X_n) + \sqrt{2\eta}\, \xi$
5:     Compute surrogate score $\tilde{s}_{\theta_n}(Y) = s(Y) + \alpha \nabla_x^2 U(Y) \cdot \theta_n$
6:     Compute log acceptance ratio:

$$\log r = -U(Y) + U(X_n) - \alpha \theta_n^\top (s(Y) - s(X_n))$$
$$- \frac{1}{4\eta} \big[ \|X_n - Y - \eta \tilde{s}_{\theta_n}(Y)\|^2 - \|Y - X_n - \eta \tilde{s}_{\theta_n}(X_n)\|^2 \big]$$

7:     Set $X_{n+1} \leftarrow Y$ with probability $\min\{1, e^{\log r}\}$; otherwise $X_{n+1} \leftarrow X_n$
8:     Update $\theta_{n+1} \leftarrow \theta_n + \gamma_{n+1} (s(X_{n+1}) - \theta_n)$
9: **end for**
10: **Output:** Trajectory $\{X_n\}_{n=0}^{N}$.

---

## B.4. Score-Repellent Hamiltonian Monte Carlo (SR-HMC)

Hamiltonian Monte Carlo (HMC) (Neal, 2011) augments the state space with an auxiliary momentum variable $v \in \mathbb{R}^d$ and simulates Hamiltonian dynamics to generate distant proposals with high acceptance probability. For the score-tilted surrogate $\pi_\theta(x) \propto \pi(x) \exp(-\alpha \langle \theta, s(x) \rangle)$, the surrogate potential energy becomes $U_\theta(x) = U(x) - \alpha \theta^\top \nabla_x U(x)$, yielding the surrogate Hamiltonian

$$H_\theta(x, v) = U(x) - \alpha \theta^\top \nabla_x U(x) + \frac{1}{2} v^\top M^{-1} v, \tag{23}$$

where $M \succ 0$ is the mass matrix. The corresponding equations of motion show that momentum updates use the surrogate score $\tilde{s}_\theta(x) = s(x) + \alpha \nabla_x^2 U(x) \cdot \theta$ in place of the original score $s(x)$.

We discretize these dynamics using the leapfrog integrator with step size $\eta > 0$ and $L$ integration steps. Each leapfrog step consists of a half-step momentum update, a full-step position update, and another half-step momentum update, all using the surrogate score $\tilde{s}_\theta$. After $L$ steps, we negate the momentum for reversibility and apply a MH correction to account for discretization error. The history variable $\theta_n$ is held constant throughout the leapfrog integration and updated only once per outer iteration based on the accepted sample.

---

**Algorithm 5** Score-Repellent Hamiltonian Monte Carlo (SR-HMC)

---

**Require:** Target $\pi(x) \propto e^{-U(x)}$; score $s(x) = -\nabla_x U(x)$; discretization step $\eta > 0$; leapfrog steps $L$; mass matrix $M$; strength of repellence $\alpha \geq 0$; step size $\gamma_n$.

1: Initialize $X_0 \in \mathbb{R}^d$, $\theta_0 \in \mathbb{R}^d$.
2: **for** $n = 0, 1, 2, \ldots, N-1$ **do**
3:      Sample momentum $v \sim \mathcal{N}(0, M)$
4:      Set $(x, \hat{v}) \leftarrow (X_n, v)$
5:      *// Leapfrog integration with surrogate score*
6:      **for** $\ell = 1, \ldots, L$ **do**
7:          Compute surrogate score $\tilde{s} = s(x) + \alpha \nabla_x^2 U(x) \cdot \theta_n$
8:          $\hat{v} \leftarrow \hat{v} + \frac{\eta}{2} \tilde{s}$             ▷ Half-step momentum
9:          $x \leftarrow x + \eta M^{-1} \hat{v}$             ▷ Full-step position
10:         Compute surrogate score $\tilde{s} = s(x) + \alpha \nabla_x^2 U(x) \cdot \theta_n$
11:        $\hat{v} \leftarrow \hat{v} + \frac{\eta}{2} \tilde{s}$            ▷ Half-step momentum
12:      **end for**
13:      Set proposal $(Y, v') = (x, -\hat{v})$          ▷ Momentum flip for reversibility
14:      Compute $\log r = -H_{\theta_n}(Y, v') + H_{\theta_n}(X_n, v)$
15:      Set $X_{n+1} \leftarrow Y$ with probability $\min\{1, e^{\log r}\}$; otherwise $X_{n+1} \leftarrow X_n$
16:      Update $\theta_{n+1} \leftarrow \theta_n + \gamma_{n+1} (s(X_{n+1}) - \theta_n)$
17: **end for**
18: **Output:** Trajectory $\{X_n\}_{n=0}^N$.

---

## B.5. Score-Repellent Discrete Gradient-Informed Samplers

SRMC naturally extends to discrete configuration state spaces $\mathcal{X} = \{0, 1\}^d$ or $\{1, \ldots, K\}^d$ when a differentiable energy function $U(x)$ is available via continuous relaxation. This setting is common in modern discrete sampling, where gradient information from a relaxed energy landscape guides proposal construction. We describe how three major families of discrete gradient-informed samplers can incorporate the score-tilted surrogate.

### B.5.1. BACKGROUND: GRADIENT-INFORMED DISCRETE MCMC

Consider a target distribution $\pi(x) \propto \exp(-U(x))$ over a discrete space $\mathcal{X}$. Although $x$ takes discrete values, the energy $U : \mathbb{R}^d \to \mathbb{R}$ is often defined as a smooth function that can be evaluated at discrete configurations. The function $-\nabla_x U(x)$ provides local gradient information even at discrete points, treating coordinates as continuous for differentiation purposes.

Modern discrete samplers exploit this gradient information to construct *locally informed proposals* that preferentially propose moves toward lower-energy configurations. These methods achieve significant improvements over uninformed random-walk proposals, particularly in high-dimensional discrete spaces (Zanella, 2020; Grathwohl et al., 2021; Zhang

et al., 2022).

### B.5.2. LOCALLY BALANCED PROPOSALS

The locally balanced framework (Zanella, 2020; Sun et al., 2022) constructs proposals of the form

$$q(x, y) \propto g\big(\pi(y)/\pi(x)\big) \cdot \mathbf{1}[y \in \mathcal{N}(x)], \tag{24}$$

where $N(x)$ denotes the neighborhood of $x$ (e.g., states differing in one coordinate), and $g : \mathbb{R}_{>0} \to \mathbb{R}_{>0}$ is a *balancing function* satisfying $g(t) = t \cdot g(1/t)$. Common choices include:

- **Barker:** $g(t) = t/(1 + t)$

- **Square-root:** $g(t) = \sqrt{t}$

- **Max:** $g(t) = \max\{1, t\}$

For the score-tilted surrogate $\pi_\theta$, the ratio becomes

$$\frac{\pi_\theta(y)}{\pi_\theta(x)} = \frac{\pi(y)}{\pi(x)} \cdot \exp\big(-\alpha\theta^\top(s(y) - s(x))\big). \tag{25}$$

Thus, the SR-locally balanced proposal is

$$q_\theta(x, y) \propto g\left(\frac{\pi(y)}{\pi(x)} \cdot e^{-\alpha\theta^\top(s(y)-s(x))}\right) \cdot \mathbf{1}[y \in \mathcal{N}(x)]. \tag{26}$$

### B.5.3. GIBBS-WITH-GRADIENTS (GWG)

Gibbs-with-Gradients (Grathwohl et al., 2021) uses the score to construct coordinate-wise proposals. For each coordinate $i \in \{1, \dots, d\}$, the method defines proposal probabilities over possible values $k \in \{1, \dots, K\}$ as

$$q_i(x, k) \propto \exp\big(\tau \cdot s_i(x) \cdot (k - x_i)\big), \tag{27}$$

where $s_i(x) = -\partial U(x)/\partial x_i$ is the $i$-th component of the score, and $\tau > 0$ is a temperature parameter. This "locally linear" approximation leverages gradient information to favor moves in directions of decreasing energy.

For the score-tilted surrogate, the surrogate score $\tilde{s}_\theta(x) = s(x) + \alpha\nabla_x^2 U(x) \cdot \theta$ replaces the original score:

$$q_{i,\theta}(x, k) \propto \exp\big(\tau \cdot \tilde{s}_{\theta,i}(x) \cdot (k - x_i)\big). \tag{28}$$

Combined with a MH correction targeting $\pi_\theta$, this yields SR-GWG.

---

**Algorithm 6** Score-Repellent Discrete Gradient-Informed Sampler (SR-DGI)

---

**Require:** Target $\pi(x) \propto e^{-U(x)}$ on $\mathcal{X} = \{1, \dots, K\}^d$; score $s(x) = -\nabla U(x)$; gradient-informed proposal family $q_\theta(x, \cdot)$; strength of repellence $\alpha \geq 0$; step size $\gamma_n$.

1: Initialize $X_0 \in \mathcal{X}, \theta_0 \in \mathbb{R}^d$.
2: **for** $n = 0, 1, 2, \dots, N - 1$ **do**
3:      Compute surrogate score $\tilde{s}_{\theta_n}(X_n) = s(X_n) + \alpha\nabla^2 U(X_n) \cdot \theta_n$         ▷ Optional for methods requiring $\tilde{s}$
4:      Sample proposal $Y \sim q_{\theta_n}(X_n, \cdot)$ using the gradient-informed rule
5:      Compute scores $s(X_n)$ and $s(Y)$         ▷ Already available from proposal
6:      Compute base MH ratio $r_0 = \frac{\pi(Y)\, q_{\theta_n}(Y, X_n)}{\pi(X_n)\, q_{\theta_n}(X_n, Y)}$
7:      Compute score-repellent factor $\Delta_\theta = \exp\big(-\alpha\theta_n^\top(s(Y) - s(X_n))\big)$
8:      Set $X_{n+1} \leftarrow Y$ with probability $\min\{1, r_0 \cdot \Delta_\theta\}$; otherwise $X_{n+1} \leftarrow X_n$
9:      Update $\theta_{n+1} \leftarrow \theta_n + \gamma_{n+1}(s(X_{n+1}) - \theta_n)$
10: **end for**
11: **Output:** Trajectory $\{X_n\}_{n=0}^N$.

---

### B.5.4. DISCRETE LANGEVIN PROPOSALS (DLP)

The discrete Langevin proposal (Zhang et al., 2022) approximates continuous Langevin dynamics on discrete spaces. The proposal distribution takes the form

$$q(x, y) \propto \exp\left(-\frac{\|y - x - \eta s(x)\|^2}{4\eta}\right) \cdot \mathbf{1}[y \in \mathcal{X}], \tag{29}$$

which centers a Gaussian-like kernel at the "Langevin target" $x + \eta s(x)$, then restricts to valid discrete states. For the surrogate target, we simply replace $s(x)$ with $\tilde{s}_\theta(x)$:

$$q_\theta(x, y) \propto \exp\left(-\frac{\|y - x - \eta \tilde{s}_\theta(x)\|^2}{4\eta}\right) \cdot \mathbf{1}[y \in \mathcal{X}]. \tag{30}$$

### B.5.5. UNIFIED ALGORITHM FOR SR-DISCRETE SAMPLERS

All the above discrete gradient-informed methods share a common structure: they use score evaluations to construct proposals and apply a MH correction. Algorithm 6 presents a unified template.

**Computational overhead.**   A key advantage of SRMC for discrete gradient-informed samplers is minimal computational overhead. Since these methods already evaluate the score $s(x)$ at both the current state $X_n$ and the proposed state $Y$ to construct proposals, the SRMC wrapper requires only:

1. One inner product $\theta_n^\top(s(Y) - s(X_n))$ for the acceptance correction.

2. One vector addition for the history update $\theta_{n+1} \leftarrow \theta_n + \gamma_{n+1}(s(X_{n+1}) - \theta_n)$.

No additional score evaluations are needed. The optional Hessian-vector product $\nabla_x^2 U(X_n) \cdot \theta_n$ is required only if the proposal itself depends on $\tilde{s}_\theta$ (e.g., in SR-GWG or SR-DLP); for pure locally balanced proposals that use only the ratio $\pi_\theta(y)/\pi_\theta(x)$, even this computation is unnecessary since only the inner product is involved.

# C. Technical Details and Proofs for Section 3

This section contains all technical arguments and the necessary proof steps for Theorem 3.3 and Proposition 3.4. For the almost sure convergence and CLT results, readers are advised to begin with Appendix C.8 for guidance, and then consult the individual subsections that provide the proofs and discussions for each required assumption.

## C.1. Verification of Assumption 1 for Common Distributions

We verify that Assumption 1 holds for several commonly used target distributions in Bayesian inference and machine learning. Most results concerning the Lipschitz property of the score function and the super-linear growth are scattered in the literature, and we include all relevant details for the sake of completeness. Asymptotic regularity is less seen in the literature, so we present our own derivation of the necessary steps.

### C.1.1. GAUSSIAN DISTRIBUTION

Consider $\pi(x) = \mathcal{N}(\mu, \Sigma)$ with $\Sigma \succ 0$, and potential $U(x) = \frac{1}{2}(x - \mu)^\top \Sigma^{-1}(x - \mu)$.

**Lipschitz score.**    The score is $s(x) = -\nabla_x U(x) = -\Sigma^{-1}(x - \mu)$, which gives

$$\|s(x) - s(x')\| = \|\Sigma^{-1}(x - x')\| \leq \|\Sigma^{-1}\| \cdot \|x - x'\|.$$

Thus, $s$ is $L$-Lipschitz with $L = \lambda_{\max}(\Sigma^{-1})$.

**Super-linear tail growth.**    We have

$$U(x) = \frac{1}{2}(x - \mu)^\top \Sigma^{-1}(x - \mu) \geq \frac{\lambda_{\min}(\Sigma^{-1})}{2}\|x - \mu\|^2.$$

Thus, condition (i) holds with $p = 2$.

**Asymptotic regularity.**    The Hessian is constant: $\nabla^2 U(x) = \Sigma^{-1}$ for all $x$. Thus,

$$r^{-(p-2)}\nabla_x^2 U(r\hat{x}) = r^0 \cdot \Sigma^{-1} = \Sigma^{-1} \succ 0,$$

so $M(\hat{x}) = \Sigma^{-1}$ independent of $\hat{x}$, and condition (ii) is satisfied.

### C.1.2. GAUSSIAN MIXTURE MODEL

Consider $\pi(x) = \sum_{k=1}^K w_k \mathcal{N}(x; \mu_k, \Sigma_k)$ with $w_k > 0$, $\sum_k w_k = 1$, and $\Sigma_k \succ 0$ for each $k$.

**Lipschitz score.**    The score is

$$s(x) = -\nabla_x U(x) = \frac{\sum_{k=1}^K w_k \phi_k(x) \Sigma_k^{-1}(x - \mu_k)}{\sum_{k=1}^K w_k \phi_k(x)},$$

where $\phi_k(x) = \mathcal{N}(x; \mu_k, \Sigma_k)$. A global Lipschitz constant $L$ exists and can be computed from $\max_k \|\Sigma_k^{-1}\|$ and the geometry of the means $\{\mu_k\}$. See Liang et al. (2025, Section 3) for more details.

**Super-linear tail growth.**    As $\|x\| \to \infty$, the potential satisfies

$$U(x) = -\log \pi(x) \geq -\log\left(\max_k w_k \phi_k(x)\right) \geq \frac{\lambda_{\min}\omega_{\min}}{2}\|x - \mu_{k^*}\|^2 - C$$

for some constant $C$, where $\lambda_{\min} = \min_k \lambda_{\min}(\Sigma_k^{-1})$, and $\omega_{\min} = \min_k \omega_k$. This yields $U(x) \geq c\|x\|^2$ for $\|x\| \geq R$ with sufficiently large $R$ value, so condition (i) holds with $p = 2$.

**Asymptotic regularity.** As $\|x\| \to \infty$ along the direction of $\hat{x} \in \mathbb{S}^{d-1}$, the mixture $\pi(x) = \sum_{k=1}^{K} w_k \mathcal{N}(x; \mu_k, \Sigma_k)$ becomes dominated by a single component. To see this, write $x = r\hat{x}$ and note that each Gaussian component satisfies

$$w_k \mathcal{N}(r\hat{x}; \mu_k, \Sigma_k) \propto \exp\left(-\frac{r^2}{2} \hat{x}^\top \Sigma_k^{-1} \hat{x} + O(r)\right).$$

For large $r$, the component with the smallest coefficient $\hat{x}^\top \Sigma_k^{-1} \hat{x}$ dominates exponentially. Defining

$$k^*(\hat{x}) = \underset{k \in \{1,\ldots,K\}}{\arg\min} \ \hat{x}^\top \Sigma_k^{-1} \hat{x},$$

we have $\nabla_x^2 U(r\hat{x}) \to \Sigma_{k^*(\hat{x})}^{-1}$ as $r \to \infty$. Thus $M(\hat{x}) = \Sigma_{k^*(\hat{x})}^{-1} \succ 0$, verifying condition (ii) with $p = 2$.

### C.1.3. BAYESIAN LOGISTIC REGRESSION WITH GAUSSIAN PRIOR

Consider the posterior for logistic regression with data $\{(a_i, y_i)\}_{i=1}^{n}$, $a_i \in \mathbb{R}^d$, $y_i \in \{0, 1\}$, and Gaussian prior $\mathcal{N}(0, \tau^2 I)$:

$$\pi(x) \propto \exp\left(-\frac{\|x\|^2}{2\tau^2} - \sum_{i=1}^{n} \ell_i(x)\right),$$

where $\ell_i(x) = \log(1 + e^{a_i^\top x}) - y_i a_i^\top x$ is the logistic loss. The potential is

$$U(x) = \frac{\|x\|^2}{2\tau^2} + \sum_{i=1}^{n} \ell_i(x).$$

**Lipschitz score.** The score is

$$s(x) = -\nabla_x U(x) = -\frac{x}{\tau^2} - \sum_{i=1}^{n} (\sigma(a_i^\top x) - y_i) a_i,$$

where $\sigma(z) \triangleq 1/(1 + e^{-z})$ is the sigmoid function. The Hessian is

$$\nabla_x^2 U(x) = \frac{1}{\tau^2} I + \sum_{i=1}^{n} \sigma(a_i^\top x)(1 - \sigma(a_i^\top x)) a_i a_i^\top.$$

Since $\sigma(z)(1 - \sigma(z)) \leq 1/4$ for all $z$, we have

$$\frac{1}{\tau^2} I \preceq \nabla_x^2 U(x) \preceq \frac{1}{\tau^2} I + \frac{1}{4} \sum_{i=1}^{n} a_i a_i^\top,$$

so $\nabla_x^2 U(x)$ is uniformly bounded. By the mean value theorem, $s(x)$ is $L$-Lipschitz with

$$L = \frac{1}{\tau^2} + \frac{1}{4} \left\| \sum_{i=1}^{n} a_i a_i^\top \right\|.$$

**Super-linear tail growth.** Since each $\ell_i(x) \geq 0$ (logistic loss is non-negative), we have

$$U(x) \geq \frac{\|x\|^2}{2\tau^2}.$$

Thus condition (i) holds with $p = 2$, and any $R > 0$.

**Asymptotic regularity.** As $\|x\| \to \infty$ along the direction of $\hat{x}$, for each $i$:

$$\sigma(a_i^\top(r\hat{x}))(1 - \sigma(a_i^\top(r\hat{x}))) \to 0 \quad \text{as } r \to \infty$$

because $\sigma(z)(1 - \sigma(z)) \to 0$ as $|z| \to \infty$. Therefore,

$$\nabla_x^2 U(r\hat{x}) \to \frac{1}{\tau^2} I \quad \text{as } r \to \infty,$$

for each $\hat{x} \in \mathbb{S}^{d-1}$ (the convergence rate depends on $\min_i |a_i^\top \hat{x}|$, but the limit is the same). Thus,

$$M(\hat{x}) = \frac{1}{\tau^2} I \succ 0,$$

independent of $\hat{x}$, and condition (ii) is satisfied with $p = 2$.

### C.1.4. STRONGLY LOG-CONCAVE DISTRIBUTIONS

A distribution $\pi \propto e^{-U}$ is *strongly log-concave* if the potential $U : \mathbb{R}^d \to \mathbb{R}$ is $\lambda$-strongly convex for some $\lambda > 0$, i.e.,

$$\nabla_x^2 U(x) \succeq \lambda I \quad \text{for all } x \in \mathbb{R}^d.$$

The Hessian is further assumed to be bounded above: $\nabla_x^2 U(x) \preceq LI$ for some $L < \infty$. This class includes Gaussian as a special case and covers many distributions used in Bayesian inference (Saumard & Wellner, 2014).

**Lipschitz score.** Since $\nabla^2 U(x) \preceq LI$, for any $x, x' \in \mathbb{R}^d$:

$$\|s(x) - s(x')\| = \|\nabla_x U(x) - \nabla_x U(x')\| \le L\|x - x'\|.$$

by the mean value theorem. Thus score function $s$ is $L$-Lipschitz.

**Super-linear tail growth.** Let $x^* = \arg\min_x U(x)$ be the unique minimizer (which exists by strong convexity). By $\lambda$-strong convexity:

$$U(x) \ge U(x^*) + \frac{\lambda}{2}\|x - x^*\|^2.$$

For $\|x\| \ge R$ with $R > 2\|x^*\|$, we have $\|x - x^*\| \ge \|x\| - \|x^*\| \ge \frac{1}{2}\|x\|$, so

$$U(x) \ge U(x^*) + \frac{\lambda}{8}\|x\|^2 \ge \frac{\lambda}{8}\|x\|^2$$

for $\|x\|$ sufficiently large (absorbing $U(x^*)$ into the threshold $R$). Condition (i) holds with $p = 2$ and $c = \frac{\lambda}{8}$.

**Asymptotic regularity.** To verify condition (ii), we require an additional assumption on the behavior of $\nabla_x^2 U(x)$ as $\|x\| \to \infty$.

- **Case 1: Asymptotically quadratic.** If $\nabla_x^2 U(x) \to H_\infty$ as $\|x\| \to \infty$ for some $H_\infty \succ 0$, then

$$r^{-(p-2)}\nabla_x^2 U(r\hat{x}) = \nabla_x^2 U(r\hat{x}) \to H_\infty$$

  for all $\hat{x} \in \mathbb{S}^{d-1}$, and $M(\hat{x}) = H_\infty$ is constant.

- **Case 2: Direction-dependent limit.** More generally, if for each $\hat{x} \in \mathbb{S}^{d-1}$,

$$\nabla_x^2 U(r\hat{x}) \to M(\hat{x}) \succ 0 \quad \text{as } r \to \infty,$$

  then condition (ii) is satisfied.

*Examples of strongly log-concave distributions satisfying Assumption 1:*

*Table 1.* Summary of Assumption 1 verification for common distributions.

| Distribution | Tail exponent $p$ | Lipschitz const. $L$ | $M(\hat{x})$ |
|---|---|---|---|
| Gaussian $\mathcal{N}(\mu, \Sigma)$ | 2 | $\|\Sigma^{-1}\|$ | $\Sigma^{-1}$ |
| Gaussian mixture | 2 | $\max_k \|\Sigma_k^{-1}\| + O(1)$ | $\Sigma_{k^*(\hat{x})}^{-1}$ |
| Bayesian logistic reg. | 2 | $\tau^{-2} + \frac{1}{4}\|\sum_{i=1}^n a_i a_i^\top\|$ | $\tau^{-2}I$ |
| Strongly log-concave | 2 | $L$ | $\lambda I$ |

- Gaussian $\mathcal{N}(\mu, \Sigma)$: $\nabla_x^2 U(x) = \Sigma^{-1}$ is constant, so $M(\hat{x}) = \Sigma^{-1}$.

- Bayesian linear regression posterior: With Gaussian prior $\mathcal{N}(0, \tau^2 I)$ and Gaussian likelihood, the posterior is $\mathcal{N}(\mu_{\text{post}}, \Sigma_{\text{post}})$, which falls under the Gaussian case.

- Regularized potentials: $U(x) = V(x) + \frac{\lambda}{2}\|x\|^2$ where $V$ is convex with $\nabla_x^2 V(x) \to 0$ as $\|x\| \to \infty$ (e.g., $V$ has bounded support or logarithmic growth). Then $\nabla_x^2 U(x) \to \lambda I$, so $M(\hat{x}) = \lambda I$.

- Huber-type potentials: $U(x) = \sum_{i=1}^d \rho(x_i) + \frac{\lambda}{2}\|x\|^2$ where $\rho$ is the Huber loss. Since $\rho''(t) \to 0$ as $|t| \to \infty$, we have $\nabla_x^2 U(r\hat{x}) \to \lambda I$.

**Remark.** The key requirement for condition (ii) is that the curvature of $U$ stabilizes at infinity. This is satisfied by potentials that are 'eventually quadratic', i.e., $U(x) \approx \frac{1}{2}x^\top H_\infty x$ for large $\|x\|$. Strongly log-concave distributions with uniformly bounded Hessian ($\lambda I \preceq \nabla^2 U \preceq L I$) automatically satisfy this when the Hessian has a limit along rays.

**Summary.** Table 1 summarizes the verification. All four listed distribution classes satisfy Assumption 1 with quadratic tail growth ($p = 2$). The strongly log-concave class, including Gaussian and Bayesian logistic regression cases, has direction-independent limiting Hessian $M(\hat{x})$, while Gaussian mixtures have a piecewise constant $M(\hat{x})$ depending on which component dominates in each direction. These results confirm that Assumption 1 covers a broad class of targets commonly encountered in machine learning and Bayesian inference.

### C.2. Proof of Lemma 3.2

*Proof.* Fix $\theta \in \mathbb{R}^d$ and $\alpha > 0$. Recall that

$$Z_\theta = \int_{\mathbb{R}^d} \pi(x)\, e^{-\alpha \theta^\top s(x)}\, dx.$$

Now we would like to bound the term $e^{-\alpha \theta^\top s(x)}$. By score Lipschitzness in Assumption 1, for an arbitrary $x_0 \in \mathcal{X}$,

$$\|s(x) - s(x_0)\| \le L\|x - x_0\| \le L\|x\| + L\|x_0\|,$$

which gives $\|s(x)\| \le L\|x\| + L\|x_0\| + \|s(x_0)\|$. We denote by $L_0 \triangleq L\|x_0\| + \|s(x_0)\| < \infty$. Then, by the Cauchy-Schwarz inequality, we have

$$-\alpha \theta^\top s(x) \le \alpha\|\theta\|\|s(x)\| = \alpha\|\theta\|(L\|x\| + L_0). \tag{31}$$

Now rewrite $Z_\theta$ as

$$Z_\theta = \int_{\|x\|<R} \pi(x)\, e^{-\alpha\theta^\top s(x)}\, dx + \int_{\|x\|\ge R} \pi(x)\, e^{-\alpha\theta^\top s(x)}\, dx.$$

Eq. (31) and the super-linear tail growth condition in Assumption 1 result in

$$\int_{\|x\|<R} \pi(x)\, e^{-\alpha\theta^\top s(x)}\, dx \le e^{\alpha\|\theta\|(LR+L_0)} \cdot \left[\int_{\|x\|<R} \pi(x)\, dx\right] \le e^{\alpha\|\theta\|(LR+L_0)} < \infty,$$

and

$$\int_{\|x\|\ge R} \pi(x)\, e^{-\alpha\theta^\top s(x)}\, dx \le \frac{1}{Z_0}\int_{\|x\|\ge R} e^{-c\|x\|^p + \alpha\|\theta\|(L\|x\|+L_0)} dx,$$

where the inequality comes from using $\pi(x) = \frac{1}{Z_0} e^{-U(x)}$, (31) and super-linear tail growth in Assumption 1. For sufficiently large $R$, we have $-c\|x\|^p + \alpha L\|\theta\| \cdot \|x\| \leq -\frac{c}{2}\|x\|^p$. Therefore,

$$\int_{\|x\| \geq R} \pi(x)\, e^{-\alpha \theta^\top s(x)}\, dx \leq \frac{1}{Z_0} \int_{\|x\| \geq R} e^{-\frac{c}{2}\|x\|^p}\, dx < \infty,$$

since $p > 1$ ensures the integrability. This completes the proof. $\qquad\square$

### C.3. Discussion on Kernel Lipschitzness in Assumption 2

For completeness, we begin by introducing the required notation and then present the (DV3) condition and the kernel Lipschitz property as in Borkar et al. (2025). Let $G : \mathcal{X} \to [1, \infty)$ be measurable. For any measurable $f : \mathcal{X} \to \mathbb{R}$, define

$$\|f\|_G \triangleq \sup_{x \in \mathcal{X}} \frac{|f(x)|}{G(x)}, \qquad L_\infty^G \triangleq \{f : \mathcal{X} \to \mathbb{R} : \|f\|_G < \infty\}.$$

For measurable $G, M : \mathcal{X} \to [1, \infty)$ and a linear operator (kernel) $K : L_\infty^G \to L_\infty^M$, define

$$\|K\|_{G,M} \triangleq \sup\{\|Kf\|_M : f \in L_\infty^G,\ \|f\|_G \leq 1\}, \qquad \|K\|_G \triangleq \|K\|_{G,G}.$$

**Assumption 3** (DV3). There exist measurable functions $V : \mathcal{X} \to \mathbb{R}_+$, $W : \mathcal{X} \to [1, \infty)$, a function $g : \mathcal{X} \to [0, 1]$, and a constant $b > 0$ such that, for all $x \in \mathcal{X}$,

$$\mathbb{E}\big[\exp\big(V(\Phi_{k+1})\big) \,\big|\, \Phi_k = x\big] \leq \exp\left(V(x) - W(x) + b\, g(x)\right). \tag{32}$$

Moreover, the kernel $P_\theta$ is *aperiodic* for all $\theta$ by satisfying the following minorization condition: there exists a probability measure $\nu$ on the Borel set $\mathcal{B}(\mathcal{X})$ such that

$$R_\theta(x, A) \geq g(x)\nu(A) \quad \text{for } A \in \mathcal{B}(\mathcal{X}) \text{ and all } x \in \mathcal{X}, \theta \in \mathbb{R}^d, \tag{33}$$

where the resolvent $R_\theta \triangleq \sum_{n=0}^\infty 2^{-n-1} P_\theta^n$.

**Assumption 4** (Kernel Lipschitzness). There exist a constant $b_d < \infty$ and a measurable function $G : \mathcal{X} \to [1, \infty)$ such that, for all $\theta, \theta' \in \mathbb{R}^d$,

$$\|P_\theta - P_{\theta'}\|_G \leq \frac{b_d}{1 + \|\theta\| + \|\theta'\|} \|\theta - \theta'\|, \quad G \in \{1 + V, 1 + V^2\}. \tag{34}$$

*Remark* 1. On finite state spaces, Assumptions 3 and 4 are essentially automatic for any geometrically ergodic Markov chain. To see this, consider a finite state space $\mathcal{X}$ with $|\mathcal{X}| = N < \infty$. The key observation is that every function $V : \mathcal{X} \to \mathbb{R}_+$ and $W : \mathcal{X} \to [1, \infty)$ is automatically bounded since each takes only finitely many values, i.e., checking $N$ scalar inequalities, one per state, and can always be satisfied by choosing $V, W$, and $g$ appropriately. For example, we can choose $V(x) \equiv 0$, $W(x) \equiv 1$, and $g(x) \equiv 1$ for all $x \in \mathcal{X}$, thereby reducing the DV3 condition to

$$\mathbb{E}\left[\exp(V(\Phi_{k+1})) \,|\, \Phi_k = x\right] = 1 \leq \exp(V(x) - W(x) + bg(x)) = \exp(-1 + b),$$

which can be trivially satisfied by picking large enough $b$ value. Moreover, every non-empty subset of a finite state space is automatically small in the sense of minorization conditions, since the transition kernel is an $N \times N$ stochastic matrix with strictly positive entries after finitely many steps under aperiodicity. Likewise, the kernel Lipschitz condition reduces to bounding entries of the transition kernel. This stands in sharp contrast to general state spaces $\mathcal{X} = \mathbb{R}^d$, where (DV3) can fail even for geometrically ergodic chains, as demonstrated by the M/M/1 queue counterexample in Borkar et al. (2025), which is geometrically ergodic yet violates (DV3) for any unbounded $W$, leading to divergent second moments in the associated SA recursion. Therefore, on general state spaces $\mathcal{X} = \mathbb{R}^d$, these conditions require explicit verification, which we provide below.

**Proposition 2** (Kernel Lipschitzness for SR-MH). *Let $P_\theta$ be the Metropolis-Hastings (MH) kernel targeting $\pi_\theta$ with a proposal kernel $q$ independent of $\theta$. Assume that the family $\{P_\theta\}$ satisfies the uniform drift condition in Assumption 3, and that for each compact set $\Theta \subset \mathbb{R}^d$ and each $\beta \in \{1, 2\}$ there exists a constant $C_{\beta,\Theta} < \infty$ such that*

$$\int_{\mathcal{X}} \|s(y) - s(x)\|\big(1 + V(y)^\beta\big)\, q(x, y)dy \leq C_{\beta,\Theta}\big(1 + V(x)^\beta\big). \tag{35}$$

*Then, for each compact $\Theta \subset \mathbb{R}^d$ and each $\beta \in \{1, 2\}$,*

$$\sup_{\theta, \theta' \in \Theta} \|\theta - \theta'\|^{-1} \|P_\theta - P_{\theta'}\|_{1+V^\beta} \leq \alpha \, C_{\beta, \Theta} < \infty.$$

*Proof.* Write the MH kernel in the usual form

$$P_\theta(x, y) = q(x, y) \, a_\theta(x, y) + \delta_x(y) \Big( 1 - \int q(x, z) \, a_\theta(x, z) dz \Big),$$

where $a_\theta(x, y) = \min\{1, r_\theta(x, y)\}$ and $r_\theta(x, y) = \frac{\pi_\theta(y) q(y, x)}{\pi_\theta(x) q(x, y)}$. Since $q$ does not depend on $\theta$, we can express, for any measurable $\varphi$,

$$P_\theta \varphi(x) - P_{\theta'} \varphi(x) = \int q(x, y) \big( a_\theta(x, y) - a_{\theta'}(x, y) \big) \big( \varphi(y) - \varphi(x) \big) dy.$$

Fix a weight $G(x) \triangleq 1 + V(x)^\beta$ and take $\varphi$ such that $|\varphi| \leq G$. Then

$$|P_\theta \varphi(x) - P_{\theta'} \varphi(x)| \leq \int q(x, y) \, |a_\theta(x, y) - a_{\theta'}(x, y)| \, \big( G(y) + G(x) \big) dy.$$

We now bound the acceptance difference. Let $u_\theta(x, y) \triangleq \log r_\theta(x, y)$ and define $a(u) \triangleq \min\{1, e^u\}$. Because the derivative of $a(u)$ is $e^u$ for $u < 0$ and $0$ for $u > 0$, therefore $a$ is 1-Lipschitz on $\mathbb{R}$. Hence,

$$|a_\theta(x, y) - a_{\theta'}(x, y)| = |a(u_\theta(x, y)) - a(u_{\theta'}(x, y))| \leq |u_\theta(x, y) - u_{\theta'}(x, y)|.$$

For the surrogate $\pi_\theta$, the MH ratio satisfies

$$\frac{\pi_\theta(y)}{\pi_\theta(x)} = \frac{\pi(y)}{\pi(x)} \exp\big( -\alpha \theta^\top (s(y) - s(x)) \big),$$

so that $u_\theta(x, y) = u_0(x, y) - \alpha \theta^\top (s(y) - s(x))$ and

$$|u_\theta(x, y) - u_{\theta'}(x, y)| = \alpha |(\theta - \theta')^\top (s(y) - s(x))| \leq \alpha \|\theta - \theta'\| \, \|s(y) - s(x)\|.$$

Combining the above and using (35),

$$\begin{aligned}
\frac{|P_\theta \varphi(x) - P_{\theta'} \varphi(x)|}{G(x) \|\theta - \theta'\|} &\leq \alpha \frac{1}{\|\theta - \theta'\|} \frac{\|\theta - \theta'\|}{G(x)} \int q(x, y) \, \|s(y) - s(x)\| \, \big( G(y) + G(x) \big) dy \\
&\leq \alpha \Big[ \frac{1}{G(x)} \int q(x, y) \, \|s(y) - s(x)\| G(y) dy + \int q(x, y) \, \|s(y) - s(x)\| dy \Big] \\
&\leq \alpha \, C_{\beta, \Theta},
\end{aligned}$$

where the last step uses (35) and the fact that $G \geq 1$. Taking the supremum over $x$ and then over all $|\varphi| \leq G$ yields the claimed operator-norm bound. $\square$

**Proposition 3** (Kernel Lipschitzness for SR-MALA). *Let $P_\theta$ be the Metropolis-adjusted Langevin algorithm (MALA) targeting $\pi_\theta$, with proposal density $q_\theta(x, \cdot) = \mathcal{N}(x + \eta s_\theta(x), \, 2\eta I)$ for a fixed $\eta > 0$ and surrogate score $s_\theta(x)$ defined in (5). Suppose target distribution $\pi$ follows Assumption 1. Assume further that the MALA family satisfies Assumption 3 with common $(V, W)$ and that the proposal moments are controlled so that (35) holds with $s$ replaced by $s_\theta$ uniformly over $\theta$ in bounded sets. Then, for each compact $\Theta \subset \mathbb{R}^d$ and each $\beta \in \{1, 2\}$, there exists $C_{\beta, \Theta} < \infty$ such that*

$$\sup_{\theta, \theta' \in \Theta} \|\theta - \theta'\|^{-1} \|P_\theta - P_{\theta'}\|_{1+V^\beta} \leq C_{\beta, \Theta}.$$

*Proof.* We highlight the two places where $\theta$ affects MALA: the proposal $q_\theta$ and the acceptance probability $a_\theta(x, y)$.

**Step 1: $\theta$-Lipschitzness of the proposal.** Recall the surrogate score

$$s_\theta(x) = \nabla_x \log \pi_\theta(x) = s(x) - \alpha \nabla_x s(x)\theta.$$

By Assumption 1, the score function $s(x)$ being Lipschitz is equivalent to $\|\nabla_x s(x)\| \leq L$. Hence, for any $\theta, \theta' \in \Theta$, $\|s_\theta(x) - s_{\theta'}(x)\| \leq \alpha\|\nabla_x s(x)\| \|\theta - \theta'\| \leq \alpha L\|\theta - \theta'\|$.

Let $m_\theta(x) \triangleq x + \eta s_\theta(x)$, the proposal means differ by $m_\theta(x) - m_{\theta'}(x) = \eta\big(s_\theta(x) - s_{\theta'}(x)\big)$, so the KL divergence between $\mathcal{N}(m_\theta(x), 2\eta I)$ and $\mathcal{N}(m_{\theta'}(x), 2\eta I)$ is $\mathrm{KL} = \frac{1}{2}\|(2\eta I)^{-1/2}(m_\theta(x) - m_{\theta'}(x))\|^2 = \frac{\eta}{4}\|s_\theta(x) - s_{\theta'}(x)\|^2$. Let $\|\cdot\|_{\mathrm{TV}}$ be the total variation. By Pinsker's inequality (Csiszár & Körner, 2011, p. 44),

$$\|q_\theta(x, \cdot) - q_{\theta'}(x, \cdot)\|_{\mathrm{TV}} \leq \sqrt{\tfrac{1}{2}\mathrm{KL}} \leq \sqrt{\tfrac{\eta}{8}}\, \alpha L\, \|\theta - \theta'\|. \tag{36}$$

**Step 2: Lipschitzness of the acceptance probability.** Same as in the MH case, we let $a_\theta(x, y) = \min\{1, r_\theta(x, y)\}$ with $r_\theta(x, y) = \frac{\pi_\theta(y)q_\theta(y,x)}{\pi_\theta(x)q_\theta(x,y)}$ and define $u_\theta(x, y) = \log r_\theta(x, y)$. Moreover, $u \mapsto \min\{1, e^u\}$ is 1-Lipschitz, thus $|a_\theta(x, y) - a_{\theta'}(x, y)| \leq |u_\theta(x, y) - u_{\theta'}(x, y)|$. A direct expansion gives

$$u_\theta(x, y) = \underbrace{\log \pi(y) - \log \pi(x)}_{\text{independent of }\theta} - \alpha\theta^\top(s(y) - s(x)) + \log q_\theta(y, x) - \log q_\theta(x, y),$$

where the normalizing constant $Z(\theta)$ cancels out in the ratio $\pi_\theta(y)/\pi_\theta(x)$. The first $\theta$-dependent term contributes at most $\alpha\|\theta - \theta'\|\|s(y) - s(x)\|$. For the proposal-density terms, since $\log q_\theta(x, y) = \mathrm{const} - \|y - x - \eta s_\theta(x)\|^2/(4\eta)$, a mean-value argument yields the pointwise bound

$$|\log q_\theta(x, y) - \log q_{\theta'}(x, y)| \leq \tfrac{1}{4}\|s_\theta(x) - s_{\theta'}(x)\|\Big(\|y - x - \eta s_\theta(x)\| + \|y - x - \eta s_{\theta'}(x)\|\Big),$$

and similarly for $|\log q_\theta(y, x) - \log q_{\theta'}(y, x)|$. Combining with $\|s_\theta(\cdot) - s_{\theta'}(\cdot)\| \leq \alpha L\|\theta - \theta'\|$ gives

$$|u_\theta(x, y) - u_{\theta'}(x, y)| \leq \|\theta - \theta'\|\, \mathsf{L}_\Theta(x, y),$$

for an explicit envelope $\mathsf{L}_\Theta(x, y)$ that is linear in $\|s(y) - s(x)\|$, $\|s_\theta(x)\|$, $\|s_\theta(y)\|$, and $\|y - x\|$.

**Step 3: operator-norm bound.** Write the MALA kernel as $P_\theta(x, dy) = q_\theta(x, dy)a_\theta(x, y) + \delta_x(dy)(1 - \int q_\theta a_\theta)$. Proceeding as in the proof of Proposition 2, for $|\varphi| \leq G$ with $G = 1 + V^\beta$,

$$|P_\theta \varphi(x) - P_{\theta'} \varphi(x)| \leq \int \big|q_\theta(x, dy)a_\theta(x, y) - q_{\theta'}(x, dy)a_{\theta'}(x, y)\big|\, (G(y) + G(x)).$$

Use the triangle inequality to split the difference into a proposal part and an acceptance part:

$$\int |(q_\theta - q_{\theta'})(x, dy)|\, (G(y) + G(x)) \quad + \quad \int q_{\theta'}(x, dy)\, |a_\theta(x, y) - a_{\theta'}(x, y)|\, (G(y) + G(x)).$$

The proposal term is controlled by the total variation bound (36) from Step 1 together with the proposal moment condition mentioned in Proposition 3. The acceptance term is controlled using Step 2 and the assumed integrability bound on $\mathsf{L}_\Theta(x, y)$ under $q_{\theta'}(x, dy)$ with weight $G(y)$. Taking $\sup_x$ and $\sup_{|\varphi| \leq G}$ yields the claim. $\qquad\square$

### C.4. Stability Analysis of the Associated ODE and its Equilibrium

The ODE associated with the SA recursion (8) is

$$\dot{\vartheta}_t = h(\vartheta_t). \tag{37}$$

We then characterize the ODE equilibrium.

**Proposition 4** (Unique globally asymptotically stable equilibrium). *Under Assumption 1, the ODE (37) has a unique equilibrium*

$$\vartheta^\star = (0, \mu).$$

*Moreover, $\vartheta^\star$ is globally asymptotically stable for the coupled system (37).*

*Proof.* We first analyze the ODE $\dot{\theta} = \mathcal{S}(\theta) - \theta$ for the $\theta$ iteration.

**Step 1: $\mathcal{S}(\theta)$ is the gradient of the convex function $\psi$.** Define $\psi(\theta) \triangleq \log Z_\theta$. The function $\psi$ is convex because it serves as the log moment generating function of the random variable $-\alpha s(X)$ when $X \sim \pi$. More concretely, its Hessian is given by $\nabla_\theta^2 \psi(\theta) = \alpha^2 \mathrm{Cov}_{\pi_\theta}(s, s) \succeq 0$.

Under Assumption 1, $\psi$ is finite on $\mathcal{X}$ and differentiable with

$$\nabla_\theta \psi(\theta) = \frac{\nabla_\theta Z_\theta}{Z_\theta} = \frac{\int \pi(x)\big(-\alpha s(x)\big) e^{-\alpha \theta^\top s(x)}\, dx}{Z_\theta} = -\alpha\, \mathcal{S}(\theta).$$

Therefore, $\mathcal{S}(\theta) = -(1/\alpha)\nabla_\theta \psi(\theta)$.

**Step 2: the $\theta$-ODE is a gradient flow of a strongly convex function.** Consider

$$l(\theta) := \frac{1}{2}\|\theta\|^2 + \frac{1}{\alpha}\psi(\theta).$$

Then

$$\nabla_\theta l(\theta) = \theta + \frac{1}{\alpha}\nabla_\theta \psi(\theta) = \theta - \mathcal{S}(\theta),$$

so the $\theta$-ODE can be written as

$$\dot\theta = \mathcal{S}(\theta) - \theta = -\nabla_\theta l(\theta).$$

Since the function $\psi$ is convex, the Hessian of function $l$ can be written as

$$\nabla_\theta^2 l(\theta) = I_d + \frac{1}{\alpha}\nabla_\theta^2 \psi(\theta) = I_d + \alpha\, \mathrm{Cov}_{\pi_\theta}(s, s) \succeq I_d,$$

which implies that $l$ is 1-strongly convex on $\mathcal{X}$.

**Step 3: identify the unique equilibrium.** An equilibrium satisfies $\nabla_\theta l(\theta) = 0$, i.e. $\theta = \mathcal{S}(\theta)$. Since $\mathcal{S}(0) = 0$ by Stein's identity, we have $\nabla_\theta l(0) = 0$. Strong convexity implies $l$ has a unique minimizer; therefore $\theta^\star = 0$ is the unique equilibrium.

**Step 4: global exponential stability of $\theta^\star = 0$.** For a 1-strongly convex $l$, the gradient flow $\dot\theta = -\nabla_\theta l(\theta)$ is globally exponentially stable. We use the inequality

$$\langle \nabla_\theta l(\theta) - \nabla_\theta l(\theta^\star),\, \theta - \theta^\star \rangle \geq \|\theta - \theta^\star\|^2,$$

for 1-strongly convex function $l$. With $\theta^\star = 0$ and $\nabla_\theta l(\theta^\star) = 0$, we obtain

$$\frac{d}{dt}\frac{1}{2}\|\theta_t\|^2 = \langle \theta_t, \dot\theta_t \rangle = -\langle \theta_t, \nabla_\theta l(\theta_t) \rangle \leq -\|\theta_t\|^2,$$

hence $\|\theta_t\| \leq e^{-t}\|\theta_0\|$.

**Step 5: stability of the coupled ODE.** The $\mu$-dynamics satisfy $\dot\mu_t = \mathcal{F}(\theta_t) - \mu_t$, which is a stable linear ODE with time-varying input $\mathcal{F}(\theta_t)$. Its solution is explicit:

$$\mu_t = e^{-t}\mu_0 + \int_0^t e^{-(t-u)}\mathcal{F}(\theta_u)\, du.$$

Since $\theta_t \to 0$ exponentially and $\mathcal{F}$ is continuous at 0, we have $\mathcal{F}(\theta_t) \to \mathcal{F}(0) = \mu$. Taking $t \to \infty$ gives $\mu_t \to \mu$. Therefore, $\vartheta^\star = (0, \mu)$ is globally asymptotically stable for (37). $\qquad\square$

## C.5. Verification of the ODE@$\infty$ Condition

We verify condition (A3) of Borkar et al. (2025) for the joint iterate $\vartheta_n = (\theta_n, \mu_n) \in \mathbb{R}^{d+m}$, where $\theta_n \in \mathbb{R}^d$ is the score history (running average of scores over the trajectory) and $\mu_n \in \mathbb{R}^m$ is the running Monte Carlo estimator for a test function $f : \mathcal{X} \to \mathbb{R}^m$. The condition requires: (i) existence of the limiting vector field $h^\infty(\vartheta) := \lim_{r \to \infty} r^{-1} h(r\vartheta)$, and (ii) global asymptotic stability of the ODE $\dot{\vartheta}_t = h^\infty(\vartheta_t)$. Throughout this section, we work under Assumption 1.

Recall from Section 3 that the joint SA recursion is

$$\vartheta_{n+1} = \vartheta_n + \gamma_{n+1} H(\vartheta_n, X_{n+1}), \tag{38}$$

where $H(\vartheta_n, x) = (s(x) - \theta_n, f(x) - \mu_n)$ for $\vartheta_n = (\theta_n, \mu_n)$. The associated mean field is

$$h(\vartheta_n) = h(\theta_n, \mu_n) = \begin{pmatrix} \mathcal{S}(\theta_n) - \theta_n \\ \mathcal{F}(\theta_n) - \mu_n \end{pmatrix}, \tag{39}$$

and the Jacobian of $h(\vartheta)$ has a block triangular structure:

$$\nabla_\vartheta h(\vartheta) = \begin{pmatrix} -I - \alpha \operatorname{Cov}_{\pi_\theta}(s, s) & 0 \\ -\alpha \operatorname{Cov}_{\pi_\theta}(f, s) & -I_m \end{pmatrix}. \tag{40}$$

The detailed derivatives of $\nabla_\vartheta h(\vartheta)$ are in Appendix C.6. For notation simplicity, we let $A(\theta) := I + \alpha \operatorname{Cov}_{\pi_\theta}(s, s) \succeq I$, and $B(\theta) := \alpha \operatorname{Cov}_{\pi_\theta}(f, s)$ only within this section (Appendix C.5).

For $\vartheta' = (\theta', \mu')$, consider the ray $r\vartheta' = (r\theta', r\mu')$. Using the fundamental theorem of calculus:

$$h(r\vartheta') = h(0) + \int_0^r \nabla h(t\vartheta') \cdot \vartheta' \, dt. \tag{41}$$

Since $h(0) = (\mathcal{S}(0) - 0, \mathcal{F}(0) - 0) = (0, \mu)$ where $\mu = \mathbb{E}_\pi[f(X)]$ is the quantity we want to estimate, we have:

$$h(r\vartheta') = \begin{pmatrix} 0 \\ \mu \end{pmatrix} + \int_0^r \begin{pmatrix} -A(t\theta') & 0 \\ -B(t\theta') & -I_m \end{pmatrix} \begin{pmatrix} \theta' \\ \mu' \end{pmatrix} dt. \tag{42}$$

Computing each component:

$$h_\theta(r\theta') = -\int_0^r A(t\theta')\theta' \, dt = -r\bar{A}_r \theta', \tag{43}$$

$$h_\mu(r\theta', r\mu') = \mu - \int_0^r B(t\theta')\theta' \, dt - r\mu' = \mu - r\bar{B}_r \theta' - r\mu', \tag{44}$$

where the averaged matrices are:

$$\bar{A}_r := \frac{1}{r} \int_0^r A(t\theta') \, dt = I_d + \frac{\alpha}{r} \int_0^r \operatorname{Cov}_{\pi_{t\theta'}}(s, s) \, dt, \tag{45}$$

$$\bar{B}_r := \frac{1}{r} \int_0^r B(t\theta') \, dt = \frac{\alpha}{r} \int_0^r \operatorname{Cov}_{\pi_{t\theta'}}(f, s) \, dt. \tag{46}$$

To establish existence of the limit $h^\infty(\vartheta')$, we analyze the behavior of $\operatorname{Cov}_{\pi_{t\theta'}}(s, s)$ and $\operatorname{Cov}_{\pi_{t\theta'}}(f, s)$ as $t \to \infty$. For large $t$, the surrogate $\pi_{t\theta'}(x) \propto \pi(x)e^{-\alpha t\theta'^\top s(x)}$ concentrates around the minimizer $x^*(t)$ of the effective potential $U_t(x) = U(x) + \alpha t\theta'^\top s(x)$. Writing $x = r\hat{x}$ with $\hat{x} \in \mathbb{S}^{d-1}$, Assumption 1(ii) governs the behavior of $\nabla_x^2 U(x)$ along rays to infinity.

**Lemma 5** (Convergence of Covariance Terms). *Under Assumption 1, suppose additionally that test function $f$ satisfies the growth condition $\|f(x)\| \le C_f(1 + \|x\|^q)$ for some $q < p$. Then for each $\theta \ne 0$ with $\hat{\theta} = \theta/\|\theta\|$:*

*(i) $t^{-(p-2)}\operatorname{Cov}_{\pi_{t\theta}}(s, s) \to \Sigma_{ss}(\hat{\theta})$ as $t \to \infty$,*

*(ii)* $t^{-(p-2)}\mathrm{Cov}_{\pi_{t\theta}}(f, s) \to \Sigma_{fs}(\hat{\theta})$ *as* $t \to \infty$,

*for some* $\Sigma_{ss}(\hat{\theta}) \succeq 0$ *and* $\Sigma_{fs}(\hat{\theta}) \in \mathbb{R}^{m \times d}$.

*Proof.* As $t \to \infty$, the surrogate $\pi_{t\theta}$ concentrates at the minimizer $x^*(t)$ of $U_t(x) = U(x) + \alpha t \theta^\top s(x)$. The first-order condition gives:
$$\nabla_x U(x^*) + \alpha t \nabla_x(\theta^\top s(x^*)) = 0 \implies s(x^*) = -\alpha t \nabla_x^2 U(x^*)\theta.$$

Under the super-linear tail growth from Assumption 1 (i), this implies $\|x^*(t)\| \to \infty$. From the scaling of $U(x) \sim c\|x\|^p$ and $s(x) \sim \|x\|^{p-1}$, one can show $\|x^*(t)\| = O(t)$.

By Laplace approximation, the covariance satisfies:
$$\mathrm{Cov}_{\pi_{t\theta}}(s, s) \approx \nabla_x s(x^*) \cdot \left[\nabla_x^2 U_t(x^*)\right]^{-1} \cdot \nabla_x s(x^*)^\top. \tag{47}$$

To see this, at large $t$, expanding $U_t$ to second order around $x^*$:
$$U_t(x) \approx U_t(x^*) + \frac{1}{2}(x - x^*)^\top H_t(x - x^*),$$

where $H_t := \nabla_x^2 U_t(x^*) \succ 0$ and the linear term vanishes since $\nabla_x U_t(x^*) = 0$. Thus, $\pi_{t\theta}$ is approximately Gaussian:
$$\pi_{t\theta}(x) \approx \mathcal{N}(x^*, H_t^{-1}).$$

For any smooth function $g : \mathbb{R}^d \to \mathbb{R}^k$, expand around $x^*$ to get:
$$g(x) \approx g(x^*) + \nabla_x g(x^*)^\top(x - x^*).$$

Under the Gaussian approximation, $\mathbb{E}[X - x^*] = 0$ and $\mathbb{E}[(X - x^*)(X - x^*)^\top] = H_t^{-1}$, so:
$$\mathbb{E}_{\pi_{t\theta}}[g(X)] \approx g(x^*), \quad \mathrm{Cov}_{\pi_{t\theta}}(g, g) \approx \nabla_x g(x^*) \cdot H_t^{-1} \cdot \nabla_x g(x^*)^\top.$$

Applying this to $g = s$ yields (47). This approximation is classical; see, e.g., Tierney & Kadane (1986); Wong (2001).

The asymptotic regularity condition in Assumption 1 (ii) states that $r^{-(p-2)}\nabla_x^2 U(r\hat{x}) \to M(\hat{x}) \succ 0$. Since $\nabla_x s = -\nabla_x^2 U$, both $\nabla_x s(x^*)$ and $\nabla_x^2 U_t(x^*)$ scale as $O(t^{p-2})$, yielding $\mathrm{Cov}_{\pi_{t\theta}}(s, s) = O(t^{p-2})$. The limit $\Sigma_{ss}(\hat{\theta})$ exists by the regularity of $M(\hat{\theta})$.

For part (ii), the growth condition $\|f(x)\| \le C_f(1 + \|x\|^q)$ with $q < p$ ensures that $f$ is dominated by the score behavior as $\|x\| \to \infty$. The same Laplace approximation argument gives convergence of $t^{-(p-2)}\mathrm{Cov}_{\pi_{t\theta}}(f, s)$. $\qquad\square$

**Proposition 6** (Existence of $h^\infty$). *Under Assumption 1 and the growth condition on $f$ from Lemma 5, the limit $h^\infty(\vartheta) := \lim_{r \to \infty} r^{-1}h(r\vartheta)$ exists for all $\vartheta \in \mathbb{R}^{d+m}$.*

*For targets with quadratic tails ($p = 2$), the limit takes the form:*
$$h^\infty(\theta', \mu') = -\begin{pmatrix} A_\infty(\hat{\theta}) & 0 \\ B_\infty(\hat{\theta}) & I_m \end{pmatrix}\begin{pmatrix} \theta' \\ \mu' \end{pmatrix}, \tag{48}$$

*where* $A_\infty(\hat{\theta}) = I_d + \alpha\Sigma_{ss}(\hat{\theta}) \succeq I_d$ *and* $B_\infty(\hat{\theta}) = \alpha\Sigma_{fs}(\hat{\theta})$.

*Proof.* From (43)-(44), the scaled mean field is:
$$\frac{h(r\vartheta')}{r} = \begin{pmatrix} -\bar{A}_r\theta' \\ \mu/r - \bar{B}_r\theta' - \mu' \end{pmatrix}. \tag{49}$$

We analyze the two cases based on the tail exponent $p$ from Assumption 1 (i).

*Case $p = 2$ (quadratic tails):* By Lemma 5, $\mathrm{Cov}_{\pi_{t\theta}}(s, s) \to \Sigma_{ss}(\hat{\theta})$ and $\mathrm{Cov}_{\pi_{t\theta}}(f, s) \to \Sigma_{fs}(\hat{\theta})$ as $t \to \infty$. By Cesaro's lemma (i.e., if $a_t \to a$, then $\frac{1}{r} \int_0^r a_t \, dt \to a$):

$$\bar{A}_r \to I_d + \alpha \Sigma_{ss}(\hat{\theta}) =: A_\infty(\hat{\theta}), \quad \bar{B}_r \to \alpha \Sigma_{fs}(\hat{\theta}) =: B_\infty(\hat{\theta}).$$

Since $\mu/r \to 0$, we obtain (48).

*Case $1 < p < 2$ (sub-quadratic tails):* The covariances decay as $t^{p-2} \to 0$. Thus:

$$\frac{1}{r} \int_0^r t^{p-2} \, dt = \frac{r^{p-2}}{p-1} \to 0.$$

Therefore, $\bar{A}_r \to I_d$ and $\bar{B}_r \to 0$, giving $h^\infty(\theta', \mu') = (-\theta', -\mu')$. □

**Proposition 7** (Stability of Augmented ODE@$\infty$). *Under Assumption 1, the ODE@$\infty$ given by $\dot{\vartheta}_t = h^\infty(\vartheta_t)$ is globally asymptotically stable with unique equilibrium $(0, 0)$.*

*Proof.* The ODE@$\infty$ has block triangular structure:

$$\dot{\theta}_t = -A_\infty(\hat{\theta}_t)\theta_t, \tag{50}$$

$$\dot{\mu}_t = -B_\infty(\hat{\theta}_t)\theta_t - \mu_t. \tag{51}$$

*Step 1: Stability of $\theta$-subsystem.* Since $A_\infty(\hat{\theta}) \succeq I_d$ for all $\hat{\theta}$:

$$\frac{d}{dt} \frac{1}{2} \|\theta_t\|^2 = \langle \theta_t, -A_\infty(\hat{\theta}_t)\theta_t \rangle = -\theta_t^\top A_\infty(\hat{\theta}_t)\theta_t \le -\|\theta_t\|^2.$$

Thus $\|\theta_t\| \le \|\theta_0\| e^{-t}$, establishing exponential convergence $\theta_t \to 0$.

*Step 2: Stability of $\mu$-subsystem.* With $\theta_t \to 0$ exponentially, equation (51) becomes a perturbed linear system. The homogeneous part $\dot{\mu} = -\mu$ is exponentially stable. The forcing term $-B_\infty(\hat{\theta}_t)\theta_t$ decays exponentially since $\|B_\infty(\hat{\theta})\|$ is bounded (under the growth condition on $f$) and $\|\theta_t\| \le \|\theta_0\| e^{-t}$.

By the variation of constants formula:

$$\mu_t = e^{-t}\mu_0 - \int_0^t e^{-(t-s)} B_\infty(\hat{\theta}_s)\theta_s \, ds.$$

The first term decays as $e^{-t}$. For the integral:

$$\left\| \int_0^t e^{-(t-s)} B_\infty(\hat{\theta}_s)\theta_s \, ds \right\| \le C \int_0^t e^{-(t-s)} e^{-s} ds \cdot \|\theta_0\| = C\|\theta_0\| t e^{-t} \to 0.$$

Thus $\mu_t \to 0$ as $t \to \infty$. Therefore, $(0, 0)$ is globally asymptotically stable for the ODE@$\infty$ (50)-(51). □

*Remark* 8 (Block triangular structure). The key simplification arises from the block triangular structure of $\nabla_\vartheta h(\vartheta)$: the $\theta$-dynamics decouple from $\mu$, allowing us to first establish stability of the score-history subsystem, then use it to drive stability of the estimator subsystem. This structure is a consequence of the fact that the surrogate $\pi_\theta$ depends only on $\theta$, not on $\mu$.

### C.6. Explicit Derivation of Jacobian Matrix $A^\star$

In this part, we compute the derivative of $h(\vartheta)$ at the unique equilibrium $\vartheta^\star = (0, \mu)$.

**Step 1: derivatives of $\mathcal{S}(\theta)$ and $\mathcal{F}(\theta)$.** For any measurable $g : \mathcal{X} \to \mathbb{R}^m$ with suitable integrability, write

$$\mathbb{E}_{\pi_\theta}[g(X)] = \int g(x)\pi_\theta(x)dx = \frac{\int g(x)\pi(x)e^{-\alpha\theta^\top s(x)}\,dx}{Z_\theta} = \frac{\int g(x)\pi(x)e^{-\alpha\theta^\top s(x)}\,dx}{\int \pi(x)e^{-\alpha\theta^\top s(x)}\,dx}.$$

Differentiating $\mathbb{E}_{\pi_\theta}[g(X)]$ with respect to $\theta$ yields

$$\nabla_\theta\mathbb{E}_{\pi_\theta}[g(X)] = -\alpha\int \pi_\theta(x)g(x)s(x)^\top dx + \alpha\left[\int \pi_\theta(x)g(x)dx\right]\left[\int \pi_\theta(x)s(x)dx\right]^\top = -\alpha\operatorname{Cov}_{\pi_\theta}(g,s), \quad (52)$$

Specializing to $g = s$ gives $\nabla_\theta\mathcal{S}(\theta) = -\alpha\operatorname{Cov}_{\pi_\theta}(s,s)$. Letting $g = f$ gives $\nabla_\theta\mathcal{F}(\theta) = -\alpha\operatorname{Cov}_{\pi_\theta}(f,s)$.

Evaluating at $\theta = 0$ (so that $\pi_\theta = \pi$) then yields

$$\nabla_\theta\mathcal{S}(0) = -\alpha\operatorname{Cov}_\pi(s,s), \qquad \nabla_\theta\mathcal{F}(0) = -\alpha\operatorname{Cov}_\pi(f,s).$$

**Step 2: assemble the Jacobian of $h$.** Recall $h(\theta,\mu) = (\mathcal{S}(\theta) - \theta,\ \mathcal{F}(\theta) - \mu)$. Hence $\nabla h(\vartheta^\star)$ has blocks

$$\partial_\theta h_\theta(\vartheta^\star) = \nabla_\theta\mathcal{S}(0) - I_d = -\alpha\operatorname{Cov}_\pi(s,s) - I_d, \qquad \partial_\mu h_\theta(\vartheta^\star) = 0,$$

$$\partial_\theta h_\mu(\vartheta^\star) = \nabla_\theta\mathcal{F}(0) = -\alpha\operatorname{Cov}_\pi(f,s), \qquad \partial_\mu h_\mu(\vartheta^\star) = -I_m,$$

which gives (11).

### C.7. Boundedness and Lipschitzness of the Jacobian $\nabla_\vartheta h$

We verify that the mean field $h : \mathbb{R}^{d+m} \to \mathbb{R}^{d+m}$ is continuously differentiable in $\vartheta = (\theta,\mu)$, and that its Jacobian $A = \nabla_\vartheta h$ is uniformly bounded and uniformly Lipschitz continuous.

*Proof.* Recall from Section 3 that the mean field is

$$h(\vartheta) = h(\theta,\mu) = \big(\mathcal{S}(\theta) - \theta,\ \mathcal{F}(\theta) - \mu\big).$$

**Step 1: Continuous differentiability of $h$.** The continuous differentiability of $h$ follows directly from the fact that the surrogate target $\pi_\theta$ is continuous and differentiable in $\theta$. Consequently, the functions $\mathcal{S}(\theta)$ and $\mathcal{F}(\theta)$ are also continuous and differentiable in $\theta$, which in turn implies that the mean field $h$ is continuously differentiable.

**Step 2: Uniform boundedness of $A$.** We show that $\|A(\vartheta)\|$ is uniformly bounded over $\vartheta \in \mathbb{R}^{d+m}$. By the block structure (40), it suffices to bound the covariance blocks.

*Bounding $\operatorname{Cov}_{\pi_\theta}(s,s)$:* By the Lipschitz property of the score (Assumption 1), $\|s(x)\| \le \|s(x_0)\| + L\|x - x_0\|$ for any fixed $x_0 \in \mathcal{X}$. Under $\pi_\theta$, the super-linear tail growth ensures that $\mathbb{E}_{\pi_\theta}[\|X\|^2] < \infty$ uniformly over bounded $\theta$-sets. Hence,

$$\|\operatorname{Cov}_{\pi_\theta}(s,s)\| \le \mathbb{E}_{\pi_\theta}[\|s(X)\|^2] < \infty.$$

For unbounded $\|\theta\|$, we use the ODE@$\infty$ analysis from Appendix C.5: as $\|\theta\| \to \infty$, the surrogate $\pi_\theta$ concentrates, and $\operatorname{Cov}_{\pi_\theta}(s,s) = O(\|\theta\|^{-(p-2)})$ by the Laplace approximation in Lemma 5, where $p > 1$ is the tail exponent. This yields a uniform bound

$$\sup_{\theta \in \mathbb{R}^d} \|\operatorname{Cov}_{\pi_\theta}(s,s)\| \le C_s < \infty.$$

*Bounding $\operatorname{Cov}_{\pi_\theta}(f,s)$:* By the growth condition $\|f(x)\| \le C_f(1 + \|x\|^q)$ with $q < p$ in Lemma 5 and Cauchy-Schwarz inequality,

$$\|\operatorname{Cov}_{\pi_\theta}(f,s)\| \le \sqrt{\mathbb{E}_{\pi_\theta}[\|f(X)\|^2]} \cdot \sqrt{\mathbb{E}_{\pi_\theta}[\|s(X)\|^2]} < \infty,$$

with the same uniformity argument as above. Therefore, $\|A(\vartheta)\| \le 1 + \alpha C_s + \alpha C_{fs} =: C_A$ for all $\vartheta$.

**Step 3: Uniform Lipschitz continuity of $A$.** We show that $\|A(\vartheta) - A(\vartheta')\| \leq L_A \|\vartheta - \vartheta'\|$ for some $L_A < \infty$.

Since $A$ depends only on $\theta$ (not on $\mu$), it suffices to prove Lipschitz continuity in $\theta$. The key step is to bound the derivative of the covariance with respect to $\theta$.

**Claim.** For vector-valued functions $g_1 : \mathcal{X} \to \mathbb{R}^{k_1}$ and $g_2 : \mathcal{X} \to \mathbb{R}^{k_2}$ with finite moments under $\pi_\theta$,

$$\frac{\partial}{\partial \theta_\ell} \mathrm{Cov}_{\pi_\theta}(g_1, g_2) = -\alpha \Big( \mathrm{Cov}_{\pi_\theta}(g_1 g_2^\top, s_\ell) - \mathrm{Cov}_{\pi_\theta}(g_1, s_\ell) \mathbb{E}_{\pi_\theta}[g_2]^\top - \mathbb{E}_{\pi_\theta}[g_1] \mathrm{Cov}_{\pi_\theta}(g_2, s_\ell)^\top \Big), \tag{53}$$

where $s_\ell$ denotes the $\ell$-th component of the score $s(x)$.

*Proof of Claim.* Recall that $\mathrm{Cov}_{\pi_\theta}(g_1, g_2) = \mathbb{E}_{\pi_\theta}[g_1 g_2^\top] - \mathbb{E}_{\pi_\theta}[g_1] \mathbb{E}_{\pi_\theta}[g_2]^\top$. By the product rule,

$$\frac{\partial}{\partial \theta_\ell} \mathrm{Cov}_{\pi_\theta}(g_1, g_2) = \frac{\partial}{\partial \theta_\ell} \mathbb{E}_{\pi_\theta}[g_1 g_2^\top] - \frac{\partial \mathbb{E}_{\pi_\theta}[g_1]}{\partial \theta_\ell} \mathbb{E}_{\pi_\theta}[g_2]^\top - \mathbb{E}_{\pi_\theta}[g_1] \frac{\partial \mathbb{E}_{\pi_\theta}[g_2]^\top}{\partial \theta_\ell}.$$

Using the derivative identity $\frac{\partial}{\partial \theta_\ell} \mathbb{E}_{\pi_\theta}[g] = -\alpha \mathrm{Cov}_{\pi_\theta}(g, s_\ell)$ in (52), we obtain (53). $\square$

**Bounding $\left\| \frac{\partial}{\partial \theta} \mathrm{Cov}_{\pi_\theta}(s, s) \right\|$.** Specializing (53) to $(g_1, g_2) = (s, s)$ and summing over $\ell = 1, \ldots, d$, we need to bound:

(i) **Third-order centered moment:** $\|\mathrm{Cov}_{\pi_\theta}(s_i s_j, s_\ell)\| = |\mathbb{E}_{\pi_\theta}[s_i s_j s_\ell] - \mathbb{E}_{\pi_\theta}[s_i s_j] \mathbb{E}_{\pi_\theta}[s_\ell]|$.

By the Cauchy–Schwarz inequality for covariances,

$$|\mathrm{Cov}_{\pi_\theta}(s_i s_j, s_\ell)| \leq \sqrt{\mathrm{Var}_{\pi_\theta}(s_i s_j)} \cdot \sqrt{\mathrm{Var}_{\pi_\theta}(s_\ell)} \leq \sqrt{\mathbb{E}_{\pi_\theta}[(s_i s_j)^2]} \cdot \sqrt{\mathbb{E}_{\pi_\theta}[s_\ell^2]}.$$

Since $(s_i s_j)^2 \leq \|s\|^4$ and $s_\ell^2 \leq \|s\|^2$, we have

$$|\mathrm{Cov}_{\pi_\theta}(s_i s_j, s_\ell)| \leq \sqrt{\mathbb{E}_{\pi_\theta}[\|s(X)\|^4] \cdot \mathbb{E}_{\pi_\theta}[\|s(X)\|^2]}.$$

(ii) **Second-order product terms:** $\|\mathrm{Cov}_{\pi_\theta}(s, s_\ell) \mathbb{E}_{\pi_\theta}[s]^\top\|$ and $\|\mathbb{E}_{\pi_\theta}[s] \mathrm{Cov}_{\pi_\theta}(s, s_\ell)^\top\|$.

Each factor satisfies

$$\|\mathrm{Cov}_{\pi_\theta}(s, s_\ell)\| \leq \sqrt{\mathbb{E}_{\pi_\theta}[\|s\|^2] \cdot \mathbb{E}_{\pi_\theta}[s_\ell^2]} \leq \mathbb{E}_{\pi_\theta}[\|s(X)\|^2],$$

and $\|\mathbb{E}_{\pi_\theta}[s]\| \leq \mathbb{E}_{\pi_\theta}[\|s(X)\|]$. Hence,

$$\|\mathrm{Cov}_{\pi_\theta}(s, s_\ell)\| \cdot \|\mathbb{E}_{\pi_\theta}[s]\| \leq \mathbb{E}_{\pi_\theta}[\|s(X)\|^2] \cdot \mathbb{E}_{\pi_\theta}[\|s(X)\|] \leq \mathbb{E}_{\pi_\theta}[\|s(X)\|^2]^{3/2},$$

where the last inequality uses Jensen's inequality $\mathbb{E}[\|s\|] \leq \sqrt{\mathbb{E}[\|s\|^2]}$.

Combining bounds (i) and (ii), and using sub-multiplicativity of norms across the $d^3$ index triples $(i, j, \ell)$,

$$\left\| \frac{\partial}{\partial \theta} \mathrm{Cov}_{\pi_\theta}(s, s) \right\| \leq \alpha d^{3/2} \Big( \sqrt{\mathbb{E}_{\pi_\theta}[\|s\|^4] \cdot \mathbb{E}_{\pi_\theta}[\|s\|^2]} + 2 \mathbb{E}_{\pi_\theta}[\|s\|^2]^{3/2} \Big). \tag{54}$$

**Uniform moment bounds.** Under Assumption 1, the super-linear tail growth $U(x) \geq c\|x\|^p$ for $\|x\| \geq R$ and score Lipschitzness $\|s(x)\| \leq L\|x\| + L_0$ imply:

- For any $q \geq 1$, $\mathbb{E}_{\pi_\theta}[\|s(X)\|^q] < \infty$ uniformly over compact $\theta$-sets, since the exponential tilt $e^{-\alpha \theta^\top s(x)}$ is dominated by the super-linear decay of $\pi(x)$.

- As $\|\theta\| \to \infty$, the Laplace approximation from Lemma 5 shows $\pi_\theta$ concentrates at a minimizer $x_\theta^*$ with $\mathbb{E}_{\pi_\theta}[\|s\|^q] = O(\|\theta\|^{q(p-1)})$, while $\mathrm{Cov}_{\pi_\theta}(s, s) = O(\|\theta\|^{-(p-2)})$ decays. Therefore, the product $\|\partial_\theta \mathrm{Cov}_{\pi_\theta}(s, s)\|$ remains bounded.

Thus, we can define

$$C' \triangleq \sup_{\theta \in \mathbb{R}^d} d^{3/2}\Big(\sqrt{\mathbb{E}_{\pi_\theta}[\|s\|^4] \cdot \mathbb{E}_{\pi_\theta}[\|s\|^2]} + 2\,\mathbb{E}_{\pi_\theta}[\|s\|^2]^{3/2}\Big) < \infty.$$

**Bounding** $\left\|\frac{\partial}{\partial\theta}\mathrm{Cov}_{\pi_\theta}(f,s)\right\|$. Applying (53) with $(g_1, g_2) = (f, s)$ and using the growth condition $\|f(x)\| \le C_f(1 + \|x\|^q)$ with $q < p$, an analogous argument yields

$$\left\|\frac{\partial}{\partial\theta}\mathrm{Cov}_{\pi_\theta}(f,s)\right\| \le \alpha d^{1/2}\Big(\sqrt{\mathbb{E}_{\pi_\theta}[\|f\|^2\|s\|^2] \cdot \mathbb{E}_{\pi_\theta}[\|s\|^2]} + \mathbb{E}_{\pi_\theta}[\|f\|^2]^{1/2}\mathbb{E}_{\pi_\theta}[\|s\|^2] + \mathbb{E}_{\pi_\theta}[\|f\|]\mathbb{E}_{\pi_\theta}[\|s\|^2]\Big).$$

The growth condition ensures all moments remain finite, giving a uniform bound $C'' < \infty$.

**Conclusion.** By the mean value theorem, for any $\theta, \theta' \in \mathbb{R}^d$,

$$\|A(\theta) - A(\theta')\| \le \sup_{\tilde{\theta} \in [\theta, \theta']} \|\nabla_\theta A(\tilde{\theta})\| \cdot \|\theta - \theta'\| \le \alpha(C' + C'')\|\theta - \theta'\|.$$

Setting $L_A := \alpha(C' + C'')$ completes the verification of uniform Lipschitz continuity. $\qquad\square$

### C.8. Proof of Theorem 3.3

The proof combines all technical results from the previous subsections. We apply the SA framework of Borkar et al. (2025) to the SRMC recursion (8).

*Proof.* For almost sure convergence and central limit theorem, we resort to Borkar et al. (2025, Theorem 1, Theorem 4).

**Step 1: Verify the SA assumptions.**

- *Step size conditions (A1, A5b):* Our choice of step size $\gamma_n = (n+1)^{-\rho}$ for $\rho \in (\frac{1}{2}, 1]$ $\sum_n \gamma_n = \infty$ and $\sum_n \gamma_n^2 < \infty$, which matches the Robbins-Monro conditions in Borkar et al. (2025, Assumption A1, A5b).

- *Lipschitz/linear growth (A2, A4):* Our Assumption 1 guarantees that $H(\vartheta, x)$ satisfies the 1-Lipschitz and linear growth requirements in Borkar et al. (2025, Assumption A2). Moreover, since our Lipschitz constant is independent of state $x$, Borkar et al. (2025, Assumption A4) is automatically satisfied.

- *ODE@$\infty$ condition (A3).* By Appendix C.5, the scaled mean field $r^{-1}h(r\vartheta)$ converges to a well-defined limit $h^\infty(\vartheta)$ as $r \to \infty$, and the ODE@$\infty$ $\dot{\vartheta}_t = h^\infty(\vartheta_t)$ is globally asymptotically stable. This removes the need for any *a priori* bounded-iterates assumption by ensuring that $\{\vartheta_n\}$ remains bounded almost surely, and satisfies Borkar et al. (2025, Assumption A3).

- *Differentiability of mean field and Jacobian condition (A5a).* By Appendix C.4, the ODE (37) has a unique globally asymptotically stable equilibrium $\vartheta^\star = (0, \mu)$. From Appendix C.6, we know that the Jacobian $A^\star$ of the ODE (37) is Hurwitz because all of its eigenvalues are strictly negative. Moreover, the Jacobian matrix satisfies uniform boundedness and uniformly Lipschitzness in Appendix C.7, which meets Borkar et al. (2025, Assumption A5a).

- *Base Markov kernel regularity (DV3 and kernel Lipschitzness):* Assumption 2 provides the uniform drift condition (32) and the kernel Lipschitzness (34). These match the controlled Markovian noise requirements in Borkar et al. (2025, (DV3)). The verification for SR-MH and SR-MALA is provided in Appendix C.3.

**Step 2: Conclude almost sure convergence and central limit theorem.** Under the preceding conditions, Borkar et al. (2025, Theorem 1 and Theorem 4) (applied to the $(d+m)$-dimensional iterate $\vartheta_n$) gives almost sure convergence and the CLT to the augmented iterates $\vartheta_n$.

$\qquad\square$

## C.9. Proof of Proposition 3.4

*Proof.* Let $\Sigma_\vartheta$ solve (13). Partition

$$\Sigma_\vartheta = \begin{bmatrix} \Sigma_{\theta\theta}(\alpha) & \Sigma_{\theta\mu}(\alpha) \\ \Sigma_{\mu\theta}(\alpha) & \Sigma_{\mu\mu}(\alpha) \end{bmatrix}, \qquad \Sigma_\Delta = \begin{bmatrix} U_{\theta\theta} & U_{\theta\mu} \\ U_{\mu\theta} & U_{\mu\mu} \end{bmatrix},$$

conformably with $(\theta, \mu) \in \mathbb{R}^{d+m}$. Let $\beta \triangleq 1 - 1_{\rho=1}/2$ so that $1_{\rho=1}/2 - 1 = -\beta$. Then, with $S \triangleq \mathrm{Cov}_\pi(s, s)$ and $C \triangleq \mathrm{Cov}_\pi(f, s)$, the Lyapunov equation (13) is equivalent to the block system

$$\left(-\beta I_d - \alpha S\right)\Sigma_{\theta\theta}(\alpha) + \Sigma_{\theta\theta}(\alpha)\left(-\beta I_d - \alpha S\right)^\top + U_{\theta\theta} = 0, \tag{55}$$

$$\left(-\beta I_d - \alpha S\right)\Sigma_{\theta\mu}(\alpha) + \Sigma_{\theta\mu}(\alpha)(-\beta I_m)^\top + \Sigma_{\theta\theta}(\alpha)(-\alpha C^\top) + U_{\theta\mu} = 0, \tag{56}$$

$$(-\beta I_m)\Sigma_{\mu\mu}(\alpha) + \Sigma_{\mu\mu}(\alpha)(-\beta I_m)^\top + (-\alpha C)\Sigma_{\theta\mu}(\alpha) + \Sigma_{\theta\mu}(\alpha)^\top(-\alpha C)^\top + U_{\mu\mu} = 0. \tag{57}$$

Assume an eigendecomposition of the positive semi-definite matrix $S = Q\Lambda Q^\top$ with the nonnegative diagonal matrix $\Lambda = \mathrm{diag}(\lambda_1, \ldots, \lambda_d)$. We now examine the property of $\Sigma_{\theta\theta}(\alpha)$. As mentioned in the main body, note that $\Sigma_\Delta$ is independent of $\alpha$. Define $\tilde{\Sigma}(\alpha) \triangleq Q^\top \Sigma_{\theta\theta}(\alpha)Q$ and $\tilde{U} \triangleq Q^\top U_{\theta\theta} Q$. Let

$$A(\alpha) \triangleq \beta I + \alpha\Lambda = \mathrm{diag}(\beta + \alpha\lambda_1, \ldots, \beta + \alpha\lambda_d)$$

for use within this proof only. Then we have $\beta I_d + \alpha S = QA(\alpha)Q^\top$, which we substitute into (55) to obtain

$$-QA(\alpha)Q^\top\Sigma_{\theta\theta}(\alpha) + \Sigma_{\theta\theta}(\alpha)\left(-QA(\alpha)Q^\top\right)^\top + U_{\theta\theta} = 0.$$

By multiplying the orthonormal basis $Q$ to the above equation, we have

$$-Q^\top QA(\alpha)Q^\top\Sigma_{\theta\theta}(\alpha)Q + Q^\top\Sigma_{\theta\theta}(\alpha)\left(-QA(\alpha)Q^\top\right)^\top Q + Q^\top U_{\theta\theta}Q = 0.$$

Therefore, by the definition of $\tilde{U}$, $Q^\top Q = I$, and $\tilde{\Sigma}(\alpha)$, we have

$$A(\alpha)\tilde{\Sigma}(\alpha) + \tilde{\Sigma}(\alpha)A(\alpha) = \tilde{U}. \tag{58}$$

Note that for diagonal $A(\alpha)$, the $(i, j)$-entry of (58) gives

$$(\beta + \alpha\lambda_i + \beta + \alpha\lambda_j)\tilde{\Sigma}_{ij}(\alpha) = \tilde{U}_{ij},$$

which yields

$$\tilde{\Sigma}_{ij}(\alpha) = \frac{\tilde{U}_{ij}}{2\beta + \alpha(\lambda_i + \lambda_j)} \tag{59}$$

after rearranging. Thus, $\tilde{\Sigma}(\alpha) = O(1/\alpha)$ element-wise for large $\alpha$. In turn, we have

$$\Sigma_{\theta\theta}(\alpha) = Q\tilde{\Sigma}(\alpha)Q^\top = O(1/\alpha). \tag{60}$$

Moreover, the Frobenius norm satisfies $\|\Sigma_{\theta\theta}(\alpha)\|_F = \|Q\tilde{\Sigma}(\alpha)Q^\top\|_F = \|\tilde{\Sigma}(\alpha)\|_F$, which follows from the invariance of the Frobenius norm under cyclic permutations and $Q^\top Q = I$. Then,

$$\|\tilde{\Sigma}(\alpha)\|_F^2 = \sum_{i,j} \frac{\tilde{U}_{ij}^2}{[2\beta + \alpha(\lambda_i + \lambda_j)]^2}.$$

Because $2\beta + \alpha(\lambda_i + \lambda_j) > 0$ holds for all $\alpha \geq 0$ and all $i, j$, it follows that for any $\alpha_1 \geq \alpha_2 \geq 0$, we obtain

$$\frac{\tilde{U}_{ij}^2}{[2\beta + \alpha_1(\lambda_i + \lambda_j)]^2} \leq \frac{\tilde{U}_{ij}^2}{[2\beta + \alpha_2(\lambda_i + \lambda_j)]^2} \leq \frac{\tilde{U}_{ij}^2}{[2\beta]^2}.$$

Summing over all $i, j$ leads to

$$\|\tilde{\Sigma}(\alpha_1)\|_F^2 \leq \|\tilde{\Sigma}(\alpha_2)\|_F^2 \leq \|\tilde{\Sigma}(0)\|_F^2,$$

or equivalently,

$$\|\Sigma_{\theta\theta}(\alpha_1)\|_F \leq \|\Sigma_{\theta\theta}(\alpha_2)\|_F \leq \|\Sigma_{\theta\theta}(0)\|_F. \tag{61}$$

This completes the proof of Proposition 3.4. $\qquad\square$

**Scaling of the limiting covariance blocks.** In this part, we establish the precise scaling of all remaining blocks of $\Sigma_\vartheta(\alpha)$ as $\alpha \to \infty$.

**(i) $\Sigma_{\theta\mu}(\alpha) = O(\alpha^{-1})$.** From (56), $\Sigma_{\theta\mu}$ solves the Sylvester equation

$$A(\alpha)\Sigma_{\theta\mu}(\alpha) + \Sigma_{\theta\mu}(\alpha)(\beta I_m) = U_{\theta\mu} - \alpha\Sigma_{\theta\theta}(\alpha)C^\top.$$

Since $\Sigma_{\theta\theta}(\alpha) = O(\alpha^{-1})$, the product $\alpha\Sigma_{\theta\theta}(\alpha)C^\top$ is $O(1)$, thus $\|U_{\theta\mu} - \alpha\Sigma_{\theta\theta}(\alpha)C^\top(\alpha)\| = O(1)$ as $\alpha \to \infty$.

The Sylvester equation has an integral representation

$$\Sigma_{\theta\mu}(\alpha) = \int_0^\infty e^{-A(\alpha)t}\left(U_{\theta\mu} - \alpha\Sigma_{\theta\theta}(\alpha)C^\top\right)e^{-\beta I_m t}\,dt. \tag{62}$$

Taking norms:

$$\|\Sigma_{\theta\mu}(\alpha)\| \leq \int_0^\infty e^{-\alpha\lambda_{\min}(S)t} \cdot e^{-\beta t}\,dt \cdot \|R(\alpha)\| = \frac{\|U_{\theta\mu} - \alpha\Sigma_{\theta\theta}(\alpha)C^\top\|}{\alpha\lambda_{\min}(S) + \beta}.$$

Since $\|U_{\theta\mu} - \alpha\Sigma_{\theta\theta}(\alpha)C^\top\| = O(1)$ and $\alpha\lambda_{\min}(S) + \beta = \Theta(\alpha)$, we obtain $\|\Sigma_{\theta\mu}(\alpha)\| = O(\alpha^{-1})$.

**(ii) $\Sigma_{\mu\mu}(\alpha) = O(1)$.** From (57), $\Sigma_{\mu\mu}$ solves

$$(\beta I_m)\Sigma_{\mu\mu}(\alpha) + \Sigma_{\mu\mu}(\alpha)(\beta I_m) = U_{\mu\mu} - \alpha C\Sigma_{\theta\mu}(\alpha) - \alpha\Sigma_{\theta\mu}(\alpha)^\top C^\top.$$

Define the right-hand side as $Z(\alpha) \triangleq U_{\mu\mu} - \alpha C\Sigma_{\theta\mu}(\alpha) - \alpha\Sigma_{\theta\mu}(\alpha)^\top C^\top$. Since $\Sigma_{\theta\mu}(\alpha) = O(\alpha^{-1})$, the coupling terms satisfy

$$\|\alpha C\Sigma_{\theta\mu}(\alpha)\| \leq \alpha\|C\|\|\Sigma_{\theta\mu}(\alpha)\| = O(1).$$

Thus $\|Z(\alpha)\| = O(1)$.

The solution is

$$\Sigma_{\mu\mu}(\alpha) = \int_0^\infty e^{-\beta I_m t}Z(\alpha)e^{-\beta I_m t}\,dt = \int_0^\infty e^{-2\beta t}Z(\alpha)\,dt = \frac{Z(\alpha)}{2\beta}.$$

Since $\beta > 0$ is independent of $\alpha$ and $\|Z(\alpha)\| = O(1)$, we conclude $\|\Sigma_{\mu\mu}(\alpha)\| = O(1)$.

**Summary.** The asymptotic scalings are:

| Block | Scaling | Interpretation |
|---|---|---|
| $\Sigma_{\theta\theta}(\alpha)$ | $O(\alpha^{-1})$ | Score-history variance vanishes |
| $\Sigma_{\theta\mu}(\alpha)$ | $O(\alpha^{-1})$ | Cross-covariance vanishes |
| $\Sigma_{\mu\mu}(\alpha)$ | $O(1)$ | Estimator variance bounded away from zero |

The key mechanism is that stronger repellence ($\alpha \uparrow$) increases the contraction rate in the $\theta$-dynamics via the drift $-\alpha S$, which propagates through the block structure: faster $\theta$-contraction reduces $\Sigma_{\theta\theta}$, which in turn bounds the forcing in the $\Sigma_{\theta\mu}$ equation, while $\Sigma_{\mu\mu}$ is insulated because its drift $-\beta I_m$ is $\alpha$-independent.

## C.10. Limiting Covariance of Monte Carlo Estimators in Specific Cases

We derive explicit formulas for the asymptotic covariance $\Sigma_{\mu\mu}(\alpha)$ under two instructive settings: (i) independent surrogate sampling, and (ii) Gaussian target with mean estimation. Throughout, we use the notation from Appendix C.9: $S \triangleq \mathrm{Cov}_\pi(s,s)$, $C \triangleq \mathrm{Cov}_\pi(f,s)$, $\beta \triangleq 1 - \frac{1_{\rho=1}}{2}$, and the Jacobian $A^\star$ from (11).

### C.10.1. INDEPENDENT SURROGATE SAMPLING

Suppose the base kernel $P_\theta$ draws $X_{n+1} \sim \pi_\theta$ independently at each step, conditional on $\theta_n = \theta$. This idealized setting isolates the effect of the score-tilt mechanism from temporal correlations in the Markov chain.

**Proposition 9** (Asymptotic covariance under independent sampling). *Under independent surrogate sampling, the asymptotic covariance of the Monte Carlo estimator satisfies*

$$\Sigma_{\mu\mu}(\alpha) = \text{Cov}_\pi(f, s)\, M(\alpha)^{-1} \text{Cov}_\pi(s, f) + R, \tag{63}$$

*where $M(\alpha) \triangleq \text{Cov}_\pi(s, s)\big(2\alpha\, \text{Cov}_\pi(s, s) + (2 - 1_{\rho=1})I\big)$ and $R \succeq 0$ is a residual matrix independent of $\alpha$. Consequently:*

(i) *$\Sigma_{\mu\mu}(\alpha) \preceq \Sigma_{\mu\mu}(0)$ for all $\alpha \geq 0$ in Loewner order;*

(ii) *$\Sigma_{\mu\mu}(\alpha) \searrow R$ in Loewner order (and thus also in Frobenius norm) as $\alpha \to \infty$;*

(iii) *The $\alpha$-dependent term satisfies $\text{Cov}_\pi(f, s)\, M(\alpha)^{-1} \text{Cov}_\pi(s, f) = O(\alpha^{-1})$.*

*Proof.* **Step 1: Structure of the noise covariance.** Under independent sampling from $\pi_\theta$, consecutive samples $(X_n, X_{n+1})$ are conditionally independent given $\theta_n$. As $\theta_n \to 0$ a.s., the limiting noise covariance $\Sigma_\Delta$ in (10) reduces to single-step variances:

$$U_{\theta\theta} = \text{Cov}_\pi(s, s) = S, \tag{64}$$

$$U_{\theta\mu} = \text{Cov}_\pi(s, f) = C^\top, \tag{65}$$

$$U_{\mu\mu} = \text{Cov}_\pi(f, f) =: V_f. \tag{66}$$

**Step 2: Solve for $\Sigma_{\theta\theta}(\alpha)$.** From the $(\theta, \theta)$-block of the Lyapunov equation (13):

$$(\beta I + \alpha S)\Sigma_{\theta\theta}(\alpha) + \Sigma_{\theta\theta}(\alpha)(\beta I + \alpha S) = S.$$

Using the eigendecomposition $S = Q\Lambda Q^\top$ with $\Lambda = \text{diag}(\lambda_1, \ldots, \lambda_d)$, the element-wise solution in the rotated basis $\tilde{\Sigma}_{\theta\theta}(\alpha) := Q^\top \Sigma_{\theta\theta}(\alpha)Q$ is:

$$[\tilde{\Sigma}_{\theta\theta}]_{ii} = \frac{\lambda_i}{2(\beta + \alpha\lambda_i)}, \quad [\tilde{\Sigma}_{\theta\theta}]_{ij} = 0 \text{ for } i \neq j. \tag{67}$$

In matrix form:

$$\Sigma_{\theta\theta}(\alpha) = Q\, \text{diag}\left(\frac{\lambda_i}{2(\beta + \alpha\lambda_i)}\right) Q^\top = \frac{1}{2}S(\beta I + \alpha S)^{-1}. \tag{68}$$

**Step 3: Solve for $\Sigma_{\theta\mu}(\alpha)$.** From the $(\theta, \mu)$-block:

$$(\beta I + \alpha S)\Sigma_{\theta\mu}(\alpha) + \Sigma_{\theta\mu}(\alpha)(\beta I) = C^\top - \alpha\Sigma_{\theta\theta}(\alpha)C^\top.$$

The solution via the integral representation is:

$$\Sigma_{\theta\mu}(\alpha) = \int_0^\infty e^{-(\beta I + \alpha S)t}\big(I - \alpha\Sigma_{\theta\theta}(\alpha)\big)C^\top e^{-\beta t}\, dt.$$

Since $(\beta I + \alpha S)$ and $\beta I$ commute with functions of $S$, and

$$I - \alpha\Sigma_{\theta\theta}(\alpha) = I - \frac{\alpha}{2}(\beta I + \alpha S)^{-1}S = \frac{1}{2}(\beta I + \alpha S)^{-1}(2\beta I + \alpha S),$$

this simplifies to:

$$\Sigma_{\theta\mu}(\alpha) = (2\beta I + \alpha S)^{-1}\big(I - \alpha\Sigma_{\theta\theta}(\alpha)\big)C^\top = \frac{1}{2}(\beta I + \alpha S)^{-1}C^\top. \tag{69}$$

**Step 4: Solve for $\Sigma_{\mu\mu}(\alpha)$.** From the $(\mu, \mu)$-block:

$$2\beta\Sigma_{\mu\mu}(\alpha) = V_f - \alpha C\Sigma_{\theta\mu}(\alpha) - \alpha\Sigma_{\theta\mu}(\alpha)^\top C^\top.$$

Define the $\alpha$-independent residual:

$$R \triangleq \frac{1}{2\beta}(V_f - CS^{-1}C^\top) = \frac{1}{2\beta}\text{Cov}_\pi(g, g) \succeq 0, \tag{70}$$

where we define $g(x) \triangleq f(x) - CS^{-1}s(x)$ in this part only. Equivalently, $V_f - CS^{-1}C^\top$ can be viewed as the Schur complement of $S$ in the joint covariance matrix of $(s, f)$, which ensures that the resulting matrix $R$ is positive semi-definite. Then:

$$\Sigma_{\mu\mu}(\alpha) - R = C \left[ \frac{1}{2\beta} S^{-1} - \frac{\alpha}{2\beta}(\beta I + \alpha S)^{-1} \right] C^\top.$$

Using

$$\frac{1}{2\beta} S^{-1} - \frac{\alpha}{2\beta}(\beta I + \alpha S)^{-1} = S^{-1}(2\alpha S + 2\beta I)^{-1},$$

we obtain

$$\Sigma_{\mu\mu}(\alpha) = R + C\big(S(2\alpha S + 2\beta I)\big)^{-1} C^\top, \tag{71}$$

and in the main body we use $M(\alpha) = S\big(2\alpha S + 2\beta I\big)$.

**Step 5: Loewner ordering.** For $\alpha_2 > \alpha_1 \geq 0$:

$$M(\alpha_2) = S(2\alpha_2 S + 2\beta I) \succeq S(2\alpha_1 S + 2\beta I) = M(\alpha_1) \succ 0,$$

since $S \succeq 0$ and $\alpha \mapsto 2\alpha S + 2\beta I$ is increasing in Loewner order.

By the operator monotonicity of matrix inversion on the positive definite cone:

$$M(\alpha_2)^{-1} \preceq M(\alpha_1)^{-1}.$$

Applying the congruence $C(\cdot)C^\top$ preserves the Loewner order:

$$C\, M(\alpha_2)^{-1} C^\top \preceq C\, M(\alpha_1)^{-1} C^\top.$$

By adding the $\alpha$-independent residual $R$ and from (71), we have:

$$\Sigma_{\mu\mu}(\alpha_2) \preceq \Sigma_{\mu\mu}(\alpha_1).$$

**Step 6: Asymptotic rate.** As $\alpha \to \infty$, assuming $S \succ 0$:

$$M(\alpha)^{-1} = (2\alpha S^2 + 2\beta S)^{-1} = \frac{1}{2\alpha} S^{-2}\big(I + \tfrac{\beta}{\alpha} S^{-1}\big)^{-1} = O(\alpha^{-1}).$$

Thus, $CM(\alpha)^{-1}C^\top = O(\alpha^{-1})$, and $\Sigma_{\mu\mu}(\alpha) \searrow R$ as $\alpha \to \infty$. $\qquad\square$

*Remark* 10 (Interpretation of the residual $R$). The residual $R = \frac{1}{2\beta}\mathrm{Cov}_\pi(g, g)$ represents the irreducible variance from the stochasticity of $g(X)$ under $\pi$, which cannot be reduced by increasing $\alpha$. The reducible component $CM(\alpha)^{-1}C^\top$ arises from the coupling between $f$ and the score $s$: when $C = \mathrm{Cov}_\pi(f, s) \neq 0$, the history-dependent score averaging induces negative correlations that reduce the Monte Carlo variance. The entire $\alpha$-effect resides in $M(\alpha)^{-1}$.

C.10.2. GAUSSIAN TARGET AND MEAN ESTIMATION

We now specialize to estimating the mean $\mathbb{E}_\pi[X]$ when the target is Gaussian: $\pi = \mathcal{N}(\mu, V)$ with $V \succ 0$.

**Proposition 11** (CLT for sample mean under Gaussian target). *Let $\pi = \mathcal{N}(\mu, V)$ and consider the test function $f(x) = x$ (mean estimation). For the sequence $\{X_i\}_{i=1}^n$ generated by SRMC with any base sampler satisfying Assumption 2, we have*

$$\sqrt{n}\left( \frac{1}{n}\sum_{i=1}^n X_i - \mathbb{E}_\pi[X] \right) \xrightarrow[n\to\infty]{dist.} \mathcal{N}(0, \Sigma_X(\alpha)), \tag{72}$$

*where the limiting covariance satisfies $\Sigma_X(\alpha) = V\Sigma_{\theta\theta}(\alpha)V^\top = O(1/\alpha)$, as $\alpha \to \infty$.*

*Proof.* **Step 1: Score and covariance structure for Gaussian.** For $\pi(x) = \mathcal{N}(\mu, V)$, the potential is $U(x) = \frac{1}{2}(x - \mu)^\top V^{-1}(x - \mu) + \mathrm{const}$, and the score is:

$$s(x) = -\nabla U(x) = -V^{-1}(x - \mu). \tag{73}$$

The score is an affine function of $x$, yielding:

$$S = \text{Cov}_\pi(s,s) = V^{-1}\text{Cov}_\pi(X,X)V^{-1} = V^{-1}, \tag{74}$$

$$C = \text{Cov}_\pi(f,s) = \text{Cov}_\pi(X, -V^{-1}(X-\mu)) = -V \cdot V^{-1} = -I_d. \tag{75}$$

**Step 2: Linear relationship between $X$ and $s(X)$.** From (73), we have the exact relationship:

$$X - \mu = -V \cdot s(X). \tag{76}$$

This implies that the centered sample mean is linearly related to the score average:

$$\frac{1}{n}\sum_{i=1}^{n}(X_i - \mu) = -V \cdot \frac{1}{n}\sum_{i=1}^{n} s(X_i).$$

**Step 3: Connect to the score history $\theta_n$.** For the standard averaging case ($\rho = 1$), the score history satisfies $\theta_n = \frac{1}{n}\sum_{i=0}^{n-1} s(X_i)$. Thus:

$$\frac{1}{n}\sum_{i=1}^{n} X_i - \mu = -V \cdot \theta_n + O(n^{-1}).$$

The $O(n^{-1})$ term arises from the boundary mismatch between $\sum_{i=1}^{n} s(X_i)$ and $\sum_{i=0}^{n-1} s(X_i)$, which is negligible after scaling by $\sqrt{n}$.

**Step 4: Transfer the CLT.** From Theorem 3.3, we have:

$$\sqrt{n}\,\theta_n \xrightarrow{d} \mathcal{N}(0, \Sigma_{\theta\theta}(\alpha)).$$

By the continuous mapping theorem applied to the linear transformation $\theta \mapsto -V\theta$:

$$\sqrt{n}\left(\frac{1}{n}\sum_{i=1}^{n} X_i - \mu\right) = -V \cdot \sqrt{n}\,\theta_n + o_p(1) \xrightarrow{d} \mathcal{N}(0, V\Sigma_{\theta\theta}(\alpha)V^\top). \tag{77}$$

Thus:

$$\Sigma_X(\alpha) = V\Sigma_{\theta\theta}(\alpha)V^\top. \tag{78}$$

**Step 5: Scaling.** From Proposition 3.4, $\Sigma_{\theta\theta}(\alpha) = O(\alpha^{-1})$. Thus,

$$\Sigma_X(\alpha) = V\Sigma_{\theta\theta}(\alpha)V^\top = O(\alpha^{-1}). \qquad \square$$

*Remark* 12 (Why the Gaussian case admits a closed form). The key simplification for Gaussian targets is the *exact linear relationship* (76) between $X - \mu$ and $s(X)$. This allows the CLT for the sample mean to be directly inherited from the CLT for the score history $\theta_n$. For non-Gaussian targets, $X - \mathbb{E}[X]$ and $s(X)$ are generally nonlinearly related, and the asymptotic covariance $\Sigma_{\mu\mu}(\alpha)$ involves more complex interactions through the full block structure of the Lyapunov equation.

*Remark* 13 (Sources of difficulty for general targets). For a general target $\pi$ and test function $f$, the asymptotic covariance $\Sigma_{\mu\mu}(\alpha)$ depends on:

(i) The cross-covariance $\text{Cov}_\pi(f,s)$, which couples the $\mu$-dynamics to the $\theta$-dynamics;

(ii) The temporal correlations in the base Markov chain, encoded in $\Sigma_\Delta$;

(iii) The nonlinear interaction between $f(X)$ and the score $s(X)$ under $\pi$.

When these factors do not simplify (as they do for independent sampling or Gaussian targets), the Lyapunov equation (13) must be solved numerically, and closed-form expressions for the $\alpha$-dependence of $\Sigma_{\mu\mu}(\alpha)$ are generally unavailable.

## C.11. Central Limit Theorem Under a Fixed Computational Budget for Gradient-Based SRMC

In gradient-based implementations of SRMC, each iteration requires evaluating the surrogate score

$$s_{\theta_n}(x) = s(x) + \alpha \nabla_x^2 U(x)\,\theta_n,$$

so the main additional cost relative to the baseline sampler is the Hessian–vector product $\nabla_x^2 U(x)\theta_n$. When this term is computed by a one-sided finite-difference approximation,

$$\nabla_x^2 U(x)\theta \approx \frac{\nabla_x U(x + \varepsilon\theta) - \nabla_x U(x)}{\varepsilon},$$

one extra gradient evaluation is needed beyond the baseline gradient computation. In addition, SRMC performs the $d$-dimensional SA update of $\theta_n$. Under the conservative cost model discussed in the main text, one may therefore regard a baseline gradient-based iteration as having cost $d$, while one SRMC iteration has cost $3d$: one gradient evaluation for the base proposal, one additional gradient evaluation for the finite-difference Hessian–vector product, and one $d$-dimensional SA update.

The purpose of this subsection is to translate the time-indexed CLT in Theorem 3.3 into a cost-indexed CLT under a fixed total computational budget, following the random-change-of-time argument used in Hu et al. (2025, Section 3.3). For clarity, we work in the averaging regime $\rho = 1$, so that $\gamma_n = (n+1)^{-1}$.

**Cost processes.** Let $a_i \in (0, \infty)$ denote the computational cost of the $i$-th iteration of the baseline gradient-based sampler, and let $b_i \in (0, \infty)$ denote the computational cost of the $i$-th iteration of its SRMC counterpart. For a total computational budget $B > 0$, define

$$T_{\text{base}}(B) := \max\Big\{k \geq 0 : \sum_{i=1}^k a_i \leq B\Big\}, \qquad T_{\text{SRMC}}(B) := \max\Big\{k \geq 0 : \sum_{i=1}^k b_i \leq B\Big\}.$$

Thus $T_{\text{base}}(B)$ and $T_{\text{SRMC}}(B)$ are the total numbers of iterations that can be performed by the baseline sampler and SRMC, respectively, before exhausting budget $B$.

We assume that there exist deterministic positive constants $C_{\text{base}}$ and $C_{\text{SRMC}}$ such that

$$\frac{B}{T_{\text{base}}(B)} \xrightarrow[B \to \infty]{a.s.} C_{\text{base}}, \qquad \frac{B}{T_{\text{SRMC}}(B)} \xrightarrow[B \to \infty]{a.s.} C_{\text{SRMC}}. \tag{79}$$

These constants are the asymptotic costs per effective iteration under the two schemes. Recall

$$\vartheta_n = (\theta_n, \mu_n) \in \mathbb{R}^{d+m}, \qquad \vartheta^\star = (0, \mu),$$

where $\mu = \mathbb{E}_\pi[f(X)]$. Under Assumptions 1 - 2 and $\rho = 1$, Theorem 3.3 yields

$$\sqrt{n}\,(\vartheta_n - \vartheta_\star) \xrightarrow[n \to \infty]{dist} \mathcal{N}(0, \Sigma_\vartheta).$$

**Theorem 14** (Cost-based CLT for gradient-based SRMC). *Assume the conditions of Theorem 3.3 with $\rho = 1$, and assume* (79). *Then,*

$$\sqrt{B}\,(\vartheta_{T_{\text{SRMC}}(B)} - \vartheta_\star) \xrightarrow[B \to \infty]{dist} \mathcal{N}\big(0,\, C_{\text{SRMC}}\Sigma_\vartheta\big), \tag{80}$$

$$\sqrt{B}\,(\vartheta_{T_{\text{base}}(B)} - \vartheta_\star) \xrightarrow[B \to \infty]{dist} \mathcal{N}\big(0,\, C_{\text{base}}\Sigma_\vartheta\big). \tag{81}$$

*Proof.* We follow the same random-change-of-time argument as in the proof of the cost-based CLT of Hu et al. (2025, Appendix F). Because $T_{\text{SRMC}}(B) \to \infty$ almost surely as $B \to \infty$, the random-change-of-time theorem implies

$$\sqrt{T_{\text{SRMC}}(B)}\,(\vartheta_{T_{\text{SRMC}}(B)} - \vartheta^\star) \xrightarrow[B \to \infty]{dist} \mathcal{N}(0, \Sigma_\vartheta). \tag{82}$$

Now rewrite

$$\sqrt{B}\left(\vartheta_{T_{\mathrm{SRMC}}(B)} - \vartheta^\star\right) = \sqrt{\frac{B}{T_{\mathrm{SRMC}}(B)}}\ \sqrt{T_{\mathrm{SRMC}}(B)}\left(\vartheta_{T_{\mathrm{SRMC}}(B)} - \vartheta^\star\right).$$

By (79),

$$\sqrt{\frac{B}{T_{\mathrm{SRMC}}(B)}} \xrightarrow[B\to\infty]{a.s.} \sqrt{C_{\mathrm{SRMC}}}.$$

Combining this with (82) and Slutsky's theorem yields

$$\sqrt{B}\left(\vartheta_{T_{\mathrm{SRMC}}(B)} - \vartheta_\star\right) \xrightarrow[B\to\infty]{dist} \mathcal{N}\left(0,\ C_{\mathrm{SRMC}}\Sigma_\vartheta\right),$$

which proves (80).

The baseline statement (81) is identical: apply the same argument to the time-domain CLT with the random index $T_{\mathrm{base}}(B)$. $\qquad\square$

**Discussion.** Theorem 14 makes explicit the tradeoff in gradient-based SRMC. Per iteration, SRMC is more expensive because the history-dependent surrogate score requires a Hessian–vector product, and a finite-difference implementation effectively adds one extra gradient evaluation. Nevertheless, Theorem 3.3 shows that increasing $\alpha$ reduces the asymptotic fluctuation of the history variable $\theta_n$, with $\Sigma_{\theta\theta}(\alpha) = O(1/\alpha)$. Therefore, whenever this variance reduction transfers sufficiently strongly to the estimator block $\Sigma_{\mu\mu}$, the extra constant factor in per-iteration cost can be offset, and SRMC yields smaller asymptotic error under the same computational budget.

This conclusion is especially explicit in the Gaussian mean-estimation setting. There, the sample-mean covariance satisfies

$$\Sigma_X(\alpha) = V\Sigma_{\theta\theta}(\alpha)V^\top = O(1/\alpha),$$

so under the constant-cost model the budget-normalized covariance becomes

$$3d\,\Sigma_X(\alpha) = O\left(\frac{d}{\alpha}\right).$$

Hence, for sufficiently large $\alpha$, the cost-adjusted asymptotic variance of SRMC can still be strictly smaller than that of the baseline gradient-based sampler, despite the threefold increase in per-iteration cost.

**Remark on $\rho < 1$.** The cost-based argument above extends verbatim to $\rho \in (1/2, 1)$, except that the normalization becomes $\gamma_n^{-1/2} \asymp n^{\rho/2}$ instead of $\sqrt{n}$. Since Hu et al. (2025, Section 3.3) is written in the $\sqrt{n}$-CLT regime, we state the fixed-budget theorem here under $\rho = 1$, which is also the most natural choice when $\theta_n$ is interpreted as the ordinary running average of past score evaluations.

### C.12. Proof of Proposition 3.6

*Proof.* We divide the proof into four steps.

**Step 1: Well-posedness and deterministic compactness of the iterates.** Since $\mathcal{X}$ is finite, $Z_\theta < \infty$ for every $\theta \in \mathbb{R}^d$, so $\pi_\theta$ is globally well-defined.

Define the compact convex sets

$$K_\theta \triangleq \mathrm{conv}\left(\{\theta_0\} \cup \{s(x) : x \in \mathcal{X}\}\right), \qquad K_\mu \triangleq \mathrm{conv}\left(\{\mu_0\} \cup \{f(x) : x \in \mathcal{X}\}\right).$$

From the recursion,

$$\theta_{n+1} = (1 - \gamma_{n+1})\theta_n + \gamma_{n+1}s(X_{n+1}),$$

$$\mu_{n+1} = (1 - \gamma_{n+1})\mu_n + \gamma_{n+1}f(X_{n+1}),$$

and the fact that $0 < \gamma_{n+1} \le 1$, it follows inductively that

$$\theta_n \in K_\theta, \qquad \mu_n \in K_\mu, \qquad \forall n \ge 0.$$

Hence

$$\vartheta_n \in K \triangleq K_\theta \times K_\mu, \qquad \forall n \ge 0,$$

where $K \subset \mathbb{R}^{d+m}$ is compact. In particular,

$$\sup_{n \ge 0} \|\vartheta_n\| < \infty \quad \text{deterministically},$$

and therefore

$$\sup_{n \ge 0} \mathbb{E}\|\vartheta_n\|^4 < \infty.$$

Thus, in the finite-state case, the boundedness conclusion in Borkar et al. (2025, Theorem 2) normally obtained through the ODE@$\infty$ argument is immediate from the recursion itself.

**Step 2: Global stability of the mean ODE.** This part fully follows from Appendix C.4 without any modification, i.e., $\vartheta^\star = (0, \mu^\star)$ is the unique globally asymptotically stable equilibrium of the full mean ODE $\dot{\vartheta} = h(\vartheta)$.

**Step 3: Cutoff extension and reduction to the framework of Borkar et al. (2025).** The original finite-state recursion need not admit a meaningful direct ODE@$\infty$ analysis in the form used in Appendix C.5. To place it within the framework of Borkar et al. (2025), we introduce a cutoff extension that coincides with the original recursion on the invariant compact set $\mathcal{K}$.

Choose an open bounded neighborhood $\mathcal{U} \subset \mathbb{R}^{d+m}$ of $\mathcal{K}$, and let

$$\chi : \mathbb{R}^{d+m} \to [0,1] \quad \text{and} \quad \chi \in C^\infty$$

satisfy

$$\chi \equiv 1 \text{ on } \mathcal{U}, \qquad \chi \equiv 0 \text{ outside a sufficiently large ball.}$$

Define the modified update map

$$\bar{H}(\vartheta, x) \triangleq \chi(\vartheta) H(\vartheta, x) - (1 - \chi(\vartheta))\vartheta,$$

where function $H$ is defined in (8). Since $\vartheta_n \in \mathcal{K} \subset \mathcal{U}$ for all $n$, we have $\chi(\vartheta_n) = 1$, and therefore

$$\bar{H}(\vartheta_n, X_{n+1}) = H(\vartheta_n, X_{n+1}), \qquad \forall n \ge 0.$$

Hence the modified recursion

$$\bar{\vartheta}_{n+1} = \bar{\vartheta}_n + \gamma_{n+1} \bar{H}(\bar{\vartheta}_n, X_{n+1})$$

is pathwise identical to the original recursion when started from the same initial condition.

Let

$$\bar{h}(\vartheta) := \mathbb{E}_{\pi_\theta}[\bar{H}(\vartheta, X)].$$

Then $\bar{h} = h$ on $U$, so $\bar{h}$ has the same equilibrium $\vartheta^\star$ and the same Jacobian at $\vartheta^\star$ as the original mean field. Outside a sufficiently large ball,

$$\bar{h}(\vartheta) = -\vartheta.$$

Therefore the scaled field satisfies

$$r^{-1}\bar{h}(r\vartheta) \to -\vartheta \qquad \text{as } r \to \infty,$$

uniformly on compact subsets of $\mathbb{R}^{d+m}$. The associated ODE@$\infty$ is

$$\dot{\vartheta} = -\vartheta,$$

which is globally asymptotically stable.

Moreover, $\bar{H}$ is globally Lipschitz with linear growth, and Assumption 2 for the kernel family remains unchanged. Thus, the cutoff recursion satisfies the assumptions required in Appendix C.8 for the application of the SA asymptotic-statistics results of Borkar et al. (2025). Since the cutoff recursion is pathwise identical to the original one, all conclusions transfer directly to the original recursion.

**Step 4: Jacobian and central limit theorem.** Because $\mathcal{X}$ is finite, differentiation under the finite sum is immediate. For any bounded function $g : \mathcal{X} \to \mathbb{R}^k$, define

$$G_g(\theta) := \mathbb{E}_{\pi_\theta}[g(X)].$$

Then

$$\nabla_\theta G_g(\theta) = -\alpha \Big( \mathbb{E}_{\pi_\theta}[g(X)s(X)^\top] - \mathbb{E}_{\pi_\theta}[g(X)] \, \mathbb{E}_{\pi_\theta}[s(X)]^\top \Big) = -\alpha \mathrm{Cov}_{\pi_\theta}(g, s).$$

Applying this with $g = s$ and $g = f$ yields

$$\nabla_\theta S(\theta) = -\alpha \mathrm{Cov}_{\pi_\theta}(s, s), \qquad \nabla_\theta F(\theta) = -\alpha \mathrm{Cov}_{\pi_\theta}(f, s).$$

Therefore, at $\vartheta^\star = (0, \mu^\star)$,

$$A^\star = \nabla h(\vartheta^\star) = \begin{bmatrix} -I_d - \alpha \mathrm{Cov}_\pi(s, s) & 0 \\ -\alpha \mathrm{Cov}_\pi(f, s) & -I_m \end{bmatrix}.$$

This matrix is block lower triangular with diagonal blocks

$$-I_d - \alpha \mathrm{Cov}_\pi(s, s) \quad \text{and} \quad -I_m.$$

Since $\mathrm{Cov}_\pi(s, s) \succeq 0$, both diagonal blocks are Hurwitz, and hence so is $A^\star$. This Jacobian form is eactly the same as Appendix C.6 in the continuous-domain case.

Applying Borkar et al. (2025, Theorem 4) to the cutoff recursion, and using pathwise identity with the original recursion, we conclude that

$$\vartheta_n \xrightarrow[n \to \infty]{a.s.} \vartheta_\star$$

and

$$\gamma_n^{-1/2}(\vartheta_n - \vartheta_\star) \xrightarrow[n \to \infty]{dist.} \mathcal{N}(0, \Sigma_\vartheta),$$

where $\Sigma_\vartheta$ is the unique positive semidefinite solution of the same Lyapunov equation as in Theorem 3.3. This completes the proof. $\qquad \square$

# D. Simulation Setup for Section 4 and Additional Experiments

This section provides the simulation setup for Section 4 and additional simulation results that illustrate how our SRMC framework substantially enhances the mode-covering capability of the underlying MCMC sampler, meaning it thoroughly explores the target distribution while preserving unbiasedness. Appendix D.2 reports additional CIFAR-10 EBM results in continuous domain, including the test of Gaussian mixture validation in real-world dataset. Appendix D.3 reports additional Static MNIST results in discrete configuration space, including qualitative trajectories and AIS-based diversity evaluation.

## D.1. Simulation Setup

We provide complete specifications of the target distributions and algorithmic hyperparameters used in Section 4.2.

### D.1.1. TARGET DISTRIBUTIONS

**Correlated Gaussian ($D = 10$, $\rho = 0.9$).** The target distribution is a $D$-dimensional Gaussian with exponentially decaying correlation structure:

$$\pi(x) = \mathcal{N}(x; 0, \Sigma), \quad \Sigma_{ij} = \rho^{|i-j|}, \tag{83}$$

with $D = 10$ and $\rho = 0.9$. The resulting covariance matrix has a condition number $\kappa(\Sigma) \approx 19$, creating anisotropic level sets where the ratio between the largest and smallest principal axes spans nearly two orders of magnitude. The score function is $s(x) = -\Sigma^{-1}x$, and the true mean is $\mu^* = 0$.

**Bayesian Logistic Regression ($D = 10$, $N = 100$).** We consider posterior inference for binary classification with a Gaussian prior. The generative model is:

$$y_i \mid z_i, x \sim \text{Bernoulli}(\sigma(z_i^\top x)), \quad x \sim \mathcal{N}(0, I), \tag{84}$$

where $\sigma(\cdot)$ denotes the logistic sigmoid function. We generate synthetic data with $N = 100$ observations and $D = 10$ parameters. The design matrix $\mathbf{Z} \in \mathbb{R}^{N \times D}$ is drawn from $\mathcal{N}(0, V)$ where $V$ induces moderate multicollinearity. True regression coefficients $x \sim \mathcal{N}(0, I)$ are used to generate binary labels.

The unnormalized posterior is:

$$\tilde{\pi}(x) = \exp\left(-\frac{\|x\|^2}{2}\right) \prod_{i=1}^{N} \sigma(z_i^\top x)^{y_i}(1 - \sigma(z_i^\top x))^{1-y_i}, \tag{85}$$

with score function:

$$s(x) = -x + \sum_{i=1}^{N}(y_i - \sigma(z_i^\top x))z_i. \tag{86}$$

The ground-truth posterior mean $\mu^*$ is estimated via a long-run HMC chain ($10^6$ iterations) and used as reference for MSE computation.

### D.1.2. SAMPLER CONFIGURATIONS

**MALA and SR-MALA.** The implementation of MALA and SR-MALA is referred to Algorithm 4. Both samplers start the initial position $X_0$ from the target distribution. We use fine-tuned step size $\eta = 0.01$ for correlated Gaussian and $\eta = 0.005$ for logistic regression, tuned to achieve good acceptance rates in the range $[0.5, 0.8]$.

**HMC and SR-HMC.** Algorithm 5 details the implementation of HMC and SR-HMC. We use $L = 10$ leapfrog steps with fine-tuned step size $\eta = 0.2$ for correlated Gaussian and $\eta = 0.03$ for logistic regression. The mass matrix is set to identity. For SR-HMC, the history variable $\theta_n$ is held constant throughout leapfrog integration and updated only after the Metropolis accept/reject step.

### D.1.3. EVALUATION METRICS

**Mean Squared Error (MSE).**   The primary convergence metric measures the squared distance between the running sample mean of the trajectory $\{X_n\}$ and the true mean:

$$\text{MSE}(t) = \|\bar{x}_t - \mu\|^2, \quad \bar{x}_t = \frac{1}{t} \sum_{n=1}^{t} X_n. \tag{87}$$

### D.1.4. HYPERPARAMETER SENSITIVITY AND ADAPTIVE-$\alpha$ HEURISTICS

This subsection reports the follow-up experiments underlying the practical guidance in Section 4.1 and the brief adaptive-$\alpha$ discussion at the end of Section 4.2. Unless noted otherwise, all runs use the tuned baselines and a burn-in fraction of $0.3$.

**Sensitivity to the stochastic-approximation exponent $\rho$.**   Figure 4 reports the tuned-baseline $\rho$-sweep on the correlated Gaussian target for both MALA and HMC. We focus on this target because it provides the cleanest isolation of the history-update schedule. The qualitative ordering is stable: $\rho = 0.6$ and $\rho = 0.8$ are the strongest practical choices, whereas $\rho = 1.0$ is consistently the weakest over the tested horizons. In particular, even after extending the comparison to 500k steps, we did not observe a crossover in which $\rho = 1.0$ overtook $\rho = 0.6$. Thus, under the tuned baseline, the main practical competition is between $\rho = 0.6$ and $\rho = 0.8$, which supports the recommendation in Section 4.1 to use $\rho \in \{0.6, 0.8\}$ as the default operating range.

**Sensitivity to the finite-difference scale $\epsilon$.**   Figure 5 reports the tuned-baseline $\epsilon$-sweep on the Bayesian logistic target at $\alpha = 1$, where finite-difference error is meaningful because the score is nonlinear. We test

$$\epsilon \in \{10^{-5}, 10^{-3}, 0.1, 1, \alpha\},$$

where $\epsilon = \alpha$ corresponds to the shifted-gradient counterpart. The clearest effect appears for MALA: very small $\epsilon$ values are clearly harmful, whereas $\epsilon = 0.1, 1$, and $\epsilon = \alpha$ perform similarly well. By contrast, the HMC curves are much less sensitive. At the tuned baseline and 100k steps, the final MALA MSE at $\alpha = 1$ deteriorates from $2.46 \times 10^{-6}$ at $\epsilon = 0.1$ to $1.15 \times 10^{-3}$ at $\epsilon = 10^{-5}$, while all five HMC choices remain around $10^{-5}$. The most plausible explanation is numerical cancellation in the finite-difference score difference when the perturbation scale is taken too small. These runs therefore support the practical rule stated in Section 4.1: $\epsilon$ should not be chosen overly small, and values on the order of $\alpha$ work well in this nonlinear example.

**Adaptive $\alpha$ as a robustness mechanism.**   Figure 6 compares fixed-$\alpha$ and adaptive-$\alpha$ screening at 10k matched steps over nominal values $\{0.5, 1, 2, 3, 5\}$. The main pattern is that adaptive $\alpha$ should be interpreted as a robustness mechanism, not as a universal replacement for the hindsight-best fixed choice. The clearest practical gain appears in the more aggressive Bayesian-logistic regime, especially for HMC, where adaptive warmup prevents large transient over-tilting.

We consider two simple rules. The first is a *capped warmup-plus-freeze* schedule based on

$$\alpha_k = \frac{k}{C + k/\alpha_{\text{ref}}},$$

but allowed to increase only during a short warmup period and then frozen. Let $N$ denote the total matched computational budget and define

$$N_w = \min(3000, 0.1N).$$

For MALA, the adaptive period is the first $N_w$ iterations. For HMC, if one outer iteration uses $L$ leapfrog steps, we use

$$K_w = \left\lfloor \frac{N_w}{L} \right\rfloor$$

warmup outer iterations. During warmup, we choose $C$ so that the endpoint reaches a fixed fraction $\bar{\rho}_{\text{cap}} \alpha_{\text{ref}}$ of the nominal reference value,

$$\alpha_{K_w} = \bar{\rho}_{\text{cap}} \alpha_{\text{ref}}, \qquad \bar{\rho}_{\text{cap}} = 0.8,$$

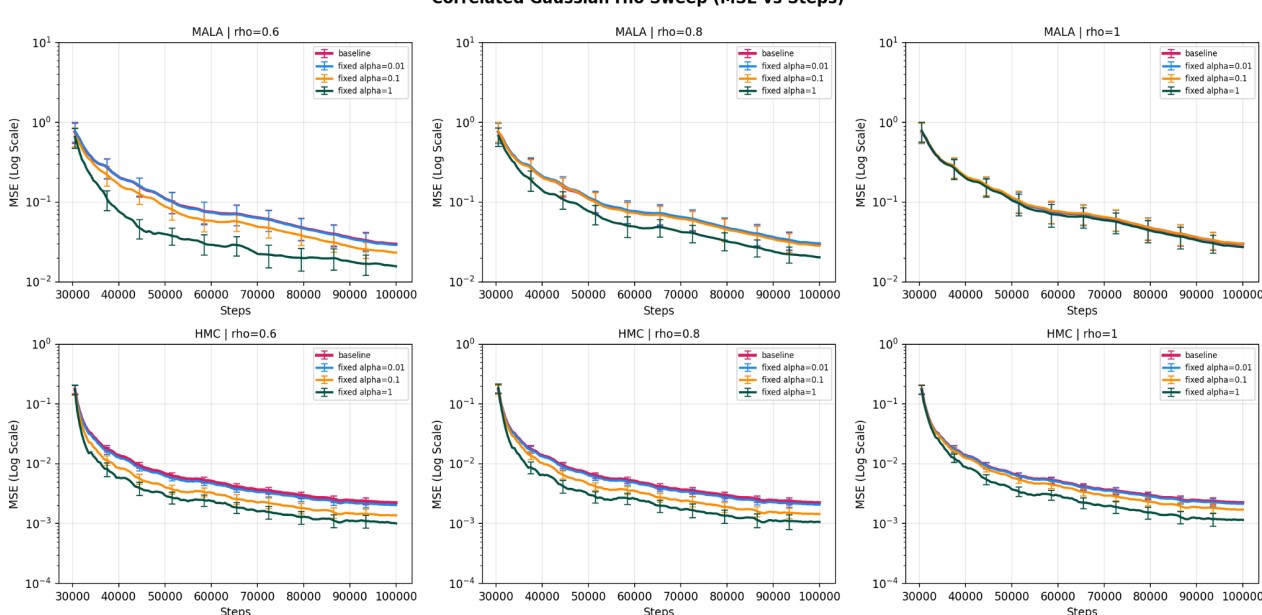

*Figure 4.* Tuned-baseline $\rho$-sensitivity on the correlated Gaussian target. The main practical competition is between $\rho = 0.6$ and $\rho = 0.8$, while $\rho = 1.0$ remains weakest over the tested horizons.

and then freeze at the realized value $\hat{\alpha} = \alpha_{K_w}$. In the 100k comparisons, the intended frozen working values are therefore approximately $0.8, 1.6$, and $4.0$ for $\alpha_{\text{ref}} = 1, 2, 5$, respectively.

Our second rule adds an *exponent-scale guardrail* to the same increasing schedule. While $\alpha_k$ is still increasing, we monitor $\alpha_k |s(X_k)^\top \theta_k|$, compute a rolling $95\%$ quantile over windows of size $\max(10, \lfloor K_w/20 \rfloor)$, and compare it with a threshold $\tau = 1$. If that rolling $95\%$ quantile exceeds $\tau$ in two consecutive windows, we stop increasing $\alpha_k$ and freeze at the last safe value. Thus, the guardrail differs from the capped rule only in allowing an earlier, data-dependent freeze when the effective exponent scale becomes too aggressive.

The interpretation of Figure 6 is therefore target dependent. On the correlated Gaussian target, where larger fixed $\alpha$ is already comparatively stable, adaptive $\alpha$ is not expected to outperform the best fixed choice. On Bayesian logistic, however, adaptive $\alpha$ clearly protects against overly aggressive nominal values. For example, for HMC at 100k steps, the final MSE at fixed $\alpha = 2$ and $5$ deteriorate to $1.55 \times 10^{-3}$ and $1.94 \times 10^{-2}$, whereas the corresponding capped-adaptive runs reduce these values to $1.12 \times 10^{-5}$ and $2.53 \times 10^{-3}$. The correct interpretation is therefore not that adaptive $\alpha$ uniformly beats the best fixed choice, but rather that it substantially reduces sensitivity to the nominal $\alpha$ level when the stable fixed-$\alpha$ range is unknown. Across the four tuned-baseline blocks, both adaptive rules improve robustness, and the exponent-scale guardrail gives the clearest protection in the aggressive HMC Bayesian-logistic regime.

**Practical takeaway.** Combining Figures 4–6 with the main continuous experiments, the most defensible practical message is: use $\rho$ around $0.6 \sim 0.8$ rather than $\rho = 1$; avoid overly small $\epsilon$, and prefer values on the order of $\alpha$ on nonlinear targets; and treat adaptive $\alpha$ as a robust default when the stable fixed-$\alpha$ range is unknown, especially in aggressive regimes where fixed large $\alpha$ can fail sharply.

### D.2. CIFAR-10 EBM Sampling

We evaluate the efficacy of the proposed SRMC in continuous configurational state spaces. Unlike discrete settings where samples occupy well-defined states, continuous spaces present unique challenges for mode exploration, as samples can drift continuously through the energy landscape. Our experiments systematically demonstrate that the score repellent mechanism significantly improves mode coverage compared to the unadjusted Langevin algorithm (ULA) in Algorithm 3. All experiments operate in the continuous space $x \in \mathbb{R}^{3 \times 32 \times 32}$ representing RGB images of CIFAR-10 resolution. Unlike discrete MCMC (e.g., Gibbs sampling on binary MNIST), Langevin Dynamics performs gradient-based updates

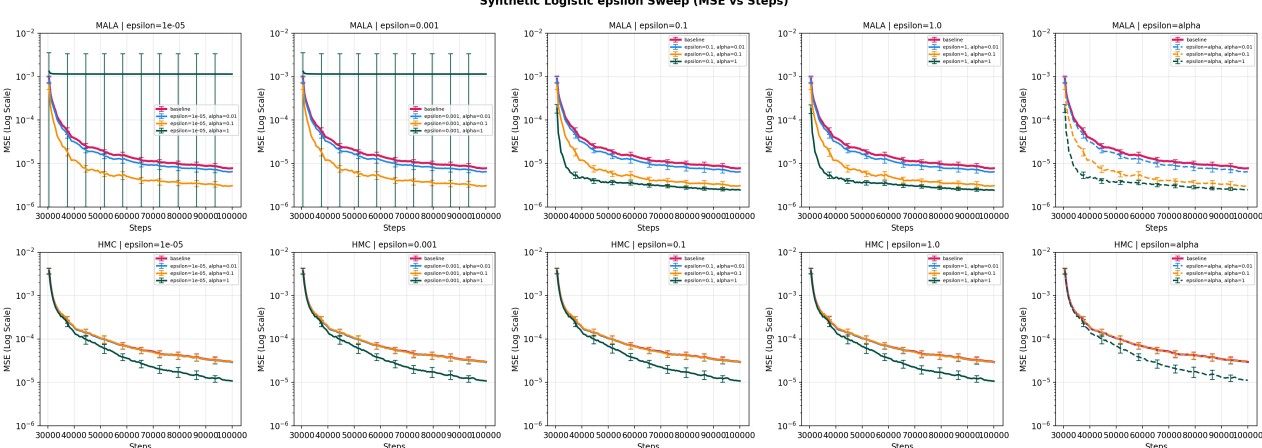

*Figure 5.* Tuned-baseline $\epsilon$-sensitivity on the Bayesian logistic target at $\alpha = 1$. Very small $\epsilon$ values are clearly harmful for MALA, whereas $\epsilon = 0.1$, 1, and $\epsilon = \alpha$ are all stable.

with continuous Gaussian noise.

### D.2.1. EXPERIMENT 1: GAUSSIAN MIXTURE VALIDATION

**Objective.** To validate the mode exploration capability of SRMC in a controlled setting with known ground truth, we construct a synthetic Gaussian mixture benchmark. This isolates the effect of the score repellent mechanism from potential confounds in pre-trained model behavior, as we can compute exact scores analytically.

**Target Distribution.** We define a Gaussian mixture model with 1,000 modes:

$$\pi(x) \propto \sum_{k=1}^{1000} \exp\left(-\frac{\|x - \mu_k\|^2}{2\sigma^2}\right), \tag{88}$$

where $\{\mu_k\}_{k=1}^{1000}$ are 1,000 CIFAR-10 training images (100 per class, 10 classes) serving as mode centers, and $\sigma = 4$. The score function $\nabla_x \log \pi(x)$ is computed analytically via the softmax-weighted average of gradients toward each mode center, where no pre-trained neural network is involved.

**Setup.** To create a worst-case initialization scenario, all 50 parallel chains are initialized at the *same* mode (a single CIFAR-10 image). This tests the sampler's ability to escape local optima and discover distant modes without diverse initialization. The Langevin step size in both LD and Algorithm 3 is $\eta = 1$ to facilitate traversal across well-separated modes. The strength of repellence is set to $\alpha = 0.15$, and the perturbation bandwidth $\epsilon = 10^{-3}$ controls the finite-difference approximation of the Hessian-vector product, $\nabla_x^2 \log \pi(x)\theta \approx \left(\nabla_x \log \pi(x + \epsilon\,\theta) - \nabla_x \log \pi(x)\right)/\epsilon$. The step size $\gamma_n$ follows the standard decaying schedule $\gamma_n = 1/(n+1)^{0.6}$ with $\rho = 0.6$ to achieve good transient performance. All 50 chains evolve independently, each maintaining its own score history with intra-chain repulsion and no shared state or inter-chain communication.

**Hyperparameter choice and sensitivity.** For the Gaussian-mixture benchmark, we select the main setting $(\alpha, \epsilon) = (0.15, 10^{-3})$ based on a coarse grid search balancing exploration speed and numerical stability. A follow-up ablation at 500 steps (50 chains; differences are most visible before near-full coverage is reached) showed that mode discovery is robust across a broad range of repellence strengths: $\alpha \in \{10^{-3}, 10^{-2}, 0.1, 0.15, 0.3, 1.0\}$ all gave competitive coverage, while excessively large values (e.g., $\alpha \geq 5$) degraded performance. We also monitored the relative size of the Hessian-vector correction, $\|H_\theta\|/\|s\|$, which remained in the 0.1 - 0.2 range across the tested values and only exhibited divergent behavior for extremely large $\alpha$ (around $\alpha \geq 100$).

Holding $\alpha = 0.15$ fixed, an ablation over $\epsilon \in \{10^{-6}, 10^{-5}, 10^{-4}, 10^{-3}, 10^{-2}\}$ showed that overly small values are dominated by floating-point noise, whereas larger values remain competitive; in particular, $\epsilon = 10^{-3}$ and $\epsilon = 10^{-2}$ performed similarly, while $\epsilon \leq 10^{-5}$ substantially degraded mode coverage. We therefore chose $\epsilon = 10^{-3}$ as the smallest value that avoids numerical instability while retaining strong empirical performance.

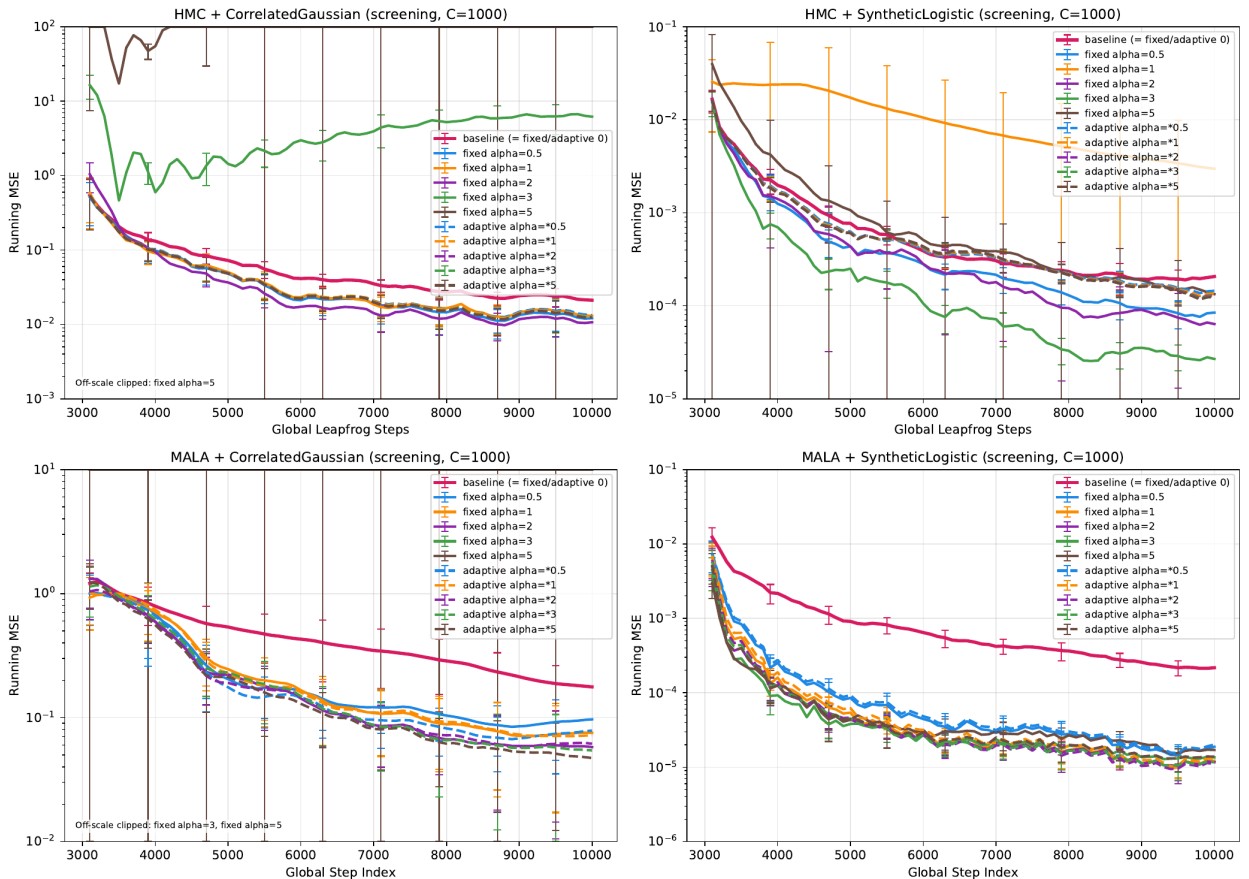

*Figure 6.* Fixed-$\alpha$ versus capped adaptive-$\alpha$ screening at 10k matched steps for nominal values $\{0.5, 1, 2, 3, 5\}$. The clearest practical gain appears in the aggressive Bayesian-logistic regime, especially for HMC, where adaptive warmup prevents large transient over-tilting.

**Mode Assignment.** At each step, each sample is assigned to the nearest mode center (argmin of Euclidean distance to the 1,000 $\mu_k$). We track the cumulative union of visited modes across all 50 chains.

**Results.** Table 2 and Figure 7 highlight the exploration efficiency gap. While ULA stagnates near initialization (indices 400–500) with only 2.8% coverage after 5,000 steps, SR-ULA leverages temporal history to 'reshape' the target $\pi_\theta$ and escape local optima. This mechanism drives chains apart to ensure uniform traversal, achieving complete discovery of all 1,000 modes within just $\sim$1,035 steps.

*Table 2.* Mode coverage on Gaussian Mixture (1,000 modes). All 50 chains initialized at the same mode.

| Method | Modes Discovered | Steps to Full Coverage |
|---|---|---|
| ULA | 28 / 1000 (2.8%) | – (never achieved) |
| SR-ULA | **1000 / 1000 (100%)** | $\sim$ **1035 steps** |

### D.2.2. EXPERIMENT 2: SINGLE-CHAIN TRAJECTORY ANALYSIS

**Objective.** While Experiment 1 uses a synthetic target with exact scores, real-world applications involve learned energy functions. This experiment evaluates whether SRMC improves temporal mixing when sampling from a pre-trained energy-based model (EBM), analyzing the trajectory of a *single* chain over time.

**Target Distribution.** The target is the Boltzmann distribution $\pi(x) \propto \exp(-E_\theta(x))$ where $E_\theta$ is a pre-trained CIFAR-10

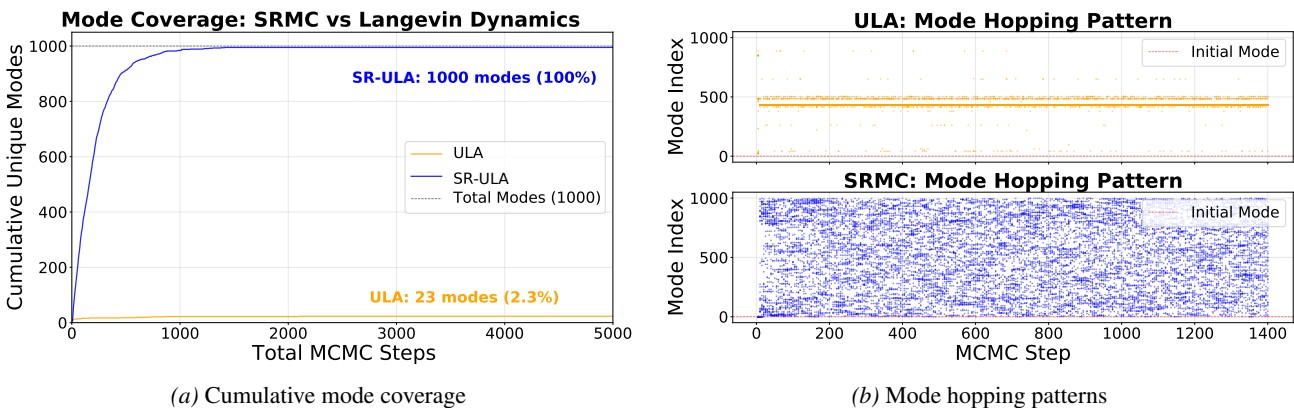

*(a)* Cumulative mode coverage    *(b)* Mode hopping patterns

*Figure 7.* **Exploration efficiency on Gaussian Mixture (1,000 modes).** (a) SR-ULA achieves complete coverage in $\sim$1,035 steps, while ULA plateaus at 2.8%. (b) ULA clusters near indices 400–500, whereas SR-ULA uniformly traverses the landscape.

EBM from Du & Mordatch (2019).[4] This model implicitly captures the distribution over CIFAR-10 images (50,000 training images, uniform across 10 classes).

**Setup.** A single chain runs for 2,000 Langevin steps, initialized from uniform noise $x_0 \sim \mathcal{U}(0,1)^{32 \times 32 \times 3}$. At each step, we record the sample and classify it using a pre-trained CIFAR-10 classifier. The *trajectory distribution*, i.e., the empirical class distribution over all 2,000 samples, reveals whether the chain explores broadly or remains trapped.

**Evaluation Metrics.** Since CIFAR-10 has a balanced class distribution (5,000 training images per class across $C = 10$ classes), ideal mode coverage corresponds to uniform visitation over classes. We therefore take the reference distribution to be the discrete uniform $u = (1/C, \ldots, 1/C) \in \Delta^{C-1}$ over $C = 10$ classes, and measure how closely the empirical class distribution $\hat{p} = (\hat{p}_1, \ldots, \hat{p}_C)$ matches $u$. Formally, let $\hat{p}_c = n_c/N$ where $n_c$ is the number of samples assigned to class $c$ and $N$ is the total number of samples. We report the following metrics:

- **KL Divergence** from $\hat{p}$ to $u$:

$$\mathrm{KL}(\hat{p} \,\|\, u) = \sum_{c=1}^{C} \hat{p}_c \log \frac{\hat{p}_c}{u_c} = \sum_{c=1}^{C} \hat{p}_c \log(C\hat{p}_c). \tag{89}$$

   This vanishes if and only if $\hat{p} = u$, and is lower-bounded by zero (lower is better).

- **Total Variation Distance** between $\hat{p}$ and $u$:

$$\mathrm{TV}(\hat{p}, u) = \frac{1}{2} \sum_{c=1}^{C} |\hat{p}_c - u_c| = \frac{1}{2} \sum_{c=1}^{C} \left| \hat{p}_c - \frac{1}{C} \right|. \tag{90}$$

   This equals zero under perfect uniformity (lower is better).

- **Normalized Entropy**:

$$H_{\mathrm{norm}}(\hat{p}) = \frac{H(\hat{p})}{\log C} = -\frac{1}{\log C} \sum_{c=1}^{C} \hat{p}_c \log \hat{p}_c, \tag{91}$$

   which is normalized to $[0, 1]$ and equals 1 if and only if $\hat{p} = u$ (higher is better).

Together, these three metrics provide complementary views of mode coverage: KL divergence is sensitive to underrepresented classes, TV distance captures the worst-case deviation, and normalized entropy summarizes the overall spread.

**Results.** As detailed in Table 3 and Figure 8, SR-ULA significantly outperforms ULA in single-chain exploration. While ULA suffers from severe mode collapse, i.e., predominantly oscillating between *airplane* and *horse* while missing four classes entirely, SR-ULA leverages score repellence to venture into unexplored regions. This history-based avoidance results

---

[4]Codebase link: https://github.com/openai/ebm_code_release

*Table 3.* Single-chain trajectory analysis: metrics are computed over all 2,000 intermediate samples visited by a single chain across time (i.e., the full temporal trajectory, not the final state).

| Metric | ULA | SR-ULA | Interpretation |
|---|---|---|---|
| Modes Covered | 5 / 10 | **8 / 10** | More classes visited |
| KL Divergence ↓ | 1.492 | **0.558** | Closer to uniform |
| TV Distance ↓ | 0.716 | **0.437** | More balanced |
| Norm. Entropy ↑ | 0.352 | **0.757** | Higher diversity |

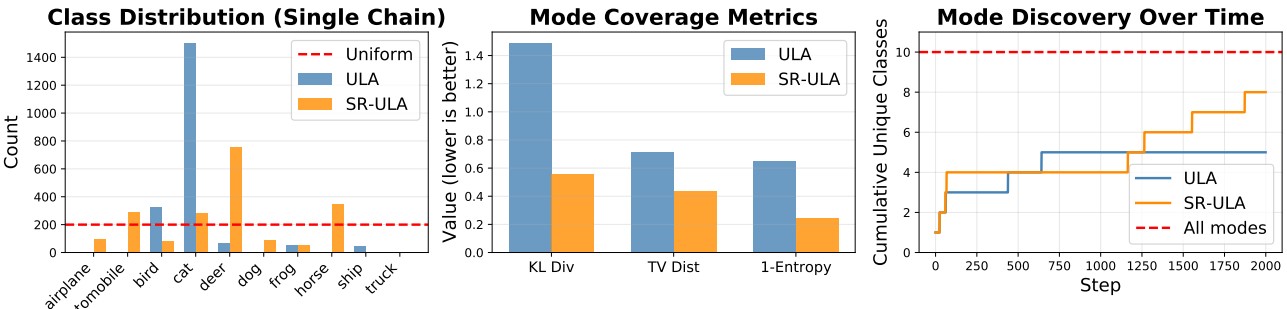

*Figure 8.* Single-chain trajectory analysis (2,000 steps). **Left:** Class distribution over trajectory. ULA collapses to *airplane* and *horse*; SR-ULA covers 7 classes. **Center:** Mode coverage metrics (lower is better). **Right:** Cumulative unique classes discovered over time.

in a 62.60% reduction in KL divergence (0.558 vs. 1.492) and 38.97% lower TV distance. Furthermore, SR-ULA discovers modes more rapidly, reaching 8 unique classes by step 1,000 compared to just 5 for ULA, demonstrating that the surrogate tilt $\pi_\theta$ from one's own score history effectively prevents local stagnation.

### D.2.3. EXPERIMENT 3: PARALLEL CHAINS (FINAL STATE DISTRIBUTION)

**Objective.** Practical MCMC often uses multiple parallel chains for parallelization and improved coverage. This experiment evaluates whether SR-ULA produces a more uniform distribution over semantic modes when examining the *final states* of many independent chains, and how this advantage changes as the number of available chains is reduced.

**Setup.** We use the same pre-trained CIFAR-10 EBM as in Experiment 2 and follow the same default sampling protocol of Du & Mordatch (2019). For each chain count

$$N_{\text{chain}} \in \{100, 50, 10\},$$

we run $N_{\text{chain}}$ independent chains for 450 Langevin steps each. Each chain is initialized from independent uniform noise, maintains its own score history with no shared state or inter-chain coupling, and retains only its final sample, simulating a practitioner collecting one sample per chain after burn-in. The $N_{\text{chain}}$ final samples are classified to obtain the empirical class distribution $\hat{p}$, providing $N_{\text{chain}}/10$ sample per class under ideal uniform coverage.

**Results.** Table 4 shows that SR-ULA consistently outperforms ULA across all chain counts. When sufficiently many chains are available, both methods already achieve broad class coverage, so the difference appears mainly in distributional balance: with 100 chains and 50 chains, SR-ULA reaches all 10 classes while ULA covers only 9, and SR-ULA also yields lower KL divergence, lower TV distance, and higher normalized entropy. The performance gap becomes substantially more pronounced in the restricted-parallelism regime. With only 10 chains, neither method fully covers all classes; however, SR-ULA still reaches 7/10 classes, whereas ULA covers only 5/10. Figure 9 shows the detailed class distributions sampled by SR-ULA and ULA and demonstrates that SR-ULA also improves all three balance metrics. In particular, KL divergence drops from 0.8318 to 0.4682, TV distance from 0.50 to 0.30, and normalized entropy rises from 0.6388 to 0.7967. Thus, the score-repellent mechanism is especially beneficial when the number of independent chains is limited, precisely because the baseline then has less opportunity to recover diversity through brute-force parallelization. These findings indicate that the repulsive mechanism effectively mitigates population-level mode collapse, promoting broader coverage and uniformity that are highly beneficial for diversity-oriented applications.

*Table 4.* Parallel-chain evaluation on final states only, with one retained sample per chain after 450 steps.

| Metric | 100 chains | | 50 chains | | 10 chains | |
|---|---|---|---|---|---|---|
| | ULA | SR-ULA | ULA | SR-ULA | ULA | SR-ULA |
| Modes Covered | 9/10 | **10/10** | 9/10 | **10/10** | 5/10 | **7/10** |
| KL Divergence ↓ | 0.3062 | **0.2021** | 0.3469 | **0.2257** | 0.8318 | **0.4682** |
| TV Distance ↓ | 0.3300 | **0.2600** | 0.3300 | **0.2800** | 0.5000 | **0.3000** |
| Norm. Entropy ↑ | 0.8670 | **0.9122** | 0.8494 | **0.9020** | 0.6388 | **0.7967** |

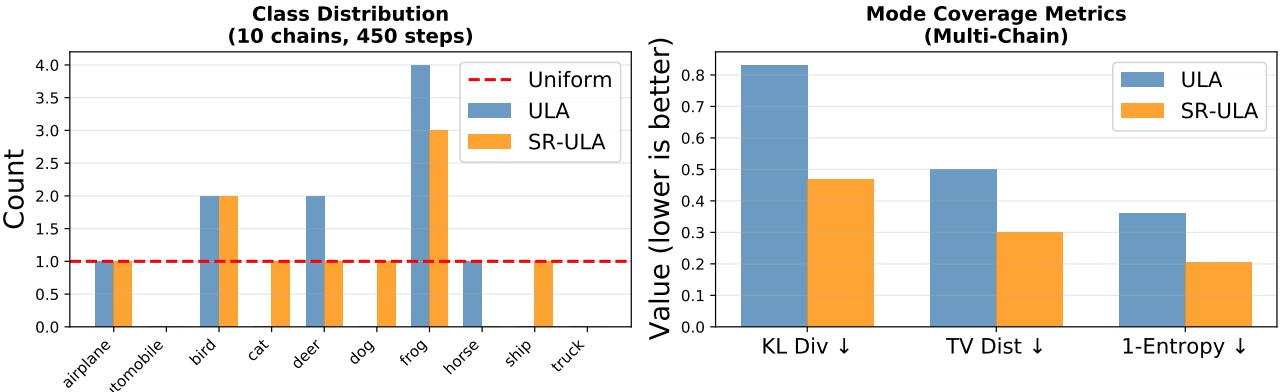

*Figure 9.* Multi-chain evaluation (10 chains × 450 steps). **Left:** Class distribution of final samples. SR-ULA covers more classes and is closer to uniform. **Right:** Mode coverage metrics (lower is better).

## D.3. Static MNIST: Qualitative Trajectories and AIS-Based Diversity

In this section, the state is a binarized MNIST image $x \in \{0, 1\}^{784}$ (i.e., $28 \times 28$ pixels) in a **discrete configuration state space** with $2^{784}$ possible states. We refer to this setting as 'static' MNIST to distinguish it from dynamic or sequential generative tasks; here the goal is to sample from a fixed energy-based model (EBM) $\pi(x) \propto \exp(-U_\phi(x))$ where $U_\phi$ is parameterized by a neural network.

We build directly on the official *Gibbs-With-Gradients* (GWG) release (Grathwohl et al., 2021)[5] and train the EBM from scratch using the GWG sampler. GWG is a gradient-informed discrete sampler that constructs locally balanced proposals using the score $s(x) = -\nabla_x U_\phi(x)$ evaluated at discrete configurations via continuous relaxation (see Appendix B.5.3 for algorithmic details). Our SR-MCMC implementation (SR-GWG) modifies only the sampler by introducing the score-repellent mechanism described in Algorithm 6; all model hyperparameters, network architecture, and training configurations remain identical to the original GWG release.

### D.3.1. Mode Definition and Diversity Diagnostics

**Digit-class modes.** We define coarse-grained modes via a pretrained MNIST classifier (a convolutional neural network achieving $> 99\%$ test accuracy) that maps each sample $x$ to one of 10 digit classes $\{0, 1, \ldots, 9\}$. This yields an interpretable proxy for mode coverage at the semantic level: if the sampler explores all digit classes uniformly, it suggests broad coverage of the EBM's probability mass. We emphasize that this classifier-based mode definition is a *diagnostic tool* for measuring diversity; the sampler itself has no access to the classifier or class labels.

**Cumulative KL divergence (mixing under worst-case initialization).** For the mode-mixing experiment, let $\{x_n^{(i)}\}_{i=1}^M$ denote the batch of samples across $M$ parallel chains at discrete time $n$. We define the cumulative class histogram by

---

[5]Codebase: https://github.com/wgrathwohl/GWG_release

aggregating predicted labels across all chains and all time steps up to $n$:

$$p_n(k) \; \propto \; \sum_{\tau=1}^{n} \sum_{i=1}^{M} \mathbf{1}\{\mathrm{cls}(x_\tau^{(i)}) = k\}, \qquad k \in \{0, \ldots, 9\}, \tag{92}$$

where $\mathrm{cls}(\cdot)$ denotes the classifier's predicted digit. We then report $\mathrm{KL}(p_n \| u)$ where $u = (0.1, \ldots, 0.1)$ is the uniform distribution over the 10 digits. Intuitively, $\mathrm{KL}(p_n \| u) = 0$ when the cumulative histogram is perfectly uniform, indicating that the sampler has visited all digit modes equally. Rapid decay of $\mathrm{KL}(p_n \| u)$ over time indicates that chains escape the initial mode quickly and explore other digits, while a persistently high KL indicates mode-trapping behavior.

**Batch Vendi Score (within-batch diversity).** At each time $n$, we compute the Batch Vendi Score (Friedman & Dieng, 2023) on the current batch $\{x_n^{(i)}\}_{i=1}^{M}$. The Vendi Score is defined as the exponential of the Shannon entropy of the eigenvalues of a similarity matrix $K$, where $K_{ij} = k(x_n^{(i)}, x_n^{(j)})$ for a chosen kernel $k(\cdot, \cdot)$. Intuitively, the Vendi Score measures the 'effective number of distinct samples' in the batch: it equals $M$ when all samples are mutually orthogonal (maximally diverse) and equals 1 when all samples are identical (complete collapse). We used the cosine similarity between the class probability vectors (from a pre-trained classifier) for the similarity matrix $K$. Higher Vendi Scores indicate reduced batch collapse, meaning chains are less likely to produce near-duplicate samples at any given time step.

### D.3.2. EXPERIMENT 1: MODE MIXING FROM SINGLE-DIGIT INITIALIZATION

**Objective.** We test whether SR-GWG helps the sampler escape a single digit mode under a strict worst-case initialization, where all chains start from the *same* image.

**Protocol.** We initialize $M = 100$ parallel chains from the *same* real image of digit '7' (chosen arbitrarily from the MNIST test set) and run $T = 10,000$ sampling steps. This initialization represents a challenging scenario: all chains begin at identical states deep within a single mode, and the sampler must overcome the energy barrier to transition to other digit modes. We compare the baseline GWG sampler against SR-GWG with repulsion strength $\alpha = 10^{-4}$. The history step size follows $\gamma_n = 0.1 \cdot (n+1)^{-0.6}$, consistent with the theoretical requirements in Section 3. We set $\alpha = 10^{-4}$ through preliminary experiments, which provides a sufficient amount of repellence and maintains a good approximation using the gradient proxy rather than the exact discrete score in (17) with huge computational costs. Larger values accelerate mixing but can introduce transient bias, while smaller values provide insufficient repulsion to escape the initial mode within the allotted iterations.

**Qualitative trajectories.** Figure 10 visualizes mixing by displaying 10 randomly selected chain states (out of 100) at checkpoints $n \in \{0, 2500, 5000, 7500, 10000\}$. Baseline GWG exhibits minimal variation: chains remain near the initialization or transition only to visually similar digits (e.g., '1' or '9', which share stroke patterns with '7'). In contrast, SR-GWG shows clear transitions to diverse digits (e.g., '0', '3', '5', '8') by $n = 2500$ and achieves broader coverage of all 10 digit classes by $n = 10000$. This qualitative difference reflects the score-repellent mechanism's ability to discourage chains from revisiting over-explored regions of the state space.

### D.3.3. EXPERIMENT 2: AIS-BASED DIVERSE GENERATION

**Objective.** Beyond mixing speed from a single mode, we evaluate whether SR-GWG improves the diversity of generated samples under Annealed Importance Sampling (AIS) (Neal, 2001), following the standard evaluation pipeline in the GWG release.

**Background on AIS.** AIS is a widely used method for both sampling and estimating normalizing constants of unnormalized distributions. It constructs a sequence of intermediate distributions $\pi_\beta(x) \propto \pi_0(x)^{1-\beta} \pi_1(x)^\beta$ for $\beta \in [0, 1]$, where $\pi_0$ is an easy-to-sample base distribution (e.g., uniform over $\{0, 1\}^{784}$) and $\pi_1 = \pi$ is the target EBM. Starting from samples drawn from $\pi_0$, AIS applies a sequence of MCMC transitions targeting each intermediate distribution, gradually annealing from $\pi_0$ to $\pi$. The quality of AIS samples depends critically on the mixing properties of the internal MCMC transitions: if transitions fail to mix well at intermediate temperatures, AIS samples may lack diversity.

**Protocol.** We run AIS with the same annealing schedule (number of intermediate distributions and $\beta$ spacing) and the same number of transition steps per intermediate distribution as in the original GWG experiments. The only modification

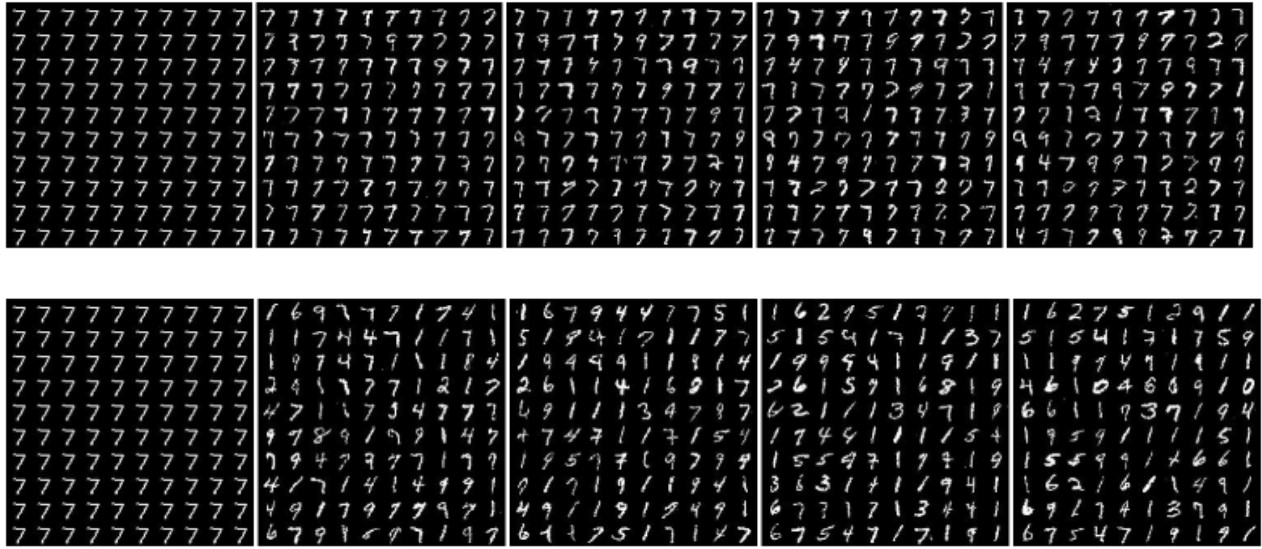

*Figure 10.* Static MNIST mode mixing ($M = 100$ chains, $T = 10{,}000$ steps, initialized at digit '7'). Each row shows 10 randomly selected chains at checkpoints $n \in \{0, 2500, 5000, 7500, 10000\}$. **Top:** Baseline GWG trajectories remain trapped near the initialization mode, with most samples resembling '7' or visually similar digits. **Bottom:** SR-GWG trajectories exhibit faster escape from the initial mode and broader coverage of diverse digit classes.

is the internal transition kernel: we compare baseline GWG transitions against SR-GWG-augmented transitions. For this diverse-generation task, we use a stronger repulsion strength $\alpha = 5 \times 10^{-4}$ compared to Experiment 1 in Appendix D.3.2. The rationale is that AIS operates across a range of temperatures, and stronger repulsion helps maintain diversity throughout the annealing process, whereas in Experiment 1 we sample from the target distribution directly and use milder repulsion to ensure asymptotic correctness. We generate $N = 2000$ samples using each method and compute diversity metrics on the resulting batches.

*Table 5.* Diversity metrics for AIS-based generation on binarized MNIST ($N = 2000$ samples).

| Method | Class Entropy ↑ | Vendi Score ↑ |
|---|---|---|
| GWG (baseline) | 2.1140 | 8.2572 |
| SR-GWG (ours) | **2.1628** | **8.4319** |

**Metrics.**   We report two complementary diversity metrics:

1. **Class entropy:** $H(\hat{p}) = -\sum_{k=0}^{9} \hat{p}(k) \log \hat{p}(k)$, where $\hat{p}(k)$ is the empirical frequency of digit class $k$ among the $N$ generated samples. Maximum entropy is $\log(10) \approx 2.303$ when all digits appear equally.

2. **Vendi Score:** Computed on the full batch of $N$ samples, measuring the effective number of distinct samples.

**Results.**   Table 5 summarizes the results. SR-GWG achieves higher values on both metrics—class entropy 2.1628 vs. 2.1140 (GWG), and Vendi Score 8.4319 vs. 8.2572 (GWG), which is consistent with broader and more uniform digit-mode coverage under the same AIS schedule and transition budget. The improvement in class entropy indicates more balanced representation across digit classes, while the improvement in Vendi Score reflects greater sample-level diversity beyond class labels.

*Remark* 15. The suitable range for $\alpha$ is primarily determined by the underlying domain (discrete versus continuous) and by how sensitively the relevant hyperparameters respond to the particular structure of the energy landscape. In the experiments with discrete variables, $\alpha$ plays the role of controlling the transition probability (i.e., the rate at which states are flipped). This differs markedly from the continuous domain, where a large step size may simply result in overshooting without

completely destroying the dynamics. In the discrete case, however, choosing a large alpha effectively instructs the sampler to attempt simultaneous updates across many variables at once. When this happens, the system enters a saturated regime in which the sampling dynamics degenerate into an almost pure random walk, rapidly erasing meaningful image structure. Consequently, $\alpha$ is suggested be set to a much smaller magnitude in discrete settings in order to preserve locality of moves and maintain stable, controlled updates.

