# OpenReview forum: "Score-Repellent Monte Carlo: Toward Efficient Non-Markovian Sampler with Constant Memory in General State Spaces"
_ICML.cc/2026/Conference — ICML 2026 spotlight_

### Official Review · Reviewer_mGWs · 2026-03-02

**Soundness:** 3
**Presentation:** 4
**Significance:** 4
**Originality:** 4
**Overall Recommendation:** 5
**Confidence:** 4

**Summary:**

The goal is to sample from a target distribution $\pi(x)$, known up to a normalizing constant and defined on a continuous or discrete space.

The paper uses any Annealed MCMC algorithm with a novel annealing path designed to encourage mode-exploration  (Eq 2). Standard annealing paths are typically interpolations (e.g. geometric mean) between an initial distribution and the target distribution: they are *independent* from the MCMC trajectory. Unlike this, the authors propose an annealing path that is *depend* on the MCMC trajectory: it is a tilted version of the target distribution $\pi(x)$ that lowers mass on regions of space that has been consistently visited and augments mass on unvisited areas of space.

**Compliance With Llm Reviewing Policy:**

Affirmed.

**Final Justification:**

This is a very good paper. I didn't have main concerns to start with, just questions. I see the authors added material for the discrete case, which additionally strengthens the paper.

**Key Questions For Authors:**

**Q1. Sufficient vs. necessary condition for convergence**

In the beginning of section 2, the authors present a necessary criterion for convergence $\mathbb{E}_{x \sim p}[s(X)] \leftarrow p = \pi$.  Later in line 328, the authors present it as a sufficient criterion for convergence. Is it both necessary and sufficient?

**Q2. Interpretation of Assumption 1-ii.**

Can the authors provide an interpretation of assumption 1-ii? It seems to say: the Hessian of the energy function decays at a certain rate in the tails and something more.

**Q3. Interpretation of Assumption 2's uniform drift condition (Eq 31)**

Can the authors provide a rough interpretation of that condition?

**Q4. Linking the convergence rate to Eq 12?**

To get an interpretable convergence rate, I can find $N$ such that $\forall \epsilon > 0, n > N \rightarrow \gamma_n Tr(\Sigma_{\upsilon}) < \epsilon$. The resulting $N$ would depend on the error tolerance $\epsilon$, the step size $\gamma_n$, the mode-repellence strength $\alpha$, and the geometry of the target distribution via $\Sigma_{\upsilon}$. Do the authors envisage a way to make the between-mode distance $R$ appear in  $\Sigma_{\upsilon}$, for example for Gaussian mixtures? This way, $N$ could have an expression like $N = \exp(R^2) / alpha$ that would indicate how to choose the repellence strength $\alpha$ to compensate for the between-mode distance $R$.

**Q5. Algorithmic stability**

The annealing path (Eq 2) depends on the MCMC chain, which itself depends on the annealing path. One could imagine running into instabilities: do the authors encounter this in practice?

**Q6. Conditions on the step size for convergence**

Eq 12 suggests that to convergence, the step size $\gamma_n$ should decrease to zero. Is there any condition on how quickly it should decrease, for convergence?

**Limitations:**

Yes

**Strengths And Weaknesses:**

**Strengths**: the paper is very clearly written. It clearly distinguishes the novelty of the analysis compared to previous works which are typically in discrete spaces only and store all previously visited states instead of a running average of score evaluations. Sampling from multi-modal distributions is a notoriously hard in theory and practice. The authors have convincing proof-of-concepts of their method (Figure 1) very well-designed and pedagogical and realistic experiments (Appendix D.2., CIFAR Gaussian Mixture Model and then Static MNIST) that go beyond 2D mixtures of Gaussians and Checkerboard experiments that are common in the sampling literature. They also have a clear theory of convergence under pretty weak assumptions on the target distribution; they do not assume log-concavity, for instance. They study special cases where the convergence rate is more explicit and interpretable (section 3). As far as I am concerned, the theory and experiments are very convincing and the paper is very well-written and addresses a hard problem. Even the concluding remarks are very pertinent: given that the authors show that the asymptotic mixing time improves as $\alpha$ increases, I was wondering what were the drawbacks of choosing a too large $\alpha$ --- the authors address this in the conclusion. I was also wondering on heuristics for tuning $\alpha$ on-the-fly which the authors similarly mention in the conclusion.

**Weaknesses**: none in particular. I have some questions: see questions section below.

---

> ### Author Rebuttal · Authors · 2026-03-30
>
> We thank the reviewer for their insightful feedback and the technical questions.
>
> *To support our rebuttal, we have provided the paper revision in the anonymous link attached in the response to Reviewer WBGY.*
>
> ---
>
> ### Q1
>
> We do not use $\mathbb{E}\_{p}[s(X)]=0$ as a sufficient condition to derive $p=\pi$, and this is not what line 328 is claiming. That line refers to a consequence of Theorem 3.3: As the theorem has already shown that the coupled iterate converges a.s. to $(0,\mu)$, we have $\theta\_n\to 0$ a.s. Because the surrogate family is parameterized only by $\theta$ and $\pi\_0=\pi$, this implies $\pi\_{\theta\_n}\to\pi$ a.s. So the role of $\mathbb{E}\_{p}[s(X)]=0$ is only to identify the correct equilibrium at $\theta^\star=0$ through $\mathbb{E}\_{\pi}[s(X)]=0$; the actual convergence is proved by Theorem 3.3, not by a sufficiency claim of the form $\mathbb{E}\_{p}[s(X)]=0 \rightarrow p=\pi$.
>
> ---
>
> ### Q2
>
> Assumption 1(ii) is a tail regularity condition on the curvature of the energy. Along each ray $x=r\hat x$, the Hessian has a well-defined leading-order behavior after rescaling by $r^{-(p-2)}$, converging to a positive-definite limit instead of oscillating irregularly. For $p=2$, this is the familiar quadratic-tail case; for $p<2$, the curvature may decay, but only in a controlled way. This is mainly used to ensure that the score-tilted surrogate $\pi\_\theta$ is well-defined and that the SA iterates remain bounded via the ODE@$\infty$ argument in [2].
>
> ---
>
> ### Q3
>
> Roughly, the uniform drift condition says that for every surrogate kernel $P\_\theta$, the chain is pulled back toward a stable region at a rate uniform in $\theta$, so the adaptive surrogate cannot create unstable tail behavior. Even though the target changes online through $\theta\_n$, the family $\{P\_\theta\}$ remains uniformly geometrically ergodic, which keeps the Markovian noise controlled.
>
> ---
>
> ### Q4
>
> Thank you for the insightful question. Focusing on $\theta\_n$, Eq. (12) suggests
> $$
> N\_{\varepsilon}(\alpha)\asymp \left(\frac{\text{Tr}(\Sigma\_{\theta\theta}(\alpha))}{\varepsilon}\right)^{1/\rho},
> $$
> ignoring pre-asymptotic mixing effects. From Line 1738, we have
> $$
> \text{Tr}(\Sigma\_{\theta\theta}(\alpha)) = \frac{1}{2\alpha}\sum\_i \frac{\widetilde{U}\_{ii}(R)}{\lambda\_i(R)} + O(\alpha^{-2}).
> $$
> For a well-separated modes, $\lambda_i(R) = O(1)$ because the marginal score covariance is dominated by the local variance of each mode. To link the long-run covariance $\widetilde U(R)$ to $R$, we heuristically relate $\widetilde U(R)$ to the base chain's mixing time $\tau_{mix}$, i.e., slow-mixing chain has a larger long-run covariance (note that these two concepts are correlated but not always hold true). Since metastability analysis (e.g., [1] Section 4.1.2) shows $\tau_{mix} \sim e\^{R\^2}$ for MH sampler, we expect $\widetilde U(R) \sim e\^{R\^2}$. This yields the intuition you suggested:
> $$
> N\_{\varepsilon}(\alpha,R)=O \left(\left[\frac{e\^{cR\^2}}{\alpha\varepsilon}\right]^{1/\rho}\right).
> $$
> We emphasize this is a heuristic connection: [1] analyzes ordinary Markov chains, not the nonlinear history-adapted SRMC. Making the $R$-dependence rigorous for SRMC would require a new spectral-gap analysis for the surrogate-kernel family $\{P\_\theta\}$, which is beyond the scope of the present paper.
>
> ---
>
> ### Q5
>
> Our algorithm is not a simultaneous cyclic definition. At time $n$, $\theta\_n$ is based on all history (running average of score over the trajectory till step $n$); we define the surrogate $\pi\_{\theta\_n}$, run one MCMC step targeting this surrogate to obtain $X\_{n+1}$, and then update $\theta\_{n+1}$. Thus, the procedure is sequential and well-defined, exactly as a stochastic approximation with state-dependent Markovian noise (cf. [2]). The real issue is stability of this adaptive recursion, which is what our analysis addresses: under Assumptions 1-2, Theorem 3.3 proves almost sure convergence of the coupled iterate to $(0,\mu)$. In practice, we did not observe divergence; larger $\alpha$ (larger repellence strength) could cause a longer calibration phase and transient oscillations.
>
> ---
>
> ### Q6
>
> To prove convergence in theory, $\gamma\_n$ should satisfy the Robbins-Monro conditions
> $$
> \sum\_n \gamma\_n=\infty,\qquad \sum\_n \gamma\_n^2<\infty,
> $$
> and $\lim\_{n\to\infty}(\gamma\_{n+1}^{-1}-\gamma\_n^{-1})$ should exist and be finite. In the paper we follow the standard choice
> $$
> \gamma\_n=(n+1)^{-\rho},\qquad \rho\in(1/2,1].
> $$
>
> ---
> [1]. Woodard, Dawn B., Scott C. Schmidler, and Huber, M. Conditions for Rapid Mixing of Parallel and Simulated Tempering on Multimodal Distributions. The Annals of Applied Probability, 19(2):617-640, 2009.
>
> [2]. Borkar, V., Chen, S., Devraj, A., Kontoyiannis, I., and Meyn, S. The ode method for asymptotic statistics in stochastic approximation and reinforcement learning. The Annals of Applied Probability, 35(2):936–982, 2025.

---

> > ### Author Rebuttal · Reviewer_mGWs · 2026-04-02
> >
> > This is a very good paper. I didn't have main concerns to start with, just questions. I see the authors added material for the discrete case, which additionally strengthens the paper.

---

### Official Review · Reviewer_KmAj · 2026-03-09

**Soundness:** 3
**Presentation:** 3
**Significance:** 3
**Originality:** 3
**Overall Recommendation:** 5
**Confidence:** 3

**Summary:**

This paper proposes Score-Repellent Monte Carlo (SRMC), a non-Markovian MCMC wrapper that compresses trajectory history into a d-dimensional running average of score evaluations and uses that history to define a normalization-free score-tilted surrogate target. The framework is presented as broadly compatible with standard samplers, and the paper backs this up with a stochastic-approximation analysis establishing almost sure convergence and a joint CLT. Empirically, the paper evaluates SRMC on continuous targets with gradient-based samplers and on discrete energy-based models, reporting improved variance reduction and better mode exploration than baseline samplers

**Compliance With Llm Reviewing Policy:**

Affirmed.

**Final Justification:**

All questions have been solved.

**Key Questions For Authors:**

1. The paper mentions that an adaptive α schedule is a reliable default across target geometries, but I could not find a clear specification of that schedule in the main paper. What exactly is the adaptive rule, where is it used, and how should a practitioner choose α in settings not covered by the experiments? A clear answer would improve my confidence in the method’s practical usability.

2. Can the authors quantify the wall-clock overhead for the continuous gradient-based variants, not just the number of gradient evaluations? In particular, how expensive were the Hessian-vector products or finite-difference approximations in SR-MALA / SR-HMC, and what does the error-versus-time tradeoff look like?

3. Could the authors clarify whether Appendix D.2 uses exactly the same SRMC method as Algorithm 1, or an extension of it? In particular, what role do “bandwidth c,” “inter-chain repulsion,” and the SR-MCMC / SR-LD terminology play relative to the main method? This matters for reproducibility and for interpreting the larger-scale empirical evidence.

4. The current continuous experiments are promising but still modest in scale. Could the authors add, or at least discuss, one harder continuous benchmark or stronger finite-time analysis to better characterize when SRMC helps most?
A more demanding validation would likely improve my assessment of the paper’s overall impact.

**Limitations:**

yes

**Strengths And Weaknesses:**

Strength:
1. The paper’s main idea is genuinely novel and elegant. The authors replace explicit history tracking by a d-dimensional running average of score evaluations, and use this to build a normalization-free score-tilted surrogate that can wrap a broad class of MCMC kernels. That is a nice conceptual bridge between history-dependent sampling and practical samplers on general state spaces.
2. The paper has a meaningful theoretical contribution. It does not just provide intuition, but develops a stochastic-approximation analysis for the coupled sampler/history dynamics, proving almost sure convergence and a joint CLT. I also appreciate that the paper is fairly careful about the scope of its stronger variance claims: the clean O(1/α) behavior is stated for special cases rather than universally.
3.The empirical section is reasonably broad for a first paper on a new MCMC wrapper. The authors evaluate continuous variants on correlated Gaussian and Bayesian logistic regression targets, using 100 independent runs and matching total gradient evaluations, and they also test discrete mode exploration on Static MNIST. The reported gains are meaningful, including lower MSE in the continuous setting and large improvements in KL/Vendi Score in the discrete setting.
4.The presentation is solid overall. The motivation is easy to follow, the contrast with empirical-measure-based history tracking is clear, and the metastability illustration gives an intuitive picture for why the score-repellent mechanism can help with barrier crossing.

Weakness:
1. My main technical reservation is that the strongest efficiency claims are still somewhat narrow. The theory clearly supports asymptotic correctness and some special-case covariance improvements, but for general targets and general base kernels the paper itself notes that explicit estimator-variance characterizations are unavailable. So I would view the theory as strong, but not yet a full general explanation of when SRMC will provably improve Monte Carlo efficiency.
2. The practical-overhead story is convincing in the discrete gradient-informed setting, but less so for the continuous gradient-based instantiations. For SR-ULA/SR-MALA and related methods, the surrogate score requires Hessian-vector products or finite-difference approximations, which may be nontrivial in practice. By contrast, the paper explicitly argues that discrete gradient-informed samplers often incur little or no extra score-evaluation cost. I therefore think the “minimal overhead” claim should be calibrated more carefully for continuous variants.
3. The continuous empirical evaluation is encouraging but still somewhat limited. The main quantitative experiments are on two 10-dimensional benchmarks, which are useful but not especially demanding by modern standards. One additional harder continuous benchmark or stronger finite-time evidence would make the practical case more compelling.

---

> ### Author Rebuttal · Authors · 2026-03-30
>
> We thank the reviewer for the detailed feedback, and for the constructive questions concerning theoretical scope, computational overhead, and empirical validation.
>
> *To support our rebuttal, we have provided the paper revision in the anonymous link attached in the response to Reviewer WBGY.*
>
> ---
>
> ### W1
>
> Please refer to the response to Reviewer WBGY (Q2).
>
> ---
>
> ### Q1
>
> Please refer to the response to Reviewer kXsQ (Q2).
>
> ---
>
> ### W2 & Q2
>
> We agree that gradient-based SRMC should also be evaluated under computational budget. We therefore added a **new** Appendix C.11, which formalizes a cost-based CLT for gradient-based SRMC: under the finite-difference implementation, one baseline iteration has cost d, while one SRMC iteration has cost 3d because it uses one base gradient, one extra gradient for the Hessian-vector approximation, and one d-dimensional SA update. Appendix C.11 then translates the time-indexed CLT into a budget-indexed CLT, showing that under budget B, the asymptotic covariance is simply rescaled by the asymptotic cost per iteration. The key point is that the extra constant-factor cost from the overhead does not invalidate the storyline: whenever the reduction in $\Sigma\_{\mu\mu}(\alpha)$ is strong enough, SRMC still wins at matched budget. This becomes explicit in the correlated Gaussian case, where $\Sigma\_X(\alpha) = V\Sigma\_{\theta\theta}(\alpha)V^T = O(1/\alpha)$, so even after accounting for the 3d cost factor, the budget-normalized covariance remains $O(d/\alpha)$. We also added CPU-time plots in the **revised** Figure 2, which show the expected smaller but still clear advantage of SRMC over the baseline when wall-clock time is used as the x-axis, and is expected by our theory.
>
> ---
>
> ### Q3
>
> We have improved Appendix D.2 to remove the notation inconsistency. In particular,
> - Appendix D.2 uses the same SRMC framework as Algorithm 1;
> - 'bandwidth c' has been replaced by the finite-difference approximation $\epsilon$ to match Eq. (6);
> - 'SR-MCMC' / 'SR-LD' are unified with the main-text terminology 'SRMC' / 'SR-ULA';
> - 'inter-chain repulsion' is corrected to 'intra-chain repulsion', since the multiple chains in that appendix are independent and do not interact.
>
> ---
>
> ### W3 & Q4
>
> We would like to clarify that Appendix D.2 of our submission already contains substantially more realistic high-dimensional targets than the two main continuous benchmarks, including a 1000-mode Gaussian mixture in $\mathbb{R}^{3\times 32 \times 32}$ and a pre-trained CIFAR-10 energy-based model. In particular, SR-ULA achieves complete coverage of all 1000 modes in the Gaussian-mixture experiment in about 1035 steps, whereas baseline ULA covers only a tiny fraction, and it also improves single-chain and multi-chain mode coverage on the CIFAR-10 EBM benchmark, where we further strengthen the results in the **revised** Appendix D.2.3 by comparing the performance versus the number of independent chains, and our SRMC shows more performance gain for reduced number of chains.
>
> Existing finite-time SA results with Markovian noise do not cover our SRMC setting. Available bounds typically assume a single fixed ergodic chain or a family of chains through one mixing-time parameter [1 - 3]. In contrast, SRMC generates parameter-dependent kernels $P\_{\theta}$, so the noise process is itself controlled by the evolving iterate. Our analysis must therefore work with the whole family {$P\_{\theta}$}, verify uniform drift / DV3 and Lipschitz regularity, and prove boundedness through ODE@$\infty$, which is why we rely on the asymptotic controlled-Markov SA framework of [4]. Extending finite-time SA theory to this setting would require new uniform control of the family {$\tau\_{\theta}$} as well as the closed-loop dependence between $\theta\_n$ and the sampling kernel, which is a good theoretical extension but beyond the scope of this paper.
>
> ---
>
> [1]. Sun, T., Sun, Y., & Yin, W. On markov chain gradient descent. Advances in neural information processing systems, 31, 2018.
>
> [2]. Doan, Thinh T. Finite-time analysis of markov gradient descent. IEEE Transactions on Automatic Control 68(4): 2140-2153, 2022.
>
> [3]. Even, M. Stochastic gradient descent under markovian sampling schemes. International Conference on Machine Learning, 2023.
>
> [4]. Borkar, V., Chen, S., Devraj, A., Kontoyiannis, I., and Meyn, S. The ode method for asymptotic statistics in stochastic approximation and reinforcement learning. The Annals of Applied Probability, 35(2):936–982, 2025.

---

> > ### Author Rebuttal · Reviewer_KmAj · 2026-04-01
> >
> > The rebuttal fully resolves my concerns and answers my questions. I have no further issue.

---

### Official Review · Reviewer_kXsQ · 2026-03-10

**Soundness:** 3
**Presentation:** 4
**Significance:** 4
**Originality:** 3
**Overall Recommendation:** 5
**Confidence:** 2

**Summary:**

This paper presents a plug-in adjustment to MCMC samplers designed to improve exploration allowing the Markov chain to more easily escape wells. To achieve this, the authors endow the algorithm with a memory vector that is used to repel the sampler away from frequently revisited regions. An obvious choice is to remember how often each previous state has been visited and lower the probability of revisits. However, for samplers in continuous space this strategy is clearly ill-defined. The authors replace aversion to commonly seen states to repulsion from a weighted moving average of the score, so that states with score vectors that point away from the average score are more likely. This is intuitively motivated by the simple identity that the expectation of the score under the target is zero. The modification to the target is designed in such a way that it can be used with a wide variety of base samplers. The authors state and prove asymptotic convergence results for their algorithm. They demonstrate performance on a set of challenging experiments.

**Compliance With Llm Reviewing Policy:**

Affirmed.

**Final Justification:**

I recommend the paper for acceptance for the reasons detailed in my review and rebuttal acknowledgement.

**Key Questions For Authors:**

1.	The achieved experimental results appear to have required somewhat careful tuning of $\alpha$ and you mention that a possible direction for future work is towards tuning algorithms for $\alpha$. What were the methods used to tune $\alpha$ and do these represent a significant computational burden to using this method?

2.	Relatedly, maybe there’s something I’m misunderstanding, but in the second paragraph of page 8 you mention an “adaptive $\alpha$ schedule” but I can’t find any other mention of it in the paper nor appendices. What is being referred to here?

**Limitations:**

Yes

**Strengths And Weaknesses:**

Strengths:

1. This is a good paper. It presents a simple to understand and implement idea that has broad applicability to MCMC.

2. The clarity is impressive given the high level of technicality of the subject. I have no concerns with respect to correctness or soundness of the algorithm or mathematical content. The experiments are comprehensive and cover a set of challenging settings where prior approaches fail or perform poorly. The experimental results are convincing. I have not seem similar ideas presented elsewhere. I believe this method can find widespread usage.

Weaknesses:

My criticisms are few and weak in nature.

1.  In the abstract $\alpha$  and $d$ and are not precisely defined so it can be a little confusing to the reader what exactly is being claimed.

2. The performance claims in the final contribution bullet point appear unconvincing, giving best case results makes it unclear if these results were only achieved on a tailored setup or if they are representative of what one can practically expect.

3. The experiments focus on cases that are challenging for prior algorithms, which is the most important case. However, it is yet unclear whether the proposed algorithm represents a strict upgrade to the baseline approaches. It would be slightly more complete, and of interest to practitioners, if the presented algorithm achieves competitive results in familiar cases where existing algorithms also perform well.

---

> ### Author Rebuttal · Authors · 2026-03-31
>
> We thank Reviewer kXsQ for the positive and encouraging evaluation, and for the thoughtful questions regarding parameter tuning and practical deployment.
>
> *To support our rebuttal, we have provided the paper revision in the anonymous link attached in the response to Reviewer WBGY.*
>
> ---
>
> ### W1
>
> We have clarified what $\alpha$ and $d$ mean in the abstract in the revision.
>
> ---
>
> ### W2
>
> Please refer to
> - response to Reviewer KmAj (W3 & Q4) for the discussion about the practical high-dimensional experiment in Appendix D.2;
> - response to Reviewer WBGY (W1 & W2 & W4) the effect of hyperparameters $\epsilon, \alpha, \rho$ to the performance gain of our SRMC framework.
>
> ---
>
> ### W3
>
> We agree that this comparison is important. The Bayesian logistic regression experiment already provides such a “familiar-case” check: baseline MALA and HMC are both well-tuned and perform well, yet SR-MALA and SR-HMC still achieve consistently lower MSE, with $\alpha = 1$ clearly outperforming the baseline across the plotted horizon. Since $\alpha = 0$ exactly recovers the base sampler, SRMC is best viewed as a strict extension of the baseline: practitioners can always fall back to the baseline, while our **new tuning guidance in Section 4.1** in the revision identifies a stable range of positive $\alpha$ that yields additional gain in practice.
>
> ---
>
> ### Q1
>
> The $\alpha$ value should be chosen based on the target distribution. The practical goal is to keep the exponent scale $\alpha |\theta\^\top s(x)|$ remains moderate along the trajectory. If the magnitude changes too abruptly, we could temporarily set $\alpha=0$ to just disable the repellence and follow the baseline. This avoids over-tilting the surrogate through the factor $e^{-\alpha \theta\_n\^\top s(x)}$, which can otherwise excessively amplify or suppress $\pi(x)$. Thus, the tuning burden is modest: one can start from a small grid of positive $\alpha$ values on top of the already tuned baseline sampler, select the largest stable value, and fall back to the baseline if needed.
>
> ---
>
> ### Q2
>
> We meant to imply that an adaptive $\alpha\_n$ scheme with small initial $\alpha_0$ value and increasing in time $n$ would suppress the early-phase oscillation in the transient phase, while enjoying benefits of large repellence strength $\alpha$ in the long run. This has also been mentioend in Section 5 future directions in our submission. To accommodate reviewer's comment, we also implement the preliminary adaptive-$\alpha$ scheme in the **new** Appendix D.1.4 in the revision, but only as a practical heuristic, not as part of the main theory. The simplest version starts from a small initial $\alpha_0$, increases during warmup toward a nominal reference value, and then freezes after warmup so that the averaging regime uses a stable working $\alpha$. We also tested a more robust exponent-scale guardrail, which freezes earlier when the rolling 95\% quantile of $\alpha\_n|\theta\_n\^\top s(x)|$ becomes too large. These two adaptive-$\alpha$ schemes do not uniformly beat the hindsight-best fixed $\alpha$, but they provide **robustness** and substantially reduce sensitivity to the nominal choice, especially in aggressive regimes. For example, in the tuned HMC + Bayesian logistic experiment in the Appendix D.1.4, at 100k steps, both $\alpha = 2$ and $5$ are considered large $\alpha$ and introduce strong tilting that oscillates the sampler and deteriorates the MSE to $1.55\times 10^{-3}$ and $1.94\times 10^{-2}$, whereas the corresponding capped adaptive runs reduce these to $1.12\times 10^{-5}$ and $2.53\times 10^{-3}$. This illustrates the main role of the adaptive rule: when the nominal $\alpha$ is overly aggressive, especially at $\alpha = 5$, the fixed-$\alpha$ sampler suffers from severe transient over-tilting, and that damage still persists at 100k steps. By capping the warmup and freezing at a safer working value, the adaptive scheme substantially mitigates this transient instability.

---

> > ### Author Rebuttal · Reviewer_kXsQ · 2026-04-01
> >
> > The rebuttal fully resolves my concerns and answers my questions. I have no further issue.

---

### Official Review · Reviewer_WBGY · 2026-03-12

**Soundness:** 3
**Presentation:** 3
**Significance:** 3
**Originality:** 3
**Overall Recommendation:** 5
**Confidence:** 2

**Summary:**

The paper proposes Score-Repellent Monte Carlo (SRMC), a history-dependent MCMC framework that stores past exploration through a running average of score evaluations and uses this history to define a score-tilted surrogate target. The method is presented as a generic, normalization-free wrapper around standard MCMC kernels. The paper provides a Stochastic-Approximation (SA) analysis with almost sure convergence and a joint CLT, together with special-case asymptotic variance improvements as the repulsion strength increases. Experiments on continuous targets and discrete EBMs suggest gains in mean-estimation error and in mode exploration.

**Compliance With Llm Reviewing Policy:**

Affirmed.

**Final Justification:**

I recommend acceptance of the paper. Please refer to my review for a detailed account of my position in terms of strengths. In terms of weaknesses, the rebuttal has substantially addressed my concerns in terms of scope, clarity of claims, and practical validation.

**Key Questions For Authors:**

- Could the authors clarify the exact scope of the theory, especially relative to the discrete experiments? My current understanding is that the main analysis is developed for continuous-state with assumptions on the score field and the kernel family, and if the discrete case uses score proxies from continuous relaxations then results port. A clearer statement would help me here.

- Could the authors better explain how the strong asymptotic result for the history variable translates, in practice, to improved estimator variance beyond the special cases discussed in the paper? I ask because, as written, the most explicit $O(1 / \alpha)$ result seems to concern the score-history covariance, while the estimator-side story is less clear in general.

- Could the authors provide a clearer analysis of computational cost for SR-MALA / SR-HMC? Since the method requires Hessian-vector product or multiple evaluation, it would be good to include comparison at matched computational budget.

- Could the authors discuss more explicitly the limitations of summarizing history through a single running score average? I am unsure when this summary might be too compressed.

**Limitations:**

yes

**Strengths And Weaknesses:**

My impression is that the paper contains a genuinely interesting idea, enough theoretical substance to make it more than a heuristic proposal, and convincing empirical results, even if some parts of the practical and theoretical story remain narrower than the framing might suggest.

# Strengths

- The core idea is genuinely interesting: replacing empirical-measure history by score-history is a nontrivial compression that makes history-dependent sampling plausible in continuous and large discrete spaces, preserving constant memory.

- The theory is substantial. The paper analyzes the coupled evolution of the history variable and estimator through stochastic approximation, proving almost sure convergence and a joint CLT rather than giving only heuristic intuition.

- The paper proposes a meaningful asymptotic result: increasing repulsion improves contraction in the history dynamics, and explicit  $O(1/\alpha))$ variance scaling is obtained in special cases.

- The resulting method is a wrapper that can be placed on top of Hamiltonian Monte-Carlo or Langevin type samplers.

- The experiments provide meaningful evidence, especially the static MNIST mode-mixing result, and yields consistent MSE improvements over baselines.

# Weaknesses

My reservations are mostly about scope, clarity of claims, and practical validation.

- Empirically, the paper is encouraging but not fully decisive. The discrete Static MNIST result is quite striking, and the continuous experiments do show consistent MSE improvements over baselines MALA and HMC. But some continuous benchmarks are still fairly small and classical, e.g., a 10D correlated Gaussian and a synthetic logistic regression posterior with $D=10,N=100$. That makes it hard to judge how robust / computationally feasible the method is in more realistic high-dimensional posterior settings, especially once one accounts for Hessian-vector products, finite-difference approximations, or tuning/sensitivity to $\alpha$ and $\rho$.

- The paper itself notes that large $\alpha$ can be counterproductive at finite horizons because of a longer calibration phase, and the plots also show some transient oscillations. I appreciated that the authors mention this, but I think the current version still leaves tuning and finite-time robustness somewhat under-explored.

- Finally, while I found the overall story understandable, some claims about generality felt slightly ahead of the evidence. In particular, the theory is developed in a continuous-state framework with explicit assumptions on the score and kernel family, while the discrete experiments rely on score proxies from continuous relaxations. I would have liked the paper to be slightly more explicit about how much of the theoretical interpretation is meant to carry over to that regime.

- The presentation has some inconsistencies. Appendix D.2 switches to "SR-MCMC", introduces a bandwidth parameter $c$ and discusses "inter-chain repulsion" (D.2.2), which seems misaligned with the main SRMC discussion.

---

> ### Author Rebuttal · Authors · 2026-03-31
>
> We thank Reviewer WBGY for the detailed feedback.
>
> *To support our rebuttal, we have provided the paper revision in the anonymous link: https://www.dropbox.com/scl/fi/0vc6cq6n1jjnzkk307qcq/ICML-2026-Rebuttal-Revision.pdf?rlkey=0zp7hfyqiezhjlswf742zq9v0&st=keab2oox&dl=0*
>
> ---
>
> ### W1 & W2 & W4
>
> For the concern about realistic high-dimensional targets, please refer to our response to Reviewer KmAj (W3 & Q4).
>
> We also revised Appendix D.2 for clarity and consistency: we unified the terminology under SRMC, changed the finite-difference notation from $c$ to $\epsilon$, and corrected 'inter-chain repulsion' to 'intra-chain repulsion'. In addition, new Section 4.1 now gives practical tuning guidance. The main message is: SRMC is fairly robust to a broad range of $\epsilon$; empirically $\rho$ around $0.6,0.8$ give the best transient behavior, while $\rho=1$ shrinks the SA step too quickly; and $\alpha$ should be chosen so that the exponent scale $\alpha |\theta^\top s(x)|$ stays moderate, avoiding over-tilting through $e^{-\alpha\theta\_n^\top s(x)}$, which can otherwise excessively amplify or suppress $\pi(x)$. For the discussion about adaptive-$\alpha$ scheme, please refer to the response to Reviewer kXsQ (Q2).
>
> ---
>
> ### W3 & Q1
>
> We agree that the original submission did not state this scope sharply enough due to limited time. Our main SA theory is written for general state spaces, but the continuous-domain verification is carried out under Assumptions 1–2. In the revision, we add an explicit analysis on discrete configuration spaces: SRMC should use a discrete score satisfying $\mathbb{E}\_{\pi}[s(X)]=0$, and **new** Proposition 3.6 in the revision states that, under this condition and the kernel assumption, Theorem 3.3 carries over directly to discrete configuration spaces. The revision also makes clear that the exact discrete score in Eq. (17) in the revised paper requires intensive computational costs, so the continuous-relaxation gradient used in the original Static MNIST experiments is a practical score proxy for constructing efficient samplers with minimal computational overhead. This clearly separates two claims: the new Appendix C.12 complements the theoretical gap in discrete configuration spaces, while the original MNIST experiment remains a computationally efficient empirical proxy that still demonstrates strong practical gains.
>
> ---
>
> ### Q2
>
> From Line 1783,
> $$Z(\alpha) = U\_{\mu\mu} - \alpha [C\Sigma\_{\theta\mu}(\alpha) + \Sigma\_{\theta\mu}(\alpha)\^\top C\^\top],$$
> where there was a sign typo in the original submission, and from Line 1790,
> $$
> \Sigma\_{\mu\mu}(\alpha) = \frac{1}{2\beta} \left(U\_{\mu\mu} - \alpha [C\Sigma\_{\theta\mu}(\alpha) + \Sigma\_{\theta\mu}(\alpha)\^\top C\^\top]\right) .\qquad (1)
> $$
> Here, $C = \text{Cov}\_{\pi}(f,s)$ is the marginal covariance between the test function $f$ and the score $s$, and $\alpha = 0$ recovers the baseline covariance $ U\_{\mu\mu} / 2\beta$.
>
> We do not claim a general monotonicity theorem, but (1) gives a clear mechanism: SRMC affects estimator variance only through the estimator/history coupling $C\Sigma\_{\theta\mu}(\alpha)$. When this coupling is aligned, the symmetric term
> $$
> C\Sigma\_{\theta\mu}(\alpha) + \Sigma\_{\theta\mu}(\alpha)\^\top C\^\top
> $$
> is expected to be positive semidefinite, so the subtraction in (1) lowers $\Sigma\_{\mu\mu}(\alpha)$. This is consistent with our proven order of the $\theta$ block, $\Sigma\_{\theta\theta}(\alpha) = O(1/\alpha)$, suggesting $\Sigma\_{\theta\mu}(\alpha)$ also shrinks on the same scale, making the correction term $\alpha [C\Sigma\_{\theta\mu}(\alpha) + \Sigma\_{\theta\mu}(\alpha)\^\top C\^\top)$ on the order $O(\frac{\alpha}{\alpha+c})$ and thus (1) is likely to decrease in $\alpha$.
>
> Empirically, this variance-reduction pattern extends beyond closed-form cases: on correlated Gaussian, Bayesian logistic, the 1000-mode high-dimensional Gaussian mixture, the CIFAR-10 EBM, and Static MNIST, SRMC consistently outperforms the tuned baseline.
>
> ---
>
> ### Q3
>
> Please refer to the response to Reviewer KmAj (W2 & Q2), where we now provide both a fixed-budget theoretical discussion and matched CPU-time comparisons.
>
> ---
>
> ### Q4
>
> The single vector $\theta\_n$ is designed to capture first-order directional imbalance in visited score space and is not meant to reconstruct the full empirical measure. The tradeoff is that SRMC can miss higher-order or mode-specific occupancy structure where more than one dominant score direction is needed to speed up the exploration. At the same time, the experiments show that this first-order summary is already strong enough to deform the local drift field, reduce repeated reinforcement of over-visited directions, and improve both MSE and mode coverage on the main continuous and discrete benchmarks. The richer feature for each state is regarded as future directions.

---

> > ### Author Rebuttal · Reviewer_WBGY · 2026-04-02
> >
> > I sincerely thank the authors for their detailed rebuttal. All of my questions have been addressed properly. I think this is a solid paper, and I will raise my score accordingly.

---

### Decision · Program_Chairs · 2026-04-30

**Decision:**

Accept (spotlight)

**Comment:**

This paper proposes Score-Repellent Monte Carlo, a history-dependent MCMC method that uses a running average of past score evaluations to define a modified target distribution. The method serves as a wrapper around standard MCMC kernels and is supported by theoretical guarantees. Empirical results on both continuous and discrete targets show improvements in mean estimation and mode exploration over standard baselines.

The reviewers were unanimously positive. They appreciated the conceptual novelty of the approach, its flexibility as a wrapper around a broad class of samplers, and the strength of the theoretical contributions. Accordingly, I recommend acceptance.